# Scientific logic and spatio-temporal dependence in analyzing extreme precipitation frequency: Negligible or neglected?

Francesco Serinaldi[1,2]

[1]School of Engineering, Newcastle University, Newcastle Upon Tyne, NE1 7RU, UK
[2]Willis Research Network, 51 Lime St., London, EC3M 7DQ, UK

**Correspondence:** Francesco Serinaldi (francesco.serinaldi@ncl.ac.uk)

**Abstract.** Statistics is often misused in hydro-climatology, thus causing research to get stuck around unscientific concepts that hinder scientific advances. In particular, neglecting the scientific rationale of statistical inference results in logical and operational fallacies that prevent discerning facts, assumptions, and models, thus leading to systematic misinterpretation of the output of data analysis. This study discusses how epistemological principles are not just philosophical concepts but have very practical effects. To this aim, we focus on the iterated underestimation and misinterpretation of the role of spatio-temporal dependence in statistical analysis of hydro-climatic processes by analyzing the occurrence process of extreme precipitation ($P$) derived from 100-year daily time series recorded at 1,106 worldwide gauges of the Global Historical Climatology Network. The analysis contrasts a model-based approach compliant with the well-devised but often neglected logic of statistical inference and a widespread but theoretically problematic test-based approach relying on statistical hypothesis tests applied to unrepeatable hydro-climatic records. The model-based approach highlights the actual impact of spatio-temporal dependence and finite sample size on statistical inference, resulting in over-dispersed marginal distributions and biased estimates of dependence properties, such as autocorrelation and power spectrum density. These issues also affect the outcome and interpretation of statistical tests for trend detection. Overall, the model-based approach results in a theoretically coherent modeling framework where stationary stochastic processes incorporating the empirical spatio-temporal correlation and its effects provide a faithful description of the occurrence process of extreme $P$ at various spatio-temporal scales. On the other hand, the test-based approach leads to theoretically unsubstantiated results and interpretations along with logically contradictory conclusions such as the simultaneous equi-dispersion and over-dispersion of extreme $P$. Therefore, accounting for the effect of dependence in the analysis of the frequency of extreme $P$ has huge impact that cannot be ignored, and more importantly any data analysis can be scientifically meaningful only if it considers the epistemological principles of statistical inference such as the asymmetry between confirmatory and dis-confirmatory empiricism, the inverse probability problem affecting statistical tests, and the difference between assumptions and models.

# 1 Introduction

## 1.1 Epistemology of scientific inquiry: 'model-based' data analysis

Most of the methods reported in handbooks of applied statistics have been developed under the assumption of independence, distributional identity and stationarity, and well-behaving bell-shaped/exponential distributions. Of course, there is a wide statistical literature focused on dependence and related stochastic processes, lack of distributional identity and nonstationarity, and skewed sub-exponential distributions. However, in some applied sciences such as hydro-climatology, analysts have often neglected that moving from the former set of assumptions (commonly reported in introductory handbooks) to the latter is not just moving to a more general model from one that can be considered as a special case. A typical example is the Generalized Extreme Value (GEV) distribution widely used in the study of hydro-climatic extreme events, which converges to the Gumbel distribution as the shape parameter converges to zero. While Gumbel is mathematically a special case of the GEV, assuming that Gumbel is the distribution of choice has relevant consequences, as it means assuming exponential tails instead of super-exponential (upper bounded) or sub-exponential (possibly heavy). In particular, high values of skewness and heavy tails imply possible non-existence of the moments of high order as well as bias and/or high variability in the estimates of the moments themselves, including variance, covariance, and autocorrelation, as well as long range dependence (see e.g., Embrechts et al., 2002; Barunik and Kristoufek, 2010; Lombardo et al., 2014; Cirillo and Taleb, 2016; Taleb, 2020; Dimitriadis et al., 2021; Koutsoyiannis, 2023, and references therein).

Similar remarks hold for nonstationarity, which denotes dependence of the (joint) distribution function of a set of generic random variables $X_t$ on the parameter $t$, $-\infty < t < +\infty$, via well-specified functions of $t$ (e.g., Koutsoyiannis and Montanari, 2015; Serinaldi et al., 2018). For example, if $t$ denotes 'time', dealing with nonstationarity does not mean just adding time dependent parameters to a stationary model, using for instance Generalized Linear Models (GMLs) and their available extensions: it means that the ergodicity property, which is key in the interpretation of statistical inference, is no longer valid (e.g., Koutsoyiannis and Montanari, 2015). In these cases, any estimate of whatever summary statistics, such as the sample mean, is uninformative as it does not have a corresponding unique population counter part, as the latter does not exist anymore (e.g., Serinaldi and Kilsby, 2015).

As for the effects of nonstationarity and heavy tails, there is a wide literature on the effects of the assumption of dependence on statistical inference. Generally, dependence implies information redundancy and reduced effective sample size, along with variance inflation and bias of standard estimators of summary statistics such as marginal and joint moments (e.g., Koutsoyiannis, 2004; Lombardo et al., 2014; Dimitriadis and Koutsoyiannis, 2015, and references therein). Therefore, assuming spatio-temporal dependence means recognizing that such an assumption impacts on every sampling property of the process, including marginal distributions. Under spatio-temporal dependence, the classical estimator of the correlation itself is biased and need to be corrected (e.g., Marriott and Pope, 1954; White, 1961; Wallis and O'Connell, 1972; Lenton and Schaake, 1973; Mudelsee, 2001; Koutsoyiannis, 2003, 2011; Koutsoyiannis and Montanari, 2007; Papalexiou et al., 2010; Tyralis and Koutsoyiannis, 2011; Dimitriadis and Koutsoyiannis, 2015; Serinaldi and Kilsby, 2016a).

The foregoing examples highlight that the statistical inference cannot be reduced to the usage of multiple competing models and methods, as every aspect of statistical analysis and its interpretation depend on the underlying assumptions according to the rationale of statistical inference. Even the simplest diagnostic diagram relies on underlying assumptions and models (see e.g., Serinaldi et al., 2020a, for further discussion). This primary epistemological concept comes before any methodology and/or technicality and marks the boundary between correct interpretation and misinterpretation of inference results, and eventually

"*between engineering concepts dictated by expediency, and scientific truth*" (Klemeš, 1986). However, while it was routinely presented in statistical handbook published in the past century (e.g., Aitken, 1947; Cramér, 1946; Papoulis, 1991), most of the modern textbooks seem to miss it, perhaps taking it for granted. Nonetheless, in the hydro-climatological context, von Storch and Zwiers (2003, p. 69) well summarized those primary principles of statistical inference as follows:

1. "*A statistical model is adopted that supposedly describes both the stochastic characteristics of the observed process and the properties of the method of observation. It is important to be aware of the models implicit in the chosen statistical method and the constraints those models necessarily impose on the extraction and interpretation of information.*"

2. "*The observations are analysed in the context of the adopted statistical model.*"

These concepts are nothing but the specialization of the principles of scientific inquiry in the context of data analysis. As stressed by Box (1976), "*science is a means whereby learning is achieved, not by mere theoretical speculation on the one hand, nor by the undirected accumulation of practical facts on the other, but rather by a motivated iteration between theory and practice... Matters of fact can lead to a tentative theory. Deductions from this tentative theory may be found to be discrepant with certain known or specially acquired facts. These discrepancies can then induce a modified, or in some cases a different, theory. Deductions made from the modified theory now may or may not be in conflict with fact, and so on.*" Eventually, "*the sciences do not try to explain, they hardly even try to interpret, they mainly make models. By a model is meant a mathematical construct which, with the addition of certain verbal interpretations, describes observed phenomena. The justification of such a mathematical construct is solely and precisely that it is expected to work - that is, correctly to describe phenomena from a reasonably wide area. Furthermore, it must satisfy certain esthetic criteria - that is, in relation to how much it describes, it must be rather simple.*" (von Neumann, 1955).

Based on the foregoing remarks, appropriate statistical inference (and scientific learning) is an iterative 'model-based' procedure, which can be summarized as follows:

1. Make assumptions that are deemed to be reasonable for data and facts at hand.

2. Build tentative theories and models and make inference accounting for the effect and consequences of the underlying assumptions.

3. Interpret results according to the nature of the adopted assumptions and models.

4. Retain or change/update assumptions and models based on the agreement or disagreement of the developed theories and models with (new) data and facts.

This procedure should be iterated bearing in mind that the developed models should satisfy some formal criteria such as parsimony, accuracy, generality, and fit for purpose.

## 1.2 Box's cookbookery and mathematistry: 'test-based' data analysis

Being the prerequisite to any sound scientific investigation, the epistemological principles described in Section 1.1 should be obvious and well-known. However, this does not seem to be the case in some applied sciences where data analysis and modeling often neglect or ignore such principles, thus calling into question the scientific validity of results and conclusions.

In this respect, the statistician George E.P. Box highlighted that the scientific progress results from the feedback between theory and practice, and this feedback requires a closed loop. Therefore, when loop is open, progress stops. Box (1976)
referred to the main consequences of lack on feedback between theory and practice as maladies called 'cookbookery' and 'mathematistry', claiming that "*The symptoms of the former are a tendency to force all problems into the molds of one or two routine techniques, insufficient thought being given to the real objectives of the investigation or to the relevance of the assumptions implied by the imposed method... Mathematistry is characterized by development of theory for theory's sake, which since it seldom touches down with practice, has a tendency to redefine the problem rather than solve it... In such areas*
*as sociology, psychology, education, and even, I sadly say, engineering, investigators who are not themselves statisticians sometimes take mathematistry seriously. Overawed by what they do not understand, they mistakenly distrust their own common sense and adopt inappropriate procedures devised by mathematicians with no scientific experience*".

In the context of climate science, von Storch and Zwiers (2003) raised similar remarks in the preface of their book: "*Cookbook recipes for a variety of standard statistical situations are not offered by this book because they are dangerous for anyone*
*who does not understand the basic concepts of statistics*".

This problem is not new in hydrology and hydro-climatology either, and was already stressed by Yevjevich (1968) and Klemeš (1986), who discussed some misconceptions concerning the study and interpretation of hydrological processes and variables by methods borrowed from other disciplines such as systems/decision theory, mathematics, and statistics. For instance, often stochastic processes are no longer considered as convenient descriptors of hydrological processes for practical
purposes, but the former are identified with the latter and vice versa, thus generating confusion and a questionable approach to data analysis and modeling that contrasts with the logic of statistical inference recalled in Section 1.1.

A large body of the literature on data analysis of unrepeatable hydro-climatic processes seems to neglect or ignore epistemological principles, thus confusing the role of observations, assumptions, and models. This results in fallacious procedures that share the following general structure, which we call 'test-based' method:

1. Select several models and methods based on different and often incompatible assumptions.

2. Make inference neglecting the constraints imposed by different underlying assumptions.

3. Interpret results attempting to prove/disprove models' assumptions, which are often (if not always) erroneously attributed to physical processes, whereas they refer to models used to describe such processes.

This approach generally corresponds to a widespread mechanistic use of statistical methods/software and massive application of statistical hypothesis tests, which are not supported by the required epistemological and theoretical knowledge of the methodologies used. Such approach neglects that models cannot be used to prove or disprove their own assumptions in the same way a mathematical theory cannot prove or disprove its own axioms and definitions. This is because those models and theories are valid only under those assumptions, axioms, and definitions. Of course, specific models cannot even be used to prove or disprove alternative assumptions as they might not even exists under those alternative hypotheses.

## 1.3 Aims and organization of this study

While Yevjevich (1968) and Klemeš (1986) provided extemporaneous commentaries about the foregoing issues, we address the problem from a different perspective. Instead of discussing several misconceptions from a general and purely conceptual point of view, we focus on a specific issue (here, the role of dependence in statistical analysis of extreme $P$ occurrence), focusing on theoretical inconsistencies and showing practical consequences by performing a detailed data analysis. In this way, theoretical remarks are complemented with a side-by-side comparison of the output of 'model-based' and 'test-based' methods, emphasizing the concrete effect of conceptual mistakes. Therefore, this work is a proper neutral validation/falsification study (see e.g., Popper, 1959; Boulesteix et al., 2018, and references therein) that expands some existing literature about the independent check of the theoretical consistency in statistical methods applied in hydro-climatology (Lombardo et al., 2012, 2014, 2017, 2019; Serinaldi and Kilsby, 2016a; Serinaldi et al., 2015, 2018, 2020a, b, 2022).

Focusing on the assumption of spatio-temporal dependence in the analysis of extreme $P$ frequency, we attempt to show the practical consequences of underestimating and not properly considering and interpreting the effects of dependence as well as the logical fallacies of 'test-based' method in this context. In particular, we re-analyze a worldwide precipitation data set comparing the output of a 'model-based' framework (relying on theoretically-informed stochastic modeling and diagnostic plots) with a 'test-based' approach that led Farris et al. (2021) to conclude that "*accounting or not for the possible presence of serial correlation has a very limited impact on the assessment of trend significance in the context of the model and range of autocorrelations considered here*" and "*Accounting for serial correlation in observed extreme precipitation frequency has limited impact on statistical trend analyses*". Therefore, this study double-checks the role of scientific logic and spatio-temporal dependence in the analysis and characterization of extreme $P$ frequency at various spatio-temporal scales. Eventually, we compare the two methodologies ('model-based' and 'test-based') in terms of their rationale and output to better understand why and how epistemological fallacies and the consequent improper use of statistical analysis and treatment of dependence might result in misleading conclusions.

This study is organized according to the its specific purpose. Therefore, it does not follow the standard structure 'problem-model-application-results'. In particular, all technical details of models and methods are reported in the Supplement. Indeed, the aim is to emphasize the practical importance of epistemology in data analysis, rather than focusing on models' technicalities, which is one of the conceptual mistakes affecting 'test-based' analysis. In this respect, the specific models/methods used are secondary and replaceable with others, whereas the logical reasoning leading the analysis (but systematically neglected in 'test-based' approach) stays unchanged.

Based on the foregoing remarks, Section 2 introduces the $P$ data set. Section 3 presents and discusses the 'test-based' methodology, highlighting shortcomings and pitfalls that lead to introduce the rationale of a 'model-based"' approach, whose rationale is described in Section 4. In Section 5, we analyze the $P$ frequency data studying: (i) the marginal distribution of the annual number ($Z$) of over-threshold (OT) events; (ii) the relationship between lag-1 autocorrelation $\rho_1$ and slope $\phi$ of liner trend estimated on the $Z$ time series; and (iii) trend analysis on the $Z$ time series. In this respect, we expand the analysis reported by Farris et al. (2021) by focusing on the role of spatio-temporal dependence at various spatio-temporal scales (e.g., Koutsoyiannis, 2020; Dimitriadis et al., 2021). For each stage of the foregoing analysis, we further discuss the logical consistency or pitfalls of the used methodologies and provide empirical results. Finally, Section 6 reports general remarks about the interpretation of our study in the context of the existing literature on statistical trend analysis, and more generally about the problem of approaching statistical analysis of hydro-climatic data neglecting epistemological principles that are fundamental to properly set up the analysis itself.

## 2  Data

We analyze daily precipitation time series from a sub-set of gauges extracted from more than 100,000 stations of the Global Historical Climatology Network-Daily (GHCN-D) database (Menne et al., 2012a, b) (https://www.ncei.noaa.gov/data/global-historical-climatology-network-daily/). To allow a fair comparison with existing literature, we extracted 1,106 worldwide gauges characterized by at least 95 complete years of records in a common 100-year period from 1916 to 2015, using the same criteria applied by Farris et al. (2021). Figure 1 displays the map of the selected GHCN stations with indication of four sub-regions denoted as 'North America', 'Eurasia', 'Australia', and 'North-Western Europe'. The first three regions are identified to be as close as possible to those used by Farris et al. (2021) in their regional analysis, while 'North-Western Europe' is an additional region corresponding to the most densely gauged area of 'Eurasia'. Overall, the four regions along with the worldwide scale (hereinafter denoted as 'World') allow us to highlight the behavior of extreme $P$ in nested regions.

To allow a fair comparison with existing literature, we followed Farris et al. (2021) and selected extreme $P$ as the values exceeding given percentage thresholds according to the at-site empirical cumulative distribution function (ECDF) (including zeros). For each station, the annual number $Z$ of OT exceedances forms the time series of extreme $P$ frequencies. Note that the "*exceedances on consecutive days are counted as separate events*" (Farris et al., 2021). Farris et al. (2021) considered several percentage threshold from 90% to 97.5% and different sub-sets (i.e., the most recent 30 and 50 years as well as the complete sequences of 100 years). Since their results are consistent across different thresholds, we limit our analysis to 95% and 99.5% thresholds, as the former is the one discussed more extensively by Farris et al. (2021), while the latter serves to highlight the behavior of the occurrence process of $P$ exceedances over a high threshold. As far as the number of years is of concern, we only use 100 and 50 years, as shorter time series of 30 annual data points do not provide reliable information on occurrence processes in terms spatio-temporal properties. The GHCN-D data set was retrieved and handled by the R contributed package rnoaa (Chamberlain, 2020).

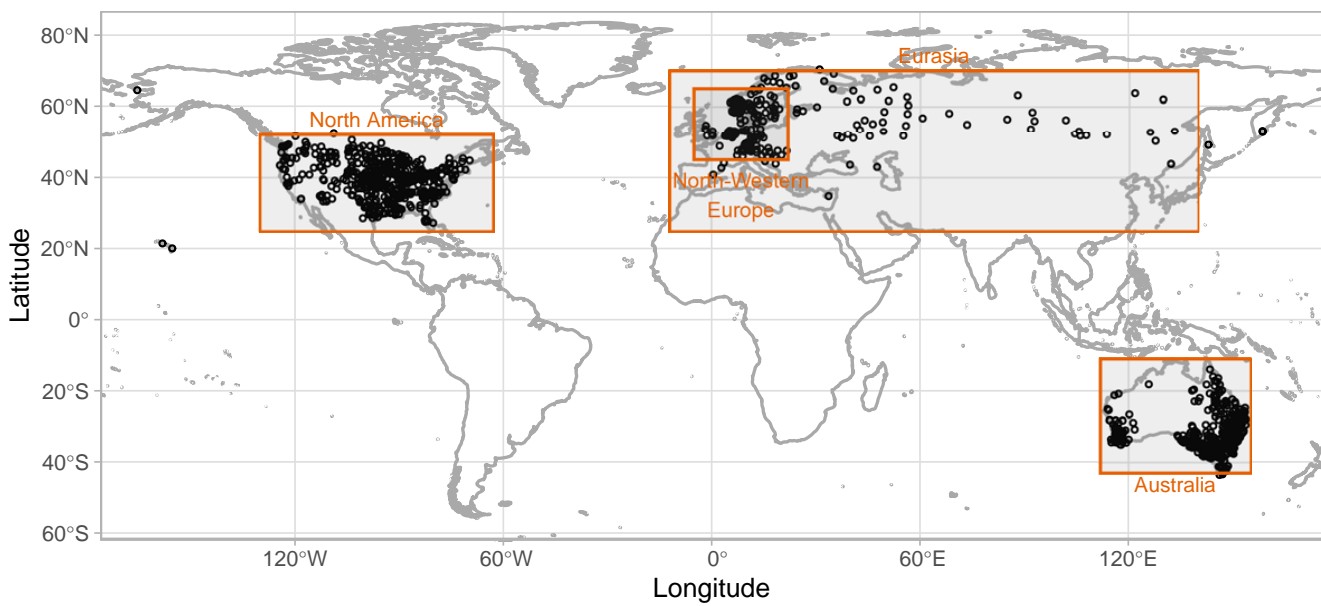

**Figure 1.** Map of GHCN rain gauges used in this study with indication of the four sub-regions (denoted as 'North America', 'Eurasia', 'North-Western Europe', and 'Australia') that are discussed in the subsequent analysis.

## 3   Test-based methodology

### 3.1   Assumptions and statistical tests

Before introducing the model-based methodology, we firstly present the test-based approach to data analysis. The aim is two-fold: (i) to explain the motivation of moving from test-based to model-based analysis, and (ii) to better understand the discussion in Section 5 concerning the differences between the output of the two methodologies. For the sake of comparison, we apply the same test-based procedure used by Farris et al. (2021). It consists of the following steps:

- Firstly, Kolmogorov-Smirnov and $\chi^2$ goodness-of-fit tests are used to check whether $Z$ follow a Poisson distribution.

- Based on the outcome of the first step, two competing models are selected: (i) Nonhomogeneous Poisson (NHP) process describing a collection of independent Poisson random variables with rate of occurrence linearly varying with time, and (ii) first-order Poisson integer autoregressive (Poisson-INAR(1)) process, which is a specific kind of stationary and temporally correlated process with Poisson marginal distribution (see Section S1 in the Supplement). These models are used to study the effect of serial correlation.

- Therefore, various statistical tests for trend detection are applied. Due to similar performance of parametric and nonparametric tests, Farris et al. (2021) retained and discussed only one test for each category, that is, a nonparametric test based on Mann-Kendall (MK) test statistic, and a parametric test based on the slope parameter of a Poisson regression (PR).

Incidentally, when performing a statistical test at the $\alpha$ significance level many times, we expect a percentage $\alpha$ of false rejections on average. Since all tests are applied to multiple time series, False Discovery Rate (FDR) method, that is, the ratio of number of false rejections to all rejections (Benjamini and Hochberg, 1995), is used to account for the effect of test multiplicity which is also known as field significance (Wilks, 2016; Farris et al., 2021).

The aim of this trend analysis of $Z$ is to investigate the impact of serial correlation on trend detection and the effect of 205 performing multiple tests. Even though technicalities (such as the selection of a particular test or model) can change from case to case, it is easy to recognize that the foregoing procedure follows the same test-based rationale of trend analyses reported in the majority of the literature on this topic. However, are we sure that such way of analyzing data is compliant with scientific method and its logic? Can we consider test-based analysis theoretically consistent solely due to its widespread application? In the next section, we start to answer these questions.

### 3.2 Remarks on logical fallacies of test-based methodology

As mentioned in Section 1, every statistical analysis (including diagnostic plots) relies on some assumptions, and these assumptions lead the interpretation of results. In this respect, while the simultaneous application of the techniques/tests listed in the previous section seems to be reasonable, a closer look reveals that the assumptions behind these statistical methods are not compatible to each other and might yield logical contradictions.

For example, Kolmogorov-Smirnov and $\chi^2$ goodness-of-fit tests, which are used to check the suitability of Poisson distribution for $Z$, are valid under the assumption that the underlying process is independent and identically distributed. Therefore, if these tests pointed to lack of rejection of Poisson distributions, this would also imply independence and distributional identity (i.e., stationarity). In fact, if the process were not independent, dependence would generate information redundancy and overdispersion (variance larger than the mean) of the observed finite-size samples of $Z$, thus making the Poisson model unsuitable 220 from a theoretical standpoint. On the other hand, if the $Z$ process were time nonstationary (i.e., not identically distributed), we could not conclude that the distribution of $Z$ is a single, specific distribution, such as Poisson. In that case, the distribution of $Z$ can be at most conditionally Poisson (i.e., a Poisson distribution describes the process resulting from filtering nonstationarity out) or compound Poisson (i.e., a compound Poisson distribution describes the process resulting from incorporating nonstationarity by model averaging over time). Therefore, if the Poisson assumption is not rejected according to tests that im-225 ply independence and stationarity, we must necessarily exclude the alternative assumptions of dependence and nonstationarity, thus concluding that there is no reason to further proceed with any subsequent analysis of trends and/or dependence. This is the first logical contradiction of the test-based procedure described in Section 3.1.

Nonetheless, let us overlook the foregoing contradiction, and move to the second step of the procedure. If we assume that the $Z$ process is NHP, this model implies that the data follow a different Poisson distribution at each time step (see Section S1 230 in the Supplement). In this case, the overall distribution of the observed $Z$ is a compound Poisson, and more importantly the nonstationarity of NHP hinders the application of Kolmogorov-Smirnov and $\chi^2$ goodness-of-fit tests, as the process does not fulfill the underlying assumptions of these tests. Indeed, these tests can at most be applied to conditional processes, that is, to values resulting from filtering the effect of 'nonhomogeneity' out (see e.g., Coles, 2001, pp. 110-111). In the present

case, nonhomogeneity can consist of random fluctuations or well-defined evolution of the rate of occurrence of extreme $P$. To summarize, if we assume NHP (and therefore nonstationarity), the results of first step are theoretically invalid, and they should be discarded a priori. This is the second logical contradiction of the test-based procedure, and it is the dual counter part of the first one.

Similar remarks hold for the assumption of dependence. It is well known that dependence introduces information redundancy that impacts on goodness-of-fit tests, inflating the variance of their test statistics, which must be adjusted accordingly. Variance inflation affects any sampling summary statistics, including sample variance and auto-/cross-correlation as well as the shape of the distribution of $Z$, which is expected to be over-dispersed (Marriott and Pope, 1954; White, 1961; Wallis and O'Connell, 1972; Lenton and Schaake, 1973; Mudelsee, 2001; Koutsoyiannis, 2003; Koutsoyiannis and Montanari, 2007; Papalexiou et al., 2010; Dimitriadis and Koutsoyiannis, 2015; Serinaldi and Kilsby, 2016a, 2018; Serinaldi and Lombardo, 2020). Therefore, under dependence, the marginal distribution of $Z$ cannot be Poisson, and the Poisson-INAR(1) models are likely unsuitable models for $Z$. In other words, the preliminary application of goodness-of-fit tests neglecting the subsequent assumption of dependence yields results that are theoretically invalid also in this case because: (i) the distribution of the test statistics under independence are not valid under dependence, and (ii) Poisson distribution is not a valid candidate model for $Z$ under dependence. Therefore, assuming Poisson-INAR(1) models is not only incompatible with the first step of the test-based procedure but also with the its own assumption of dependence.

These remarks should clarify why adding or relaxing fundamental assumptions such as (in)dependence and (non)stationarity cannot be reduced to just introducing or removing (or setting to zero) some parameters. Changing these assumptions deeply changes the inferential framework as well as the expected properties of the observed processes. Generally, conclusions and results obtained under a set of assumptions $\mathbb{A}_1$ cannot be used to support further analysis and models that are valid under a different set of hypotheses $\mathbb{A}_2$. In fact, the results under $\mathbb{A}_1$ might be not valid under $\mathbb{A}_2$, and vice versa. In some cases, models/tests need to be adjusted and results updated accordingly. In other cases, models/tests used under $\mathbb{A}_1$ might not even exist under $\mathbb{A}_2$.

The foregoing discussion indicates that an analysis based on models/tests relying on mixed assumptions is prone to severe logical inconsistencies and misleading conclusions, thus suggesting that a proper statistical analysis should rely on well-specified assumptions, adopting inferences procedures and methods that agree with those assumptions, thus guaranteeing a coherent interpretation of results according to the genuine scientific logic recalled in Section 1.1. In the next sections, we introduce this kind of approach and further investigate the aforementioned issues and their practical consequences in real-world data analysis.

## 4    Model-based methodology: Recovering the seemingly forgotten scientific method

The approach described in Section 3.1 was referred to as 'test-based' as it generally involves extensive application of several statistical tests and massive use of Monte Carlo simulations, with little or no attention to exploratory data analysis and theoretical assumptions, and thus their consequences on the interpretation of results. Therefore, we move from statistical tests and

their binary and often uninformative output (see further discussion in Section 6.1) to a 'model-based' approach supported by preliminary theoretical considerations and simple but effective graphical exploratory analysis. The underlying idea is to avoid moving among different and possibly incompatible assumptions, focusing on a single one that is considered to be realistic for the process studied. Thus, we build theoretically supported models and methods that fulfill that set of assumptions, and check if the framework is able to reproduce key properties of the process of interest (here, $Z$) over a reasonable range of spatio-temporal scales. Note that this approach is nothing but the standard procedure of scientific inquiry (Aitken, 1947; Cramér, 1946; von Neumann, 1955; Box, 1976; Papoulis, 1991; von Storch and Zwiers, 2003), which is however seemingly forgotten in a large body of literature dealing with statistical analysis of hydro-climatic data.

As recalled in Section 1.1, the model-based approach implies the following steps: (i) introduce reasonable assumptions based on preliminary observation and knowledge of the process of interest; (ii), deduce models and diagnostic tools that are consistent with those assumptions; (iii) compare model output and observations; (iv) update assumptions and/or models based on the outcome of stage (iii); and (v) iterate the procedure if required.

In the present case, the first two steps of the foregoing procedure specializes as follows:

(i) Since precipitation process exhibits recognizable spatio-temporal patterns evolving over various spatio-temporal scales, the assumptions of independence and distributional identity are reasonably untenable. Therefore, we relax the assumption of independence for $Z$ while retaining that of stationarity. The aim is to keep the modeling task as simple as possible and check whether models/methods incorporating spatio-temporal dependence provide a reasonable description of $Z$.

(ii) Based on the assumption of dependence, we introduce models for marginal distributions and temporal dependence attempting to balance parsimony and generality. These models are complemented with diagnostic diagrams and statistical tests purposely selected to be consistent with the assumption of dependence. Note that the statistical tests are introduced in the model-based framework only to allow comparison with the test-based procedure, although they are not even applicable to data from unrepeatable hydro-climatic processes (see further discussion in Section 6.1).

Sections 4.1, 4.2, and 4.3 introduce the above-mentioned models and methods in more detail, highlighting their logical consistency against the logical contradictions affecting test-based methods.

## 4.1 Modeling marginal distributions

The choice of potential distributions for $Z$ should considers four factors: (i) size of the blocks of observations over which we compute $Z$ values, (ii) finite sample size of $Z$ records, (iii) threshold used to select OT events, and (iv) effect of dependence.

The number of OT events $Z$ is calculated over 365-day time windows, meaning that $Z$ can be interpreted as the number of successes/failures occurring over a finite number of Bernoulli trials. The sample size of the resulting time series of $Z$ is at most 100, which is the number of available years of records, excluding possible missing values. Moreover, assuming a realistic average probability of zero daily $P$ equal to $p_0 = 0.7$ and OT probability $p = 0.95$ in the ECDF (including zeros), the corresponding nonexceedance probability of non-zero $P$ is $p_+ = \frac{p - p_0}{1 - p_0} \cong 0.83$, which is not a very high threshold for $P$ if one would like to focus on extreme values. For $p = 0.95$, the probability $p_+$ becomes at most $\cong 0.92$ for $p_0 = 0.4$, which is

300 quite a reasonable value for $p_0$ in wet climates (e.g., Harrold et al., 2003; Robertson et al., 2004; Serinaldi, 2009; Mehrotra et al., 2012; Olson and Kleiber, 2017). Therefore, the OT processes analyzed and the corresponding $Z$ unlikely fulfill the assumptions required by asymptotic models such as Poisson or compound Poisson (Berman, 1980; Leadbetter et al., 1983) in terms of sample size, block size, and threshold, whereas models devised for finite-size counting processes, such as Binomial, might be more appropriate.

More importantly, spatio-temporal dependence affects the marginal distribution of $Z$ and the (inter-)arrival times of OT events over finite-size blocks of observations (such as the 365 days forming one-year blocks). As mentioned in Section 3.2, spatio-temporal dependence results in information redundancy and over-dispersion, so that the distribution of (inter-)arrival times is expected to depart from exponential (which is instead valid for independent events) becoming sub-exponential Weibull-like (see e.g., Eichner et al., 2007, 2011; Serinaldi and Kilsby, 2016b, and references therein). Similarly, the distribution of

$Z$ departs from Binomial (or Poisson) and tends to be closer to over-dispersed distributions like the Beta-Binomial ($\beta\mathcal{B}$) distribution (see e.g., Serinaldi and Kilsby, 2018; Serinaldi and Lombardo, 2020). In particular, the $\beta\mathcal{B}$ model is a convenient theoretically-based distribution, as it is an extension of the Binomial distribution that accounts for over-dispersion by an additional parameter summarizing the average correlation over the spatio-temporal block of interest (see Section S2 in the Supplement and references therein).

Recalling that a Poisson distribution is characterized by equi-dispersion (i.e., equality of mean and variance), plotting sample variances ($\hat{\sigma}^2$) versus means ($\hat{\mu}$) can provide an effective diagnostic plot to check whether a Poisson distribution can be a valid model for $Z$.

To allow a direct and fair comparison with the test-based methodology described in Section 3.1, we complemented the Kolomogorov-Smirnov test with two additional tests whose test statistics are (i) the Pearson product moment correlation

coefficient (PPMCC) on the probability-probability plots (e.g., Wilk and Gnanadesikan, 1968), and (ii) the variance-to-mean ratio (VMR) $\sigma^2/\mu$, which is also known as index of dispersion (see e.g., Karlis and Xekalaki, 2000; Serinaldi, 2013, and references therein). For all tests, the reference hypothesis $\mathbb{H}_0$ is that the data are drawn from a Poisson distribution, although such an $\mathbb{H}_0$ is expected to be untenable according to foregoing discussion.

These tests are purposely chosen to highlight the inconsistencies resulting from a test-based methodology if we neglect the

325 rationale of the tests as well as informative exploratory analysis. In fact, as the PPMCC test relies on empirical and theoretical frequencies, that is, standardized ranks, it misses the information about the absolute values of $Z$, and it is therefore the less powerful test among the three. On the other hand, the KS test includes such information. However, it is also a general test devised for any distribution, while the VMR test focuses on a specific property characterizing the Poisson distribution, meaning that it is specifically tailored for the problem at hand. Therefore, the VMR test tends to have higher discrimination power under

330 the expected over-dispersion of correlated OT events. Indeed, it was found to be the most powerful among several alternatives in these circumstances (Karlis and Xekalaki, 2000; Serinaldi, 2013).

For each location, the distributions of the test statistics under $\mathbb{H}_0$ are estimated by simulating $S = 1000$ samples with the same size of the observed $Z$ time series from a Poisson distribution with rate parameter equal to the observed rate of occurrence.

## 4.2 Modeling dependence and nonstationarity: Distinguishing assumptions and models

The aim of the test-based methodology described in Section 3.1 is to investigate the nature of possible trends in $Z$ time series. The underlying idea is that such trends could be spurious effect of serial dependence or vice versa the serial dependence could be a spurious effect of deterministic trends. Therefore, Poisson-INAR(1) model is used to check the former case, while NHP the latter.

While this approach seems to be reasonable at a first glance, it suffers from technical problems related to neglecting formal 340 definitions of 'stationarity' and 'trend' along with clear distinction between population and sample properties. Note that the word 'stationarity' used throughout this study refers to the formal definition given by Khintchine (1934) and Kolmogorov (1938), which is the basis of theoretical derivations in mathematical statistics. The word 'stationarity' is often used with different meanings in hydro-climatological literature. However, such informal definitions do not apply to statistical inference and might generate confusion. These issues are discussed in depth by Koutsoyiannis and Montanari (2015), Serinaldi et al. 345 (2018, sections 4 and 5), and references therein (see also Section S3 in the Supplement for a summary). Here, we focus on additional epistemological aspects that precede such technical issues and call into question the underlying rationale of the test-based approach independently of specific models and tests used.

As mentioned in Sections 1 and 3.2, a selected statistical model should describe the stochastic properties of the observed process, and results should be interpreted according to the constraints posed by model assumptions. In the case of $Z$, the 350 underlying question is whether possible 'monotonic' fluctuations are deterministic (resulting from a well identifiable generating mechanism) or stochastic (as an effect of dependence, for instance). In the former case, we work under the assumption of independence and nonstationarity, whereas in the latter under the assumption of dependence and stationarity. Both assumptions are very general and correspond to a virtually infinite set of possible model classes and structures. Therefore, we need to recall that:

1. Every model developed under a specific set of assumptions is only valid under its own set of assumptions and cannot be used to validate the assumptions it relies on, as it cannot exist under different assumptions. For example, Poisson-INAR(1) cannot be used to assess the stationarity assumption, as it is not defined (it does not exist) under nonstationarity.

2. No specific model can be representative of the infinite types of models complying the same set of assumptions. For example, if Poisson-INAR(1) models do not provide a good description of $Z$, this does not exclude that other dependent 360 and stationarity models with different marginal distribution and linear or nonlinear dependence structure can faithfully describe $Z$.

3. Every model developed under a specific set of assumptions cannot provide information about different sets of assumptions. For example, if we assume independence and nonstationarity, and show that the NHP models describe all the properties of interest of an observed process, we can conclude that the NHP models provide a good description of data, 365 but we cannot say anything about the performance of models complying the assumption of dependence and stationarity, and vice versa.

Therefore, using Poisson-INAR(1) and NHP (with linearly varying rate of occurrence) to assess stationarity or nonstationarity of $Z$ is conceptually problematic for two key reasons:

- Stationarity and nonstationarity are not properties of the observed hydro-climatic processes (finite-size observed time series), but assumptions of the models we deem suitable to describe physical processes.

- Poisson-INAR(1) and NHP are valid only under their own assumptions, and do not represent the entire classes of possible stationary and nonstationary models. Therefore, discarding one of them does not imply invalidating their own assumptions (and the corresponding infinite classes of models), and, for sure, it does not allow invalidation of the assumptions of the alternative model, as each model could not even exist under the assumptions of the other one.

In a model-based approach, we do not compare specific models that are valid under different assumptions, but try to find models that describe as closely as possible the observed processes under a specific set of assumptions that are considered realistic. In this context, the Poisson-INAR(1) model is not a suitable option, as the marginal distribution of $Z$ is expected to be over-dispersed under dependence, and the first-order autoregressive structure can be too restrictive. In other words, while Poisson-INAR(1) is legitimate from mathematical perspective, it lacks conceptual consistency with the investigated process.

In this study, we use so-called 'surrogate' data to represent the class of stationary dependent models, minimizing the number of additional assumptions and constraints. In particular, we apply Iterative Amplitude Adjusted Fourier Transform (IAAFT), which is a simulation framework devised to preserve (almost) exactly the observed marginal distribution and power spectrum, and therefore autocorrelation function (ACF), under the assumption of stationarity (see Section S4 in the Supplement). In the present context, IAAFT can be considered a 'semi-parametric' approach. Indeed, it does not make any specific assumption on the shape of the distribution of $Z$ (preserving the empirical one), while a parametric dependence structure is needed to correct the bias of the empirical periodogram caused by temporal dependence (see Section S4 in the Supplement).

If the model of choice is well devised, it is expected to mimic the observed fluctuations of $Z$, including apparent trends. It follows that the statistical tests for trends used in the test-based methodology are expected to yield rejection rate close the nominal significance level. Here, we use IAAFT samples as a more general stationary alternative closer to the observed time series in terms of marginal distribution and ACF. IAAFT simulations are used to derive the sampling distributions of the MK and PR tests under the assumption of temporal dependence.

## 4.3 Field significance under dependence

The FDR approach is devised to control the rate of false rejections with respect to the number of rejections rather than the significance level, that is, the rate of false rejections with respect to the total number of performed tests. As a consequence, Benjamini and Hochberg (1995) highlighted that "*The power of all the methods* [Bonferroni, Hochberg, and Benjamini-Hochberg methods] *decreases when the number of hypotheses tested increases – this is the cost of multiplicity control*". However, the decreased power does not justify the recommendation of going back to at-site results, as suggested in the literature (Farris et al., 2021), thus overlooking *de facto* the field significance. In fact, the latter might be affected by unknown factors generating spurious results of statistical tests at local, regional, or global scale. In this respect, looking for instance at clusters of rejections of the

hypothesis of 'no trend', whereby trends have the same sign in a given area, as suggested for instance by Farris et al. (2021), might be misleading as this is exactly the expected behavior under spatio-temporal dependence or dependence on an exogenous (common) forcing factor (see e.g., Serinaldi and Kilsby, 2018). In other words, the dependence of extreme $P$ upon large-scale processes does not increase the 'power' (evidence) of regional trend analysis. On the contrary, it is a sign of redundancy, as the common local/regional behavior of multiple $P$ series in a given area might just be the expression of the common forcing factor (e.g., regional weather systems) driving the $P$ process. However, since the effect cannot precede its cause, the question is no longer if local trends in the $P$ process in a given area are similar/homogenous but what is the nature of the possible trends of the forcing factor. In this respect, local/areal effects (e.g., homogeneous regional patterns of $P$) can only reflect the behavior of their common cause and cannot provide information about the nature of the cause itself. Data analysis in the next sections further clarifies these issues.

Technically, the output of all statistical tests is analyzed in terms of $p$-values and FDR diagrams reporting the sorted $p$-values versus their ranks (Wilks, 2016, fig. 3). When needed, we also report results at local significance level $\alpha = 0.05$. This allows a fair comparison with results reported in the existing literature as well as a discussion about global (field) significance $\alpha_{\text{global}}$ and FDR control level $\alpha_{\text{FDR}}$ (Wilks, 2016).

### 4.4 Summary of model-based analysis

Summarizing Sections 4.1, 4.2, and 4.3, in a model-based approach, marginal distributions are parametrized by $\beta\mathcal{B}$ models (when needed), which are consistent with the assumption of spatio-temporal dependence. Goodness of fit is checked by suitable diagnostic diagrams, such as plots of $\hat{\sigma}^2$ versus $\hat{\mu}$ and probability plots, and statistical tests purposely devised to discriminate under-/equi-/over-dispersion.

For the sake of comparison with results reported in the existing literature, IAAFT is used to simulate synthetic samples of $Z$. This simulation method is 'semi-parametric' in the sense that it uses the empirical marginal distributions of $Z$, whereas a Hurst-Kolmogorov parametric dependence structure (also known as fractional Gaussian noise (fGn), Koutsoyiannis, 2003, 2010; Iliopoulou and Koutsoyiannis, 2019) is used to allow the correction of the bias of the empirical periodogram caused by temporal dependence (see Section S4 in the Supplement).

IAAFT samples are used to build the empirical distribution of the test statistics of MK and PR tests accounting for the effect of dependence in trend analysis. Such tests are performed both locally and globally, accounting for test's multiplicity via FDR. Note that the use of statistical tests is not much meaningful in the context of unrepeatable processes. They are used for the sake of comparison with test-based approach to highlight their inherent redundancy and/or inconsistency.

Finally, we stress once again that, in a model-based approach (i.e. standard statistical inference as it should be), the choice of the foregoing candidate models and methods was based on theoretical considerations about the effect of finite sample size, threshold selection, and dependence discussed in Sections 4.1, 4.2, and 4.3. This contrasts with the test-based approach, which relies instead on several distributions, diagnostic plots, and statistical tests that correspond to heterogeneous assumptions. Such test-based methods are generally selected without paying attention to their fit-for-purpose, and they are used neglecting the

effect of their assumptions on the inference procedure. This approach yields contradictory results that are further discussed in Section 5.

## 5   Data analysis

### 5.1   Marginal distribution of extreme $P$ occurrences: Poissonian?

As mentioned in Section 4, preliminary exploratory analysis based on simple but effective diagnostic plots is often neglected even though it might be more informative than the binary output of statistical tests. Graphical diagnostics are a key step of the model-based approach. Concerning the marginal distribution of $Z$, the most obvious preliminary check consists of checking under-/equi-/over-dispersion.

Figures 2a-f show the diagrams of variance versus mean ($\hat{\sigma}^2$ versus $\hat{\mu}$), comparing the scatter plots corresponding to observed OT values over the 95% and 99.5% thresholds with those corresponding to samples of the same size drawn from Poisson, NHP, and $\beta\mathcal{B}$ distributions with parameters estimated on the observed samples. In more detail, for each of the 1,106 records, 100 synthetic time series of $Z$ are simulated from Poisson, NHP, and $\beta\mathcal{B}$ models, thus calculating the ensemble averages of the 100 sample means and variances estimated for each simulated sample of $Z$. Figures 2a-f displays such ensemble averages as circles along with horizontal and vertical segments denoting the range of mean and variance values obtained over 100 simulations for each of the 1,106 records.

As expected, the variance and mean of observed $Z$ are not aligned along the theoretical 1:1 line (dashed line) characterizing the Poisson behavior, not even considering the sampling uncertainty (Figures 2a-f). The patterns of observed means and variances of OT data do not even match those of NHP samples, which under-represent the expected and remarkable over-dispersion (Figure 2b and 2e). On the other hand, the $\beta\mathcal{B}$ distribution provides variance values closer to the observed ones, indicating that the variance inflation is consistent with the assumption of temporally dependent occurrences, as expected from preliminary theoretical considerations.

For the sake of completeness, we also considered the peaks over threshold (POT), that is, the maximum values of independent clusters of OT values, where each cluster is considered as a single event. Clusters are identified as sequences of positive values of daily precipitation separated by one or more dry days. Different inter-arrival times for cluster identification can be used, but this parameter is secondary in the present context.

The Poisson distribution might be a valid asymptotic model for peaks over high threshold under suitable conditions (e.g., Davison and Smith, 1990), whereas there is no theoretical argument supporting its validity for OT data, especially if they are expected to be dependent, and the used threshold is not very high. Sample means and variances of POT show a behavior closer to Poisson and NHP for both thresholds (Figure 2g-2l). However, Poisson and NHP models still under-represent the observed variances for the 95% threshold, while $\beta\mathcal{B}$ distributions provide a faithful reproduction. According to the foregoing theoretical remarks, this is not surprising, as the 95% threshold is not high enough to identify independent events, resulting in occurrences that are still temporally dependent. As expected, observed mean and variances of POT start to exhibit a Poissonian behavior only for the high 99.5% threshold. In this case, both NHP and $\beta\mathcal{B}$ provide results close to those of the Poisson distribution, as

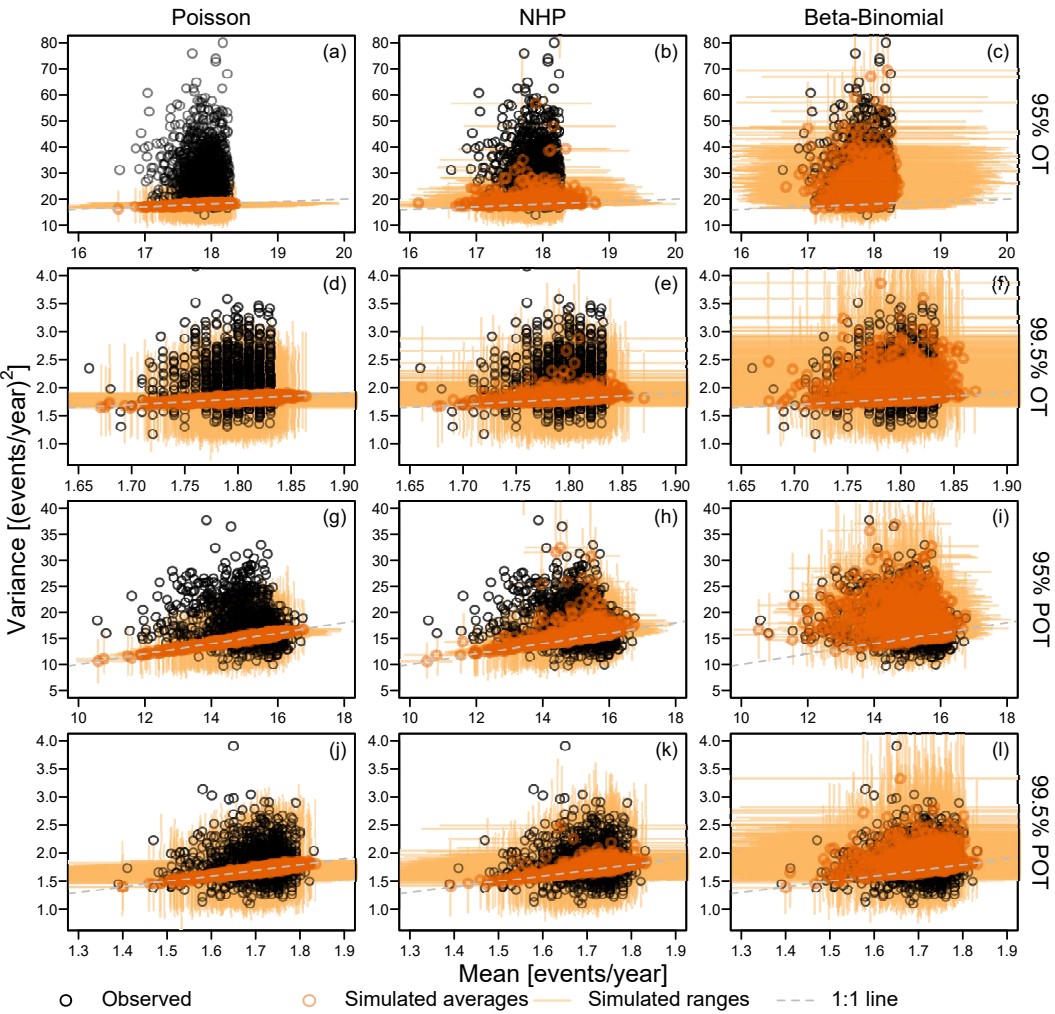

**Figure 2.** Diagrams of sample variance versus sample mean of the annual occurrences of OT values and POT for the 95% and 99.5% thresholds. Observed values (in black) are compared with those corresponding to simulated samples from Poisson, NHP, and Beta-Binomial ($\beta\mathcal{B}$) distributions (in orange). Orange circles 'o' denote ensemble averages, while horizontal and vertical segments denote the range of mean and variance values obtained over 100 simulations for each of the 1,106 records. Dashed gray lines '– – –' indicate the 1:1 lines representing the equality of mean and variance characteristic of Poisson distributions.

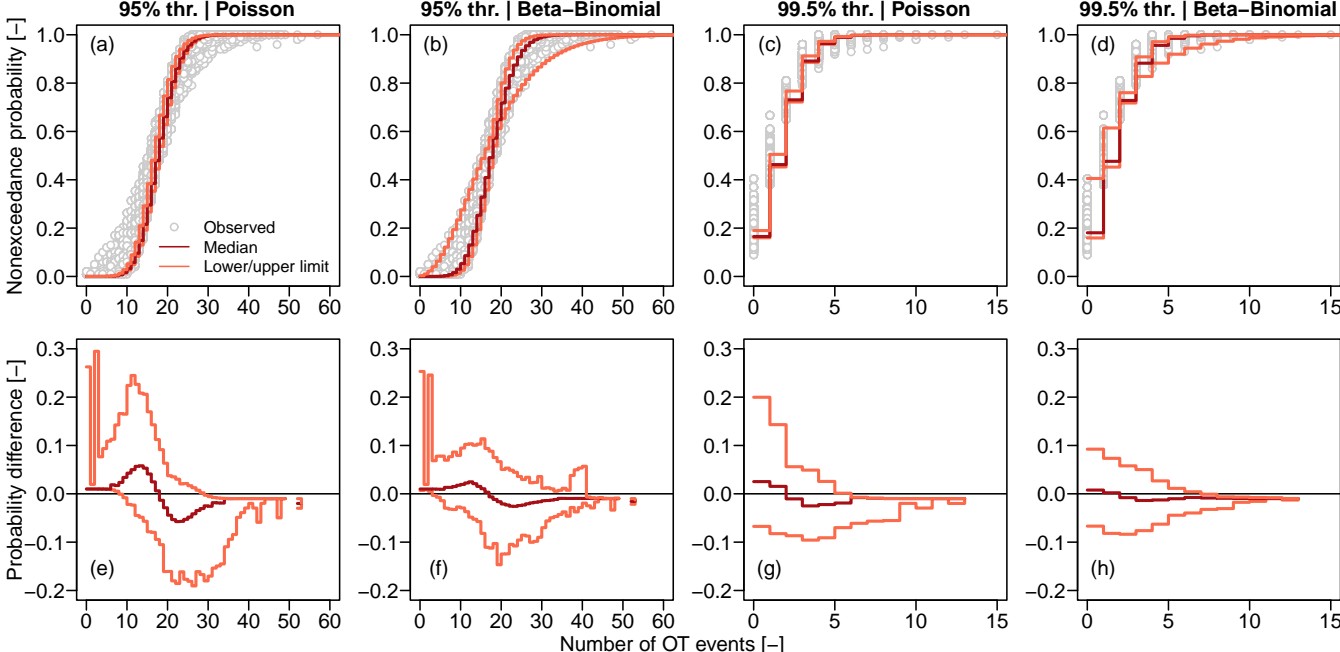

**Figure 3.** Panels (a-d) show the ECDFs of $Z$ for the 95% and 99.5% thresholds and each of the 1,106 rain gauges. ECDFs are complemented with the median, lower and upper limits of the ensemble of Poisson and Beta-Binomial ($\beta\mathcal{B}$) models corresponding to each rain gauge. Panels (e-h) show median, lower and upper limits of the differences $\delta(z)$ between ECDFs and Poisson and $\beta\mathcal{B}$ distributions. Medians, lower and upper values are computed point-wise for each value $z$ of the number of OT events $Z$.

the parameters accounting for nonstationarity in NHP and autocorrelation in $\beta\mathcal{B}$ tend to zero, and these models converge to the Poisson distribution, which is a special case of both.

Since Poisson and NHP provide similar results in terms of VMR for all cases, we retain the former and use probability plots (probability versus quantiles) to further check the agreement between observed data and models. We compare ECDFs $F_n(z)$ with the CDFs $F_{\mathcal{P}}(z)$ and $F_{\beta\mathcal{B}}(z)$ of Poisson and $\beta\mathcal{B}$ models, respectively. Probability plots are complemented with diagrams of the differences $\delta(z) = F_n(z) - F_{\mathrm{model}}(z)$ versus $z$. For the 95% threshold, Figures 3a and 3b show that the Poisson distribution cannot account for the observed variability of the empirical distributions, while $\beta\mathcal{B}$ models cover the range of variability thanks to the additional parameter summarizing the intra-block autocorrelation. This is consistent with the interpretation of over-dispersion as an effect of temporal dependence already highlighted in Figure 2. The diagrams of $\delta(z)$ versus $z$ in Figures 3e and 3f confirm that the $\beta\mathcal{B}$ models are closer to the empirical distributions for a wider range of $Z$ values compared with Poisson. For the 99.5% threshold, we have similar results (Figures 3c, 3d, 3g, and 3h), indicating that the temporal dependence of the generating processes $Y$ still plays a role despite the apparently low correlation of $Z$ time series.

The outcome of our exploratory analysis disagrees with results of goodness-of-fit tests reported by Farris et al. (2021), who concluded that the hypothesis of Poisson distribution for $Z$ cannot be rejected in more than 95% of the gauges at $\alpha_{\mathrm{global}} = 0.05$

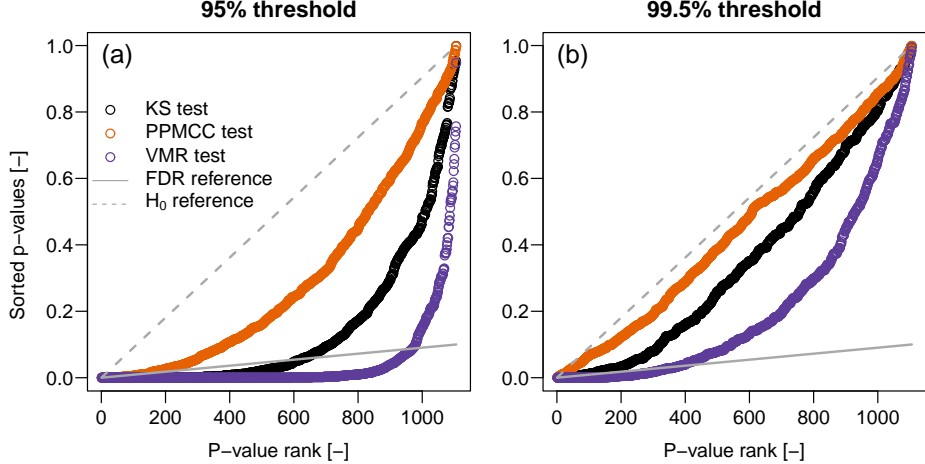

**Figure 4.** Illustration of FDR criterion for $\alpha_{\mathrm{FDR}} \cong 0.10$ (gray diagonal line) corresponding to $\alpha_{\mathrm{global}} \cong 0.05$ (Wilks, 2016). Plotted points are the sorted $p$-values of 1,106 local tests for KS, PPMCC, and VMR tests. Points below the diagonal lines represent significant results (i.e., rejections of $\mathbb{H}_0$) according to FDR control level.

for all values of percentage threshold and number of years. Therefore, we applied the three goodness-of-fit tests described in Section 4.1 to double check results of our exploratory analysis and allow a direct comparison. Figure 4 shows the FDR diagrams reporting the sorted $p$-values versus their ranks (Wilks, 2016, fig. 3). The number of $p$-values underneath the reference FDR line, which represent significant results according to the FDR control level $\alpha_{\mathrm{FDR}} = 0.10$ (Wilks, 2016), strongly depends on the specific test and threshold used. For the 95% threshold, the FDR rejection rates are 21%, 3%, and 53%, for KS, PPMCC,

and VMR tests, respectively. For the higher 99.5% threshold, the number of events decreases, and OT values tend to correspond to POT. Therefore, the rejection rates decrease, becoming 1%, 0.1%, and 12% for KS, PPMCC, and VMR tests, respectively. As anticipated in Section 4.1, these results highlight that the choice of statistical tests characterized by different power can lead to completely different and ambiguous conclusions.

Therefore, the theoretical arguments discussed in Section 4.1 and the foregoing exploratory analysis indicate that the $\beta\mathcal{B}$

distribution might be a good candidate distribution to describe $Z$, while neither Poisson nor NHP are suitable options. This calls into question the use of Poisson-INAR(1) model as a stationary dependent reference to be used in the subsequent trend analysis. Note that the emergence of $\beta\mathcal{B}$ distribution is inherently related to the assumption of serial dependence. In other words, even though we can build autocorrelated processes with whatever marginal distribution, and these models (e.g., Poisson-INAR(1)) can be technically correct from a mathematical standpoint, they are not necessarily consistent with the studied

process. In the present case, theoretical reasoning tells us that the distribution of $Z$ over finite-size blocks under the assumption of temporal dependence is over-dispersed and cannot be Poissonian. It follows that autocorrelated processes with Poisson marginal distributions are known in advance to be unsuitable for these OT processes, albeit they can be mathematically correct and suitable in other circumstances.

It is also worth noting how simple diagnostic plots supported by theoretical arguments concerning stochastic properties of the studied process provide more information than the binary output (rejection/no rejection) of whatever statistical test, and also help identification of consistent models. On the other hand, statistical tests might be misleading. They suffer not only from several logical and technical inconsistencies but also from trivial problems related to the their choice (see discussion in Section 6.1). In fact, while KS, PPMCC, and VMR tests seem to be suitable choices, they have very different power for the specific problem at hand, and can lead to contrasting conclusions, without providing insights about the actual nature of the investigated process.

## 5.2 Stationary or nonstationary models?

### 5.2.1 Linear correlation versus linear trends: Practical consequences of confusing assumptions with models

The test-based approach led Farris et al. (2021) to discard temporal dependence as a possible cause of apparent trends in $Z$ time series based on disagreement between trend slopes estimated on observed data and Poisson-INAR(1) simulated samples, respectively. In particular, their conclusion is based on diagrams of lag-1 autocorrelation coefficient $\hat{\rho}_1$ versus slope of linear trend $\hat{\phi}$ estimated on observed $Z$ time series, and sequences simulated by NHP and Poisson-INAR(1). Since the foregoing exploratory analysis indicated that NHP and Poisson-INAR(1) models are not consistent with the marginal distributions of $Z$, we used IAAFT samples as a more realistic alternative.

Figure 5a compares the observed pairs $(\hat{\rho}_1, \hat{\phi})$ with those resulting from Poisson-INAR(1) models. Figure 5a is similar to Figure 4a in Farris et al. (2021), which however compares observed pairs $(\hat{\rho}_1, \hat{\phi})$ with those corresponding to independent Poisson variables. Figure 5a confirms that the Poisson-INAR(1) models do not provide a good description of the observed behavior, as expected for models that cannot even reproduce marginal distributions. On the other hand, the pattern of the pairs $(\hat{\rho}_1, \hat{\phi})$ estimated from IAAFT samples matches that of the observed pairs much better (Figure 5b).

Following Farris et al. (2021), we also simulated 10,000 samples from (i) NHP model for fixed values of $\phi$ ranging between -0.2 and 0.2, (ii) Poisson-INAR(1) for $\rho_1$ ranging between 0 and 0.8, and additionally (iii) IAAFT. The 10,000 samples from the three models allow the estimation of two different conditional probabilities, that is, $\mathbb{P}[P_1 \leq \rho_1 | \Phi = \phi]$ and $\mathbb{P}[\Phi \leq \phi | P_1 = \rho_1]$, respectively. Therefore, we can estimate the confidence intervals (CIs) of the conditional variables $(\rho_1 | \Phi = \phi)$ for NHP and $(\phi | P_1 = \rho_1)$ for Poisson-INAR(1). Figures 5c and 5d show these point-wise CIs. Even though their comparison in unfair as they refer to different conditional distributions, Farris et al. (2021) discarded Poisson-INAR(1) as CIs of $(\rho_1 | \Phi = \phi)$ for NHP cover the observed pairs $(\hat{\rho}_1, \hat{\phi})$ better than CIs of $(\phi | P_1 = \rho_1)$.

However, as mentioned in Section 4.2, if a specific model in the class of stationary models does not fit well, this does not enable to discard the entire class. In fact, the conditional CIs of $(\rho_1 | \Phi = \phi)$ built from the IAAFT samples indicate that alternative stationary processes can yield results similar to NHP. On the other hand, IAAFT CIs of $(\phi | P_1 = \rho_1)$ are much wider than those from Poisson-INAR(1) samples, thus confirming that such a model is clearly inappropriate to describe $Z$. Therefore, the class of dependent stationary models and the assumption of stationarity cannot be discarded based on the poor performance of a single misspecified stationary model that does not even reproduce the marginal distribution of the observed data.

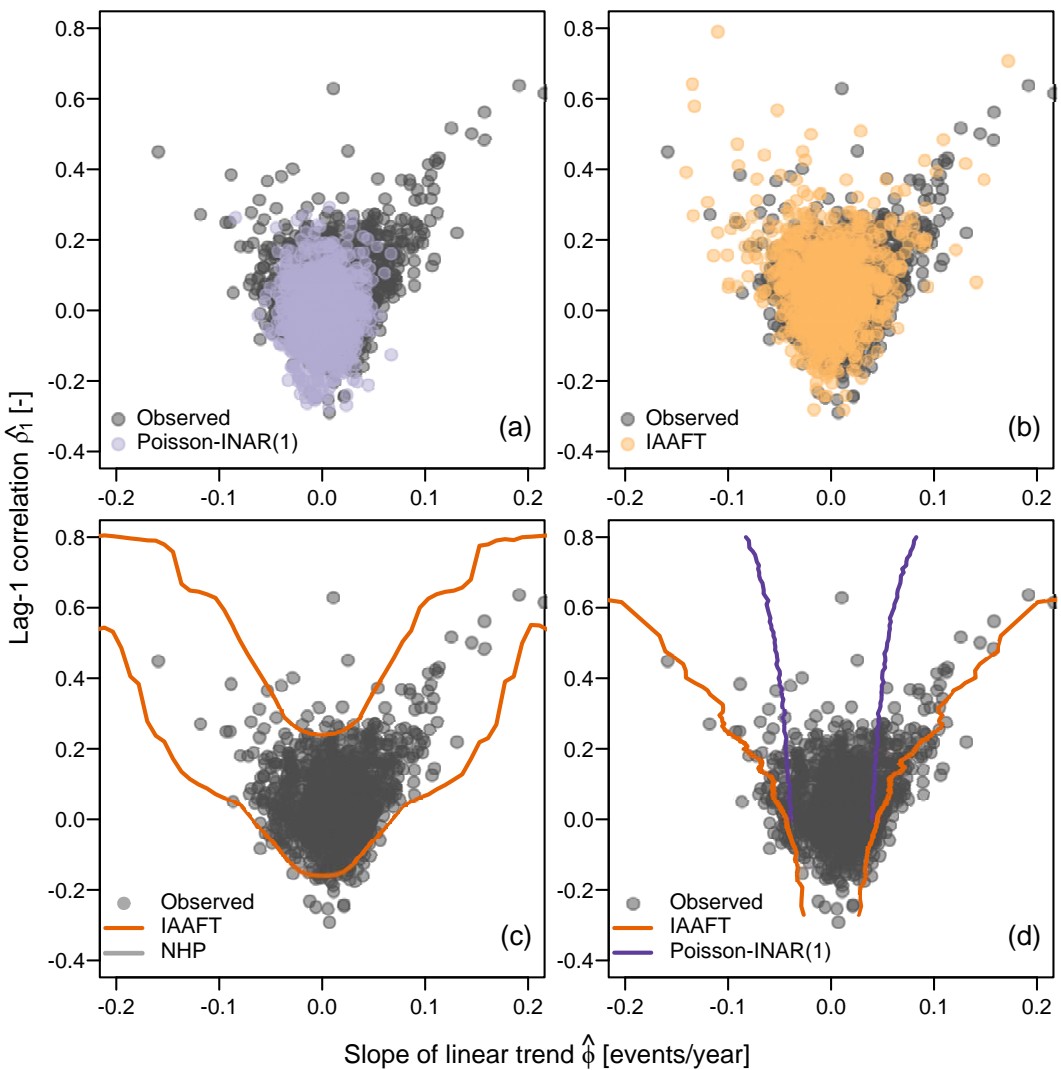

**Figure 5.** Scatter plots of the pairs $(\hat{\rho}_1, \hat{\phi})$ for the 1,106 observed $Z$ time series over the 95% threshold and 100 years (1916-2015) along with pairs corresponding to Poisson-INAR(1) samples (a), pairs corresponding to IAAFT samples (b), 95% CIs of $(\phi | \mathrm{P}_1 = \rho_1)$ for IAAFT and NHP (c), and 95% CIs of $(\rho_1 | \Phi = \phi)$ for IAAFT and Poisson-INAR(1) (d).

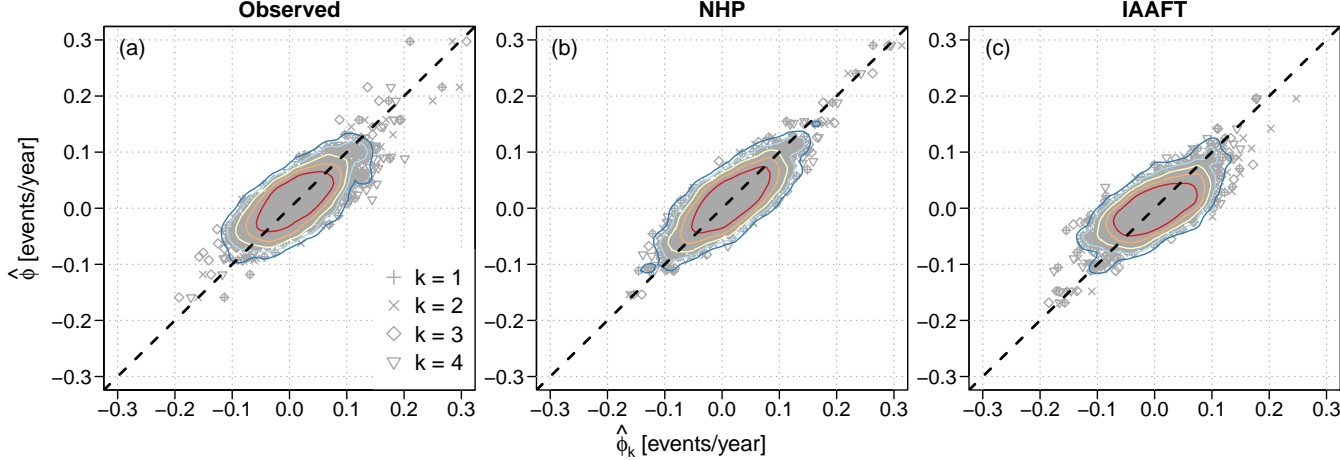

**Figure 6.** (a) Scatter plot of the pairs $(\hat{\phi}, \hat{\phi}_k)$, where $\hat{\phi}$ is the slope of the linear trend estimated on the observed $Z$ time series for the 95% threshold and 100 years, while $\hat{\phi}_k$ (with $k = 1, ..., 4$) are the slopes estimated on four sub-samples of length equal to 25 years. (b and c) Similar to (a) but for time series simulated from NHP and IAAFT, respectively.

The foregoing analysis is complemented by an additional one focusing on sub-samples. Following Farris et al. (2021), we sampled $Z$ values recorded every four years, thus extracting four sub-series of size 25, and therefore removing the effect of potential autocorrelation at lags from 1 to 3 years. For each sub-sample, we estimated the linear trend slopes $\hat{\phi}_k$ (with

$k = 1, ..., 4$) and plotted it against the slope estimated on the full series.

Figures 6a and 6b reproduce Figure 5 in Farris et al. (2021). Figure 6a shows the scatter plot of the pairs $(\hat{\phi}, \hat{\phi}_k)$ of the observed $Z$, while Figure 6b displays the pairs $(\hat{\phi}, \hat{\phi}_k)$ for synthetic series from the NHP model. The similarity of the patterns led Farris et al. (2021) to conclude that the serial dependence does not play any significant role, and observed trends are consistent with NHP behavior. However, the additional Figure 6c shows that also the pairs $(\hat{\phi}, \hat{\phi}_k)$ from stationary and temporally

correlated IAAFT samples exhibit a behavior similar to that of the observed $Z$, thus contradicting the foregoing conclusion.

Similarly to the analysis of pairs $(\hat{\rho}_1, \hat{\phi})$, the analysis of the pairs $(\hat{\phi}, \hat{\phi}_k)$ seems to be reasonable at a first glance. However, it suffers from similar inconsistencies:

– Four-year lagged sub-samples are supposed to be approximately independent based on the belief that the correlation is week and the value of ACF terms is generally low. However, this assumption misses the fact that, under dependence,

the ACF estimates on finite samples are negatively biased and need to be adjusted according to a parametric model of choice. In this respect, Figure 5 and the corresponding Figure 4 in Farris et al. (2021) are not even consistent because all panels show the $\rho_1$ values obtained by standard estimators that are only valid under independence, whereas the panels referring to Poisson-INAR(1) and IAAFT should show $\rho_1$ values adjusted for estimation bias. This further confirms that assumptions like (in)dependence and (non)stationarity influence not only the model parametrization (e.g., Poisson with

or without linear trend), but the entire inference procedure, including diagnostic plots and their interpretation.

- The performance of a specific independent nonstationary model (NHP) is incorrectly considered to be informative about the performance of an entire alternative class of dependent stationary models, while NHP is not even defined under those alternative assumptions. The ability of NHP to reproduce the patterns of $(\hat{\phi}, \hat{\phi}_k)$ cannot exclude the existence of equally good or better models based on different assumptions. The only way to understand if a class of models (and the underlying assumptions) is suitable for a given data set is to use credible members of such a class. This conceptual mistake is the same one affecting statistical tests, where the rejection of $\mathbb{H}_0$ is often misinterpreted and leads to uncritically embrace the alternative hypothesis $\mathbb{H}_1$, while rejection can be due to unknown factors that are not included in either $\mathbb{H}_0$ or $\mathbb{H}_1$ (see discussion in Section 6.1).

To summarize, our analysis shows that neither NHP nor Poisson-INAR(1) are suitable models for $Z$ in the considered range of thresholds. On the other hand, for POT and high thresholds, we recover the theoretically expected Poissonian behavior, and NHP and Poisson-INAR(1) models tend to converge to Poisson model, thus becoming almost indistinguishable.

## 5.3 Trend analysis of observed extreme $P$ occurrences: The actual role of spatio-temporal dependence

After analyzing marginal distributions and temporal dependence, we study spatio-temporal fluctuations of $Z$, comparing model-based and test-based methods. We expand the analysis of $Z$ (number of annual OT at each gauging location) considering the number of daily and annual OT events aggregated over the five regions described in Section 2 and shown Figure 1. This allows us to check if the assumption of dependence and the corresponding models provide a reasonable description of OT frequency over a range of spatio-temporal scales.

### 5.3.1 Trend analysis under temporal dependence

We analyze the presence of trends in $Z$ time series recorded at the 1,106 stations selected from the GHCN gauge network. For the sake of comparison with test-based results, trends are investigated applying the same MK and PR tests. However, the distributions of test statistics (and therefore critical values) are estimated from 10,000 IAAFT samples to properly account for the over-dispersion of the marginal distributions and temporal correlation. Moreover, tests are firstly performed at the local 5% significance level without applying FDR to check if the empirical rejection rate is close to the nominal one, as expected under correct model specification.

Figures 7 and 8 show the maps of trend test results for the 95% threshold, 100-year sample size, and the three regions North America, Eurasia, and Australia, with and without FDR, respectively (results for the other combinations of thresholds and sample sizes with and without FDR are reported in Figures S2-S7 in the Supplement). Results in Figures 7 and 8 are different from those reported by Farris et al. (2021) in their analogous Figure 9. To better understands such differences, we examine the rejection rates of both MK and PR tests for the four combinations of two thresholds (95% and 99.5%) and two sample sizes (50 and 100 years), and four different cases: ($\mathcal{A}$) critical values of test statistics obtained by IAAFT without bias correction of the autocorrelation and without FDR, ($\mathcal{B}$) critical values obtained by IAAFT with bias correction and without FDR, ($\mathcal{A}_{\text{FDR}}$) critical values obtained by IAAFT without bias correction and FDR, and ($\mathcal{B}_{\text{FDR}}$) critical values obtained by IAAFT with bias

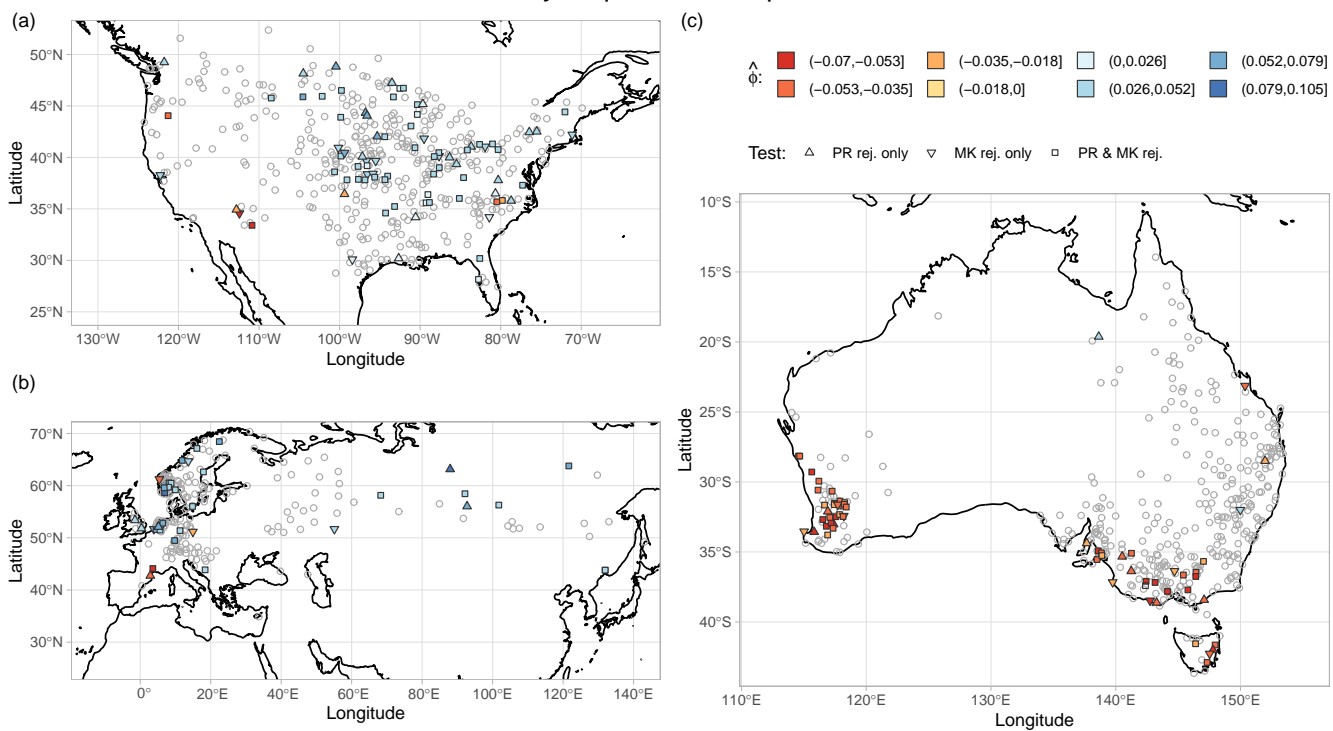

**Figure 7.** Maps of statistically significant trends at the GHCN gauges of the three regions North America (a), Eurasia (b), and Australia (c). Results refer to MK and PR tests applied to $Z$ time series for 100-year sample size and the 95% threshold without FDR. Statistical tests are performed at the local 5% significance level without applying FDR. The distributions of test statistics (and therefore critical values) are estimated from 10,000 IAAFT samples. Gray circles '∘' denote lack of rejection by both tests. Results for the other combinations of thresholds and sample sizes are reported in Figures S2-S4 in the Supplement.

correction and FDR (Table 1). The second case highlights the effect of neglecting the bias of classical ACF estimators under the assumption of temporal dependence, while the third and fourth ones show the effects of spatial correlation (albeit indirectly via FDR).

Focusing on case $\mathcal{A}$, for 50-year time series, local rejection rates are always close to the nominal 5%, as expected. For 100-year time series, local rejection rates corresponding to 95% and 99.5% thresholds reach a maximum of 22% and 14%, respectively. These values seems to be higher than expected. However, after correcting ACF bias (case $\mathcal{B}$), the maximum rejection rate for the 95% threshold drops to 13%, whereas the rejection rates for the 99.5% threshold stay almost unchanged. This is due to the higher (lower) autocorrelation of OT values corresponding to lower (higher) thresholds, and therefore stronger (weaker) bias correction. Overall, accounting for ACF bias results in rejection rates ranging between 9% and 13%. After considering (indirectly) the effects of spatial correlation via FDR (cases $\mathcal{A}_{\text{FDR}}$ and $\mathcal{B}_{\text{FDR}}$), the rejection rate drops to zero in all cases if we correct ACF bias (case $\mathcal{B}_{\text{FDR}}$), meaning that all tests are globally not significant at $\alpha_{\text{FDR}} \cong 0.10$. If we do not

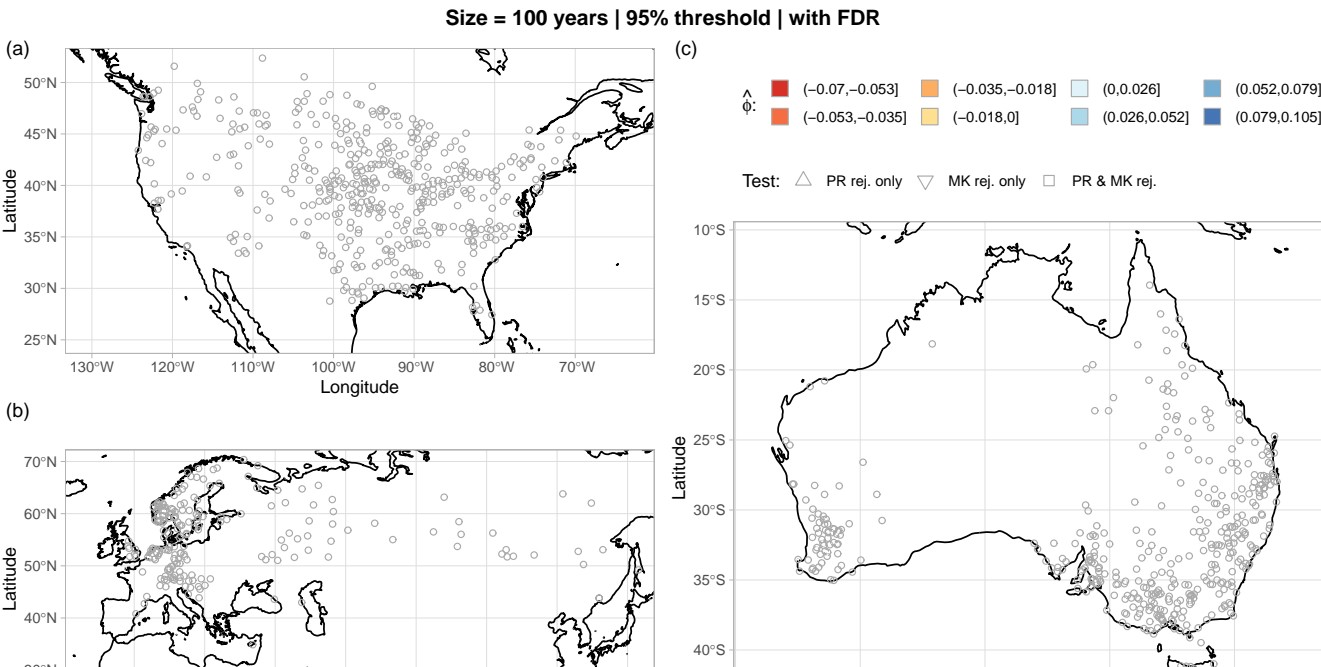

**Size = 100 years | 95% threshold | with FDR**

**Figure 8.** Similar to Figure 7 but with FDR. Results for the other combinations of thresholds and sample sizes are reported in Figures S5-S7 in the Supplement.

adjust ACF bias (case $\mathcal{A}_{\mathrm{FDR}}$) a small percentage of tests indicates global significance at $\alpha_{\mathrm{FDR}} \cong 0.10$ for sample size 100 and
595 threshold equal to the 95%. This is expected as the correction of ACF bias is more effective for larger sample sizes and lower thresholds. In fact, decreasing thresholds generally correspond to increasing temporal correlation of $Z$ samples, and larger samples allow a better quantification of the properties of a correlated process.

Some simple diagnostic plots can provide a clearer picture. Figure 9 shows the scatter plots of the pairs $(\hat{\rho}_1, \hat{\phi})$ with the rejections highlighted by different markers. Figure 9 is analogous to Figure 10 in Farris et al. (2021), but using IAAFT samples
and considering the cases $\mathcal{A}$, $\mathcal{B}$, $\mathcal{A}_{\mathrm{FDR}}$, and $\mathcal{B}_{\mathrm{FDR}}$. It shows how the number of rejections decreases as the effects of temporal and spatial correlation are progressively compounded. Focusing on case $\mathcal{B}$, rejections tend to occur for higher values of $|\hat{\phi}|$ conditioned to the value of $\hat{\rho}_1$, but there is no systematic rejection for all $|\hat{\phi}|$ exceeding a specified value as for the case $\mathcal{A}$. On the contrary, the pairs marked as 'rejections' overlap the pairs marked as 'no rejection' indicating that we can have both 'rejections' and 'no rejections' for time series with similar values of $\hat{\rho}_1$ and $\hat{\phi}$.
These results disagree with those reported by Farris et al. (2021), who found that the hypothesis of significant trend is always rejected for $|\hat{\phi}| > 0.05$ events/year, and concluded that "*the occurrence of the different cases is controlled by $\hat{\phi}$, while $\hat{\rho}_1$ is*

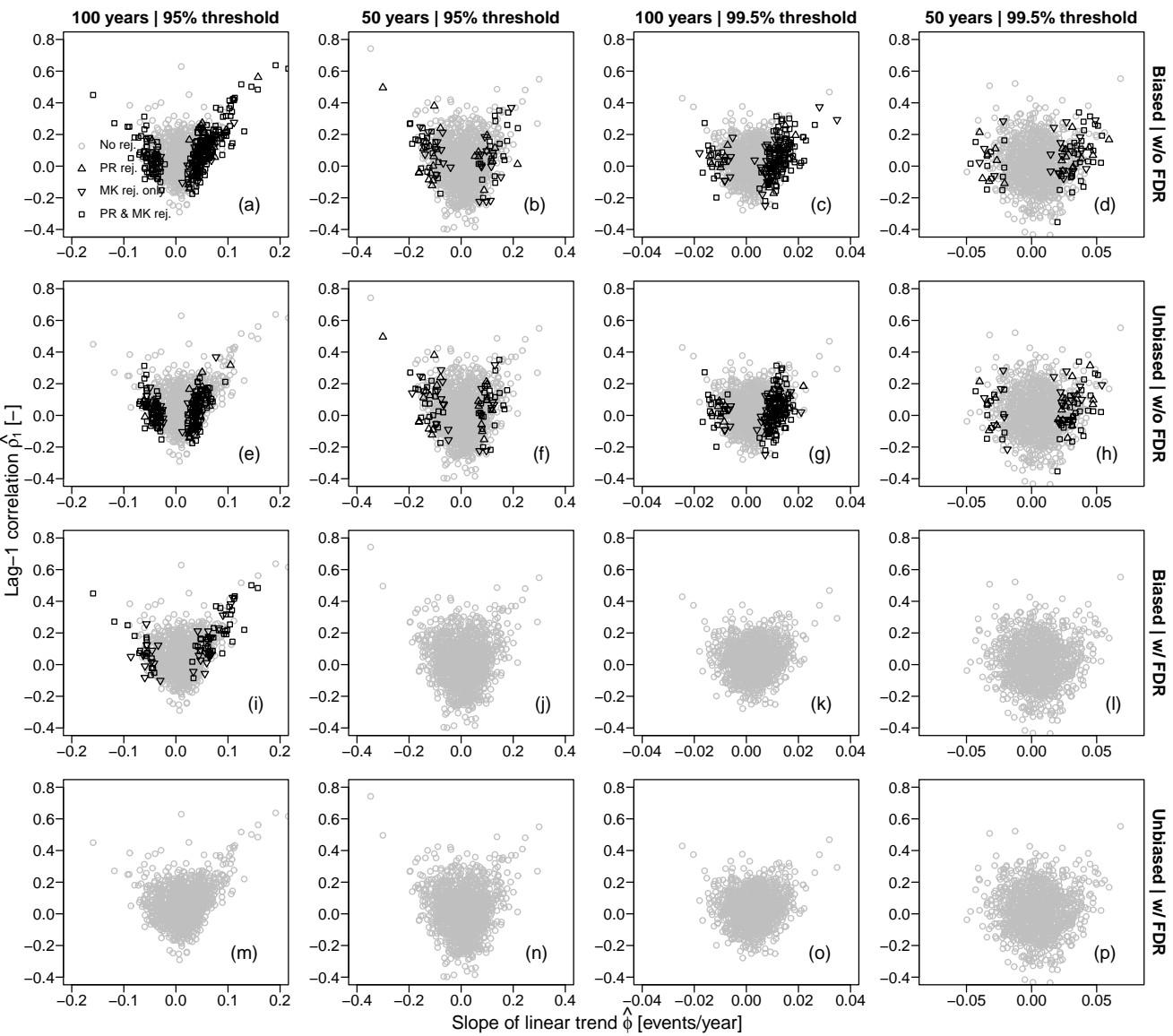

**Figure 9.** Scatter plots of the pairs $(\hat{\rho}_1, \hat{\phi})$ with the rejections highlighted by different markers. Markers refer to rejection of MK test, PR test, or both for $Z$ time series corresponding to the four combinations of two thresholds (95% and 99.5%) and two sample sizes (50 and 100 years) and the three cases $\mathcal{A} = \{\text{Biased ACF | w/o FDR}\}$, $\mathcal{B} = \{\text{Bias adjusted ACF | w/o FDR}\}$, $\mathcal{A}_{\text{FDR}} = \{\text{Biased ACF | w/ FDR}\}$, and $\mathcal{B}_{\text{FDR}} = \{\text{Bias adjusted ACF | w/ FDR}\}$. Gray circles 'o' denote lack of rejection by both tests.

**Table 1.** Rejection rates of PR and MK tests for the four combinations of two thresholds (95% and 99.5%) and two sample sizes (50 and 100 years) and different treatment of spatio-temporal dependence (cases $\mathcal{A} = \{$Biased ACF | w/o FDR$\}$, $\mathcal{B} = \{$Bias adjusted ACF | w/o FDR$\}$, $\mathcal{A}_{\mathrm{FDR}} = \{$Biased ACF | w/ FDR$\}$, and $\mathcal{B}_{\mathrm{FDR}} = \{$Bias adjusted ACF | w/ FDR$\}$)

| Test | Threshold [%] | Sample size [years] | Rejection rate [%] | | | |
|---|---|---|---|---|---|---|
| | | | case $\mathcal{A}$ | case $\mathcal{B}$ | case $\mathcal{A}_{\mathrm{FDR}}$ | case $\mathcal{B}_{\mathrm{FDR}}$ |
| PR | 95 | 100 | 22.3 | 13.4 | 3.0 | 0 |
| MK | | | 20.3 | 12.6 | 0 | 0 |
| PR & MK | | | 18.2 | 9.9 | 4.6 | 0 |
| PR | 95 | 50 | 6 | 5.8 | 0 | 0 |
| MK | | | 6 | 5.6 | 0 | 0 |
| PR & MK | | | 3.9 | 3.9 | 0 | 0 |
| PR | 99.5 | 100 | 13.7 | 12.6 | 0 | 0 |
| MK | | | 14.2 | 11.9 | 0 | 0 |
| PR & MK | | | 11 | 9.1 | 0 | 0 |
| PR | 99.5 | 50 | 5.5 | 5.4 | 0 | 0 |
| MK | | | 5.6 | 5.3 | 0 | 0 |
| PR & MK | | | 3.7 | 3.6 | 0 | 0 |

*not influential, thus providing additional evidence on the limited effect of autocorrelation on trend detection*". However, those rejections result from MK and PR tests performed under the assumption of temporal independence (case "$\rho = 0$; $\varphi = 0$" in Farris et al. (2021)). In this case, rejections are necessarily independent of $\hat{\rho}_1$ due to the implicit model under which the tests are performed. As recalled in Section 1, results should be interpreted in light of the underlying statistical model and not vice versa.

As mentioned throughout this study, a suitable diagnostic plot might be more informative than just reporting the number/rate of rejections. Figure 10 displays the FDR diagrams of the sorted $p$-values versus their ranks for MK and PR tests. All $p$-values are above the FDR reference line. Independently of the geographic region one focuses on, all tests are not significant at $\alpha_{\mathrm{FDR}} \cong 0.10$, and $\mathbb{H}_0$ cannot be rejected at global level. Moreover, FDR diagrams provide additional information. In fact, when $\mathbb{H}_0$ is consistent with the underlying (implicit or explicit) model and corresponding assumptions, $p$-values are expected to be uniformly distributed, that is, aligned along a straight line connecting the origin $(0,0)$ and the point with abscissa equal to the maximum rank and ordinate equal to the maximum $p$-value, where the latter is equal to 1 (0.5) for one-sided (two-sided) tests (e.g., Falk and Michel, 2006; Serinaldi et al., 2015). Figure 10 shows that the estimated $p$-values are reasonably aligned along such a line, thus confirming the overall (global) consistency of $\mathbb{H}_0$ with a temporally correlated and over-dispersed process for $Z$.

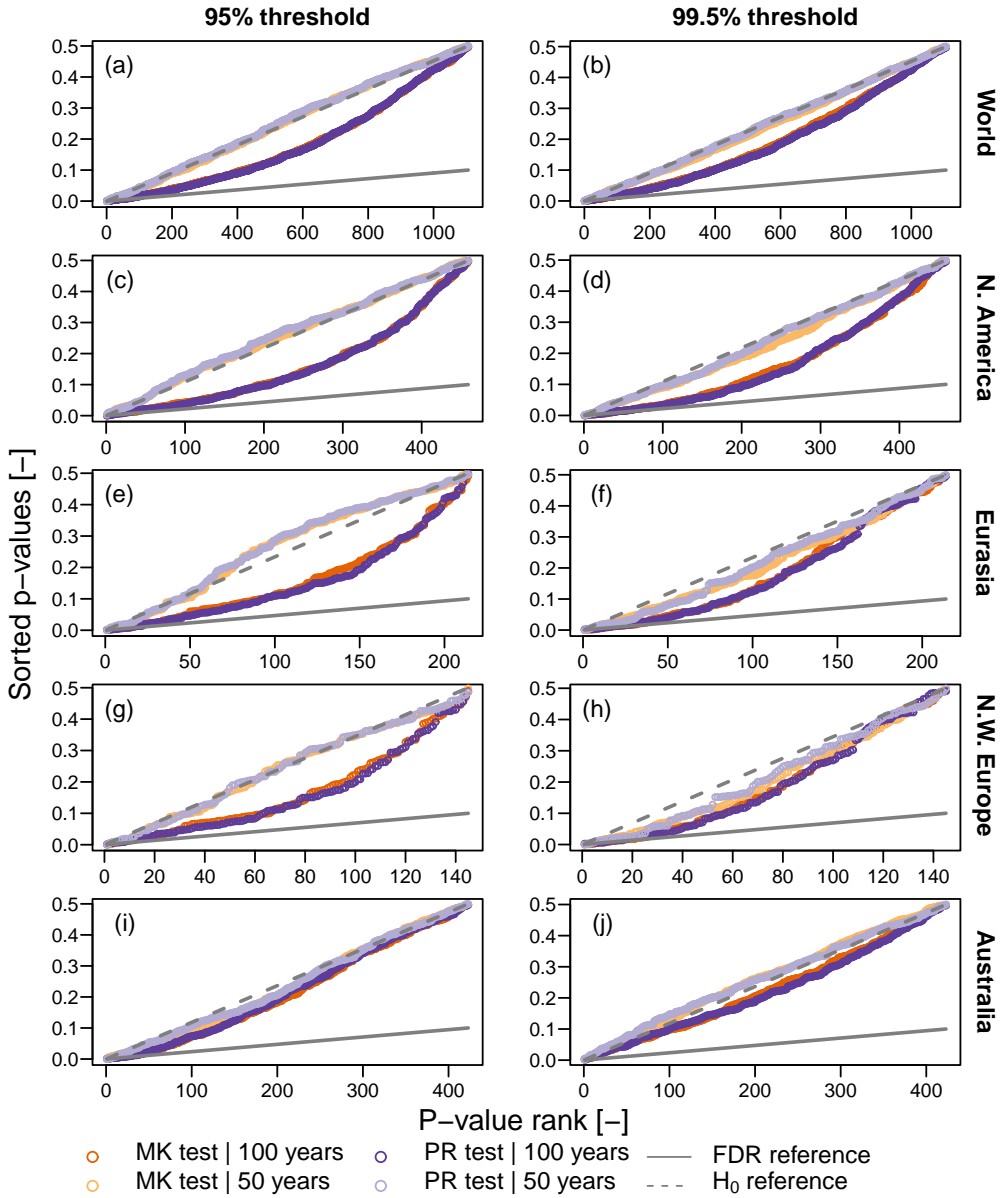

**Figure 10.** Illustration of FDR criterion with $\alpha_{\mathrm{FDR}} \cong 0.10$ (gray diagonal line). Plotted points are the sorted $p$-values resulting from the application of MK and PR tests to data recorded at each location. Results are reported for the five regions described in Section 2. The distributions of test statistics (and therefore critical values) are estimated from 10,000 IAAFT samples. Points below the diagonal lines represent significant results (i.e., rejections of $\mathbb{H}_0$) according to FDR control levels.

### 5.3.2 Space beyond time: The non-negligible effects of spatial dependence

Figure 7 (and Figures S2-S4 in the Supplement) shows that the locally significant trends tend to cluster in geographic areas where such trends exhibit the same sign (e.g., South-Western Australia, North America, and European coastal areas around the North Sea). Often this behavior is interpreted as further evidence of trend existence. However, this interpretation neglects that the spatial clustering is also an inherent expression of spatial dependence in the same way temporal clustering is the natural expression of temporal dependence (see e.g., Serinaldi and Kilsby, 2016b, 2018; Serinaldi and Lombardo, 2017a, b, 2020, for examples of spatio-temporal clustering).

For example, Farris et al. (2021) state that the detection of significant trends with similar sign/magnitude over spatially coherent areas "*is also supported by the physical argument that extreme $P$ is often controlled by synoptic processes (Barlow et al., 2019), and that their occurrence is changing in time (Zhang & Villarini, 2019)*". However, while similar evolution of the occurrence of extreme $P$ and synoptic systems is due to their physical relationships, statistical tests for trends cannot provide information about the nature of the temporal evolution of such processes. Indeed, as shown in Section 5.3.1, the outcome of statistical tests depends on the underlying assumptions. Therefore, the jointly evolving fluctuations of both processes (extreme $P$ and synoptic systems) can be identified as 'not significant' or 'significant' based on the assumptions used to perform the statistical tests. Loosely speaking, if we observe an 'increasing trend' in the occurrence of synoptic systems, a similar behavior likely emerges in local $P$ records observed over the area interested by the synoptic processes, as the latter cause the former. Therefore, what we actually need is to identify the physical mechanism causing trends in the synoptic systems, as trends in extreme $P$ are just a consequence. In this respect, performing massive statistical testing is rather uninformative, as it does not matter if the observed fluctuations are statistically significant or not. Detecting trends in multiple local processes that are known to react to fluctuations of synoptic generating processes does not add evidence, and just reflects information redundancy due to their common causing factor.

To support the foregoing statements with quantitative analysis, we checked the consistency of the spatio-temporal behavior of observed OT occurrences with the assumption of spatio-temporal dependence. Following the model-based approach, we avoid statistical tests for trend detection and rely on theoretical reasoning to formulate a coherent model, thus checking the agreement with observations by simple but effective diagnostic plots.

We firstly check the role of possible spatial dependence of OT occurrences focusing on the distribution of the number $Z_S$ of daily OT occurrences over the five regions introduced in Section 2 (i.e., World, North America, Eurasia, North-Western Europe, and Australia). Daily time scale is selected to isolate the effect of spatial dependence from that of temporal dependence, as it is the finest time scale, and the counting procedure does not involve any aggregation over time. The occurrence of OT events over $m$ locations can be seen as the outcome of $m$ Bernoulli trials. Under dependence, the distribution of $Z_S$ can be described by a $\beta\mathcal{B}$ distribution, where the parameter $\rho_{\beta\mathcal{B}}$ controlling over-dispersion can be expressed as the average of the off-diagonal terms of the lag-0 correlation matrix of the process $Y$ describing the daily occurrence/non occurrence of OT events at each spatial location (see Section S4 in the Supplement). In other words, if the spatial correlation is sufficient to describe the spatial

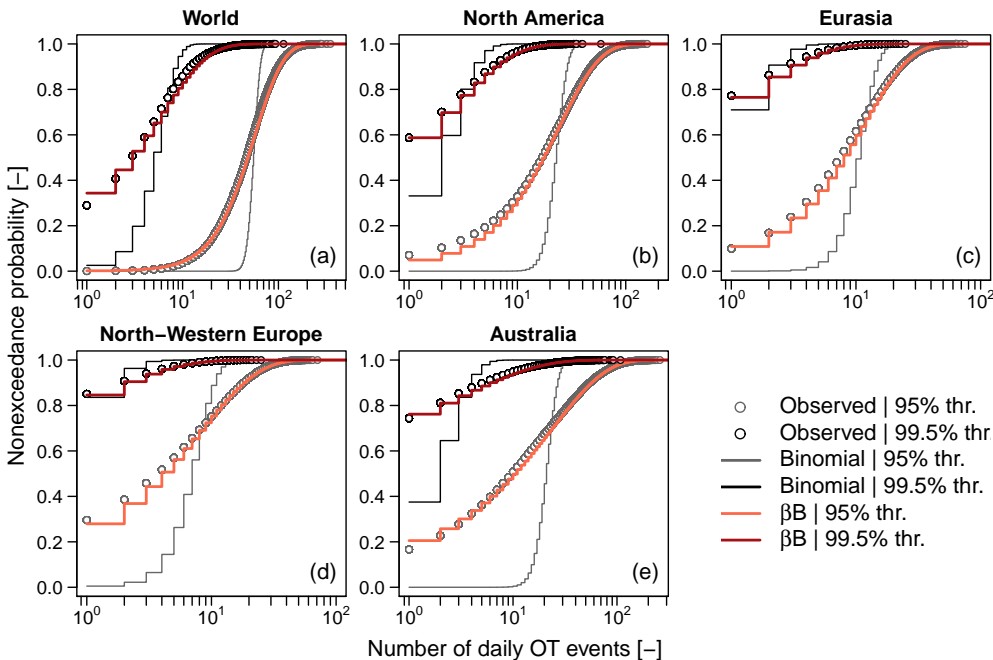

**Figure 11.** ECDFs of number of OT events (for the 95% and 99.5% thresholds) occurring at daily time scale over different regions along with Binomial and $\beta\mathcal{B}$ CDFs.

clustering, we expect that the $\beta\mathcal{B}$ distribution with $\rho_{\beta\mathcal{B}}$ estimated as the average cross-correlation between binary time series of daily OT occurrences faithfully matches the empirical distribution of $Z_S$ over any region.

Figure 11 shows that the $\beta\mathcal{B}$ distribution reproduces accurately the above-mentioned empirical distributions for any threshold and region. Note that North America, Eurasia, Australia, and North-Western Europe are nested regions of World, and North-Western Europe is also a nested region of Eurasia. Therefore, the remarkable fit of $\beta\mathcal{B}$ indicates that the spatial correlation is sufficient to describe the spatial clustering both globally and locally. In other words, the simultaneous occurrence of daily OT events in North America or North-Western Europe, for instance, is consistent with a stationary spatially correlated process. Figure 11 also reports the Binomial distribution that would be valid under independence, thus highlighting the huge (but too often neglected) impact of spatial dependence on the distribution of $Z_S$ (see also Douglas et al., 2000; Serinaldi and Kilsby, 2018; Serinaldi et al., 2018, for additional examples).

Since the aim of tests for trend should be the detection of 'deterministic' temporal patterns, we checked the possible temporal evolution of the distribution of $Z_S$ over the five regions. This information is summarized in Figure 12 in terms of box plots of $Z_S$ aggregated at decadal scale to better visualize temporal patterns along the century. Figure 12 also shows the 95% prediction intervals from Binomial and $\beta\mathcal{B}$ distributions reported in Figure 11. Of course, these prediction intervals are constant as the Binomial and $\beta\mathcal{B}$ distributions are unique under the assumption of stationarity. The empirical distributions of $Z_S$ do not show any evident temporal evolution along the ten decades, and more importantly, any possible fluctuation is well within the range of

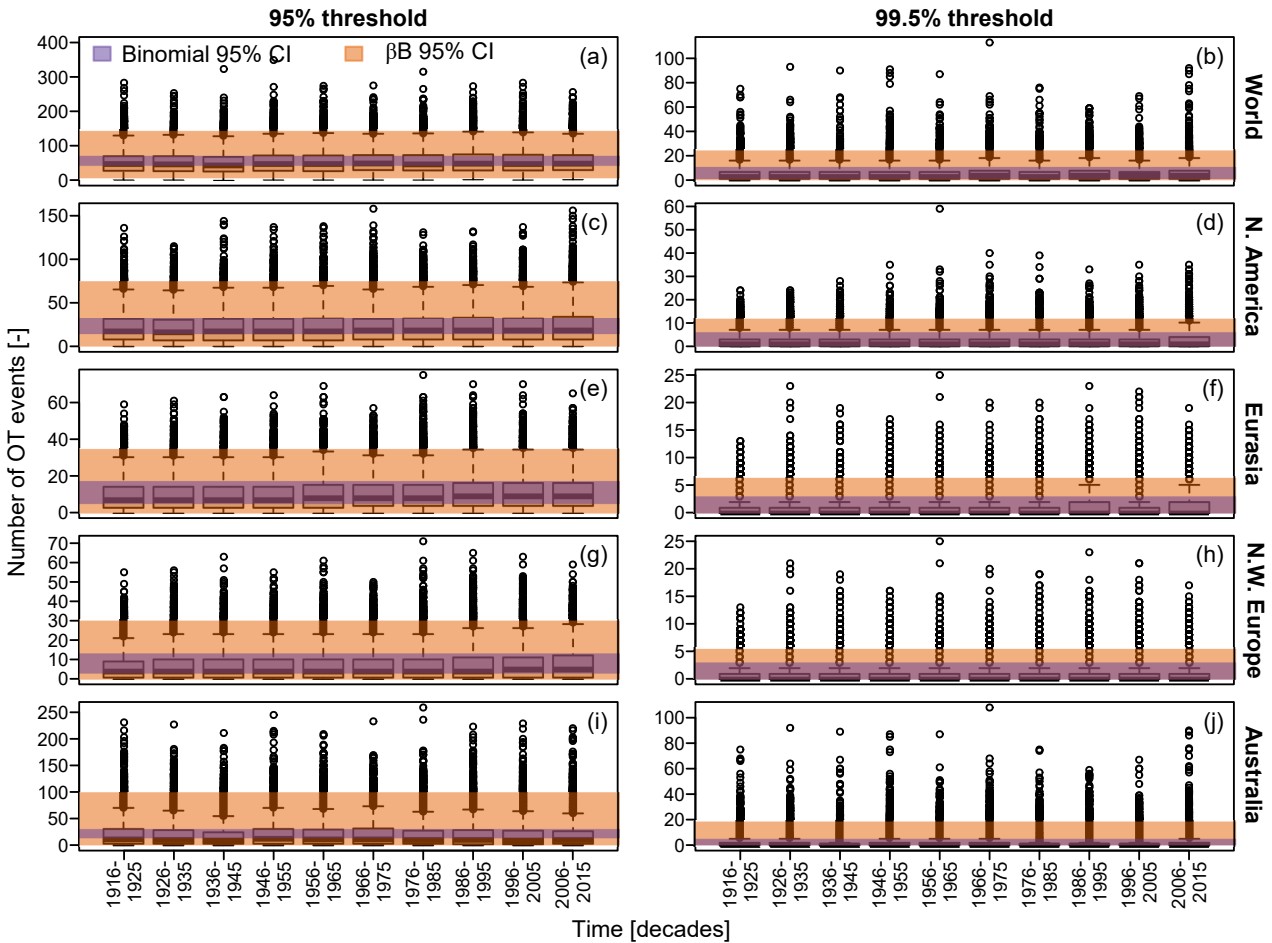

**Figure 12.** Box plots summarizing the decadal distributions of the number of OT events (for the 95% and 99.5% thresholds) occurring at daily time scale over different regions along with the 95% prediction intervals corresponding to Binomial and $\beta\mathcal{B}$ distributions.

values allowed by the $\beta\mathcal{B}$ distribution. The comparison of the 95% prediction intervals from Binomial and $\beta\mathcal{B}$ models further highlights the huge effect of spatial dependence, which can yield prediction intervals from $\cong 2$ up to $\cong 6$ times wider than those corresponding to spatial independence.

### 5.3.3  Space and time: The non-negligible effects of spatio-temporal dependence

While the analysis at daily scale allowed to focus on spatial dependence, here we focus on the annual number $Z_{\mathrm{ST}}$ of OT events over multiple locations. Studying the spatial clustering of such data implies aggregation in space and time. In other words, the occurrence of OT events can be thought of as a set of Bernoulli trials over $m$ locations (i.e., the number of stations in each geographic region) and $n$ time steps (i.e., the 365 days in one-year time interval), and we are interested in the distribution of

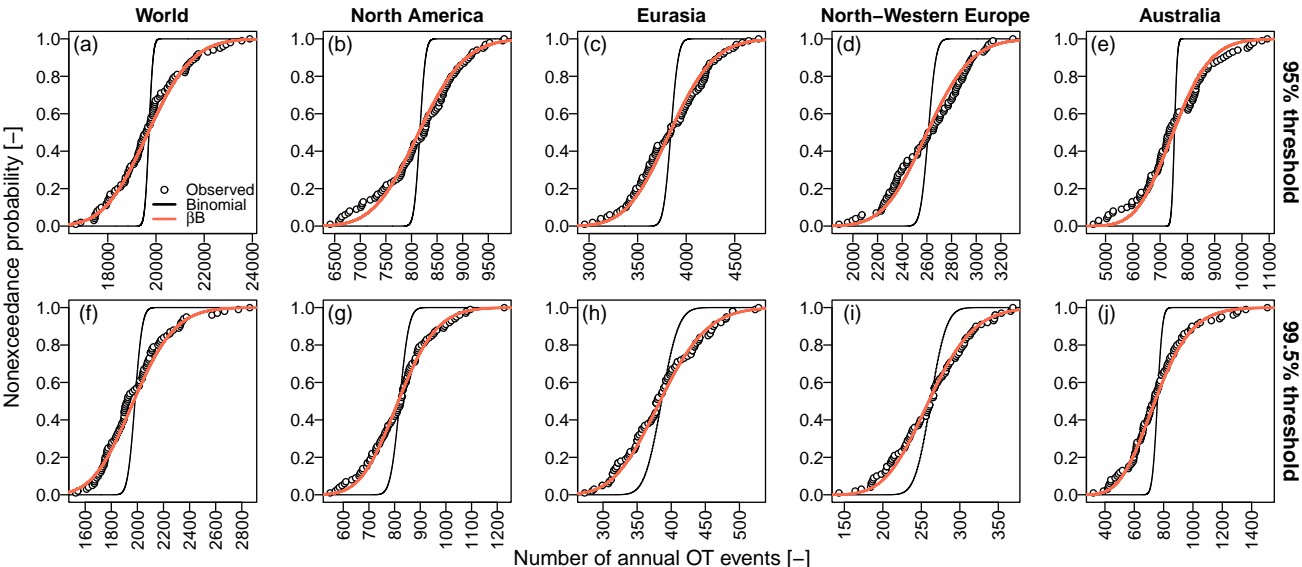

**Figure 13.** ECDFs of number of OT events (for the 95% and 99.5% thresholds) occurring at annual time scale over different regions along with Binomial and $\beta\mathcal{B}$ distributions.

$Z_{\mathrm{ST}}$ resulting from $m \cdot n$ Bernoulli trials. This case is analogous to that concerning daily OT occurrences. The $\beta\mathcal{B}$ distribution
is still a theoretically consistent model for $Z_{\mathrm{ST}}$, and its over-dispersion parameter $\rho_{\beta\mathcal{B}}$ can be expressed as the average of
all lagged auto- and cross-correlation values of the generating binary process $Y$ up to time lag $n = 365$ (see Section S4 in the
Supplement). Also in this case, we use probability plots and box plots to assess the validity of the $\beta\mathcal{B}$ distribution, and therefore
its underlying assumption of spatio-temporal dependence.

Figure 13 shows that the $\beta\mathcal{B}$ model faithfully describes the ECDFs of $Z_{\mathrm{ST}}$ for any threshold and region. This means that the
local spatio-temporal correlation is sufficient to describe the differences in all regions and sub-regions without introducing any
ad hoc local models, involving for instance physically unjustified linear trends or generic links with local exogenous factors.
We stress again that the parameter $\rho_{\beta\mathcal{B}}$ is not estimated on the 100 values of $Z_{\mathrm{ST}}$ in each region, but it comes from the spatio-
temporal correlation values of the generating binary process $Y$. Therefore, the goodness of fit of the $\beta\mathcal{B}$ distribution is not
related to the minimization of some distance metric for 100-size samples, but depends on the agreement of the observed binary
time series with the hypothesized stationary spatio-temporal stochastic process $Y$.

For any threshold and region, Figure 14 confirms that the temporal fluctuations of the distribution of $Z_{\mathrm{ST}}$ is well within
the range of values expected from a stationary stochastic process characterized by the observed spatio-temporal correlation
structure. In this case, the 95% prediction intervals from $\beta\mathcal{B}$ models under spatio-temporal dependence are from $\cong 3$ up to $\cong 13$
times wider than those yielded by Binomial distribution under spatio-temporal independence. Of course, an increasing pattern
along the decades is evident in the regions of the Northern hemisphere. However, accounting for spatio-temporal correlation
dramatically changes their interpretation. Such fluctuations are obviously inconsistent with the assumption of independence

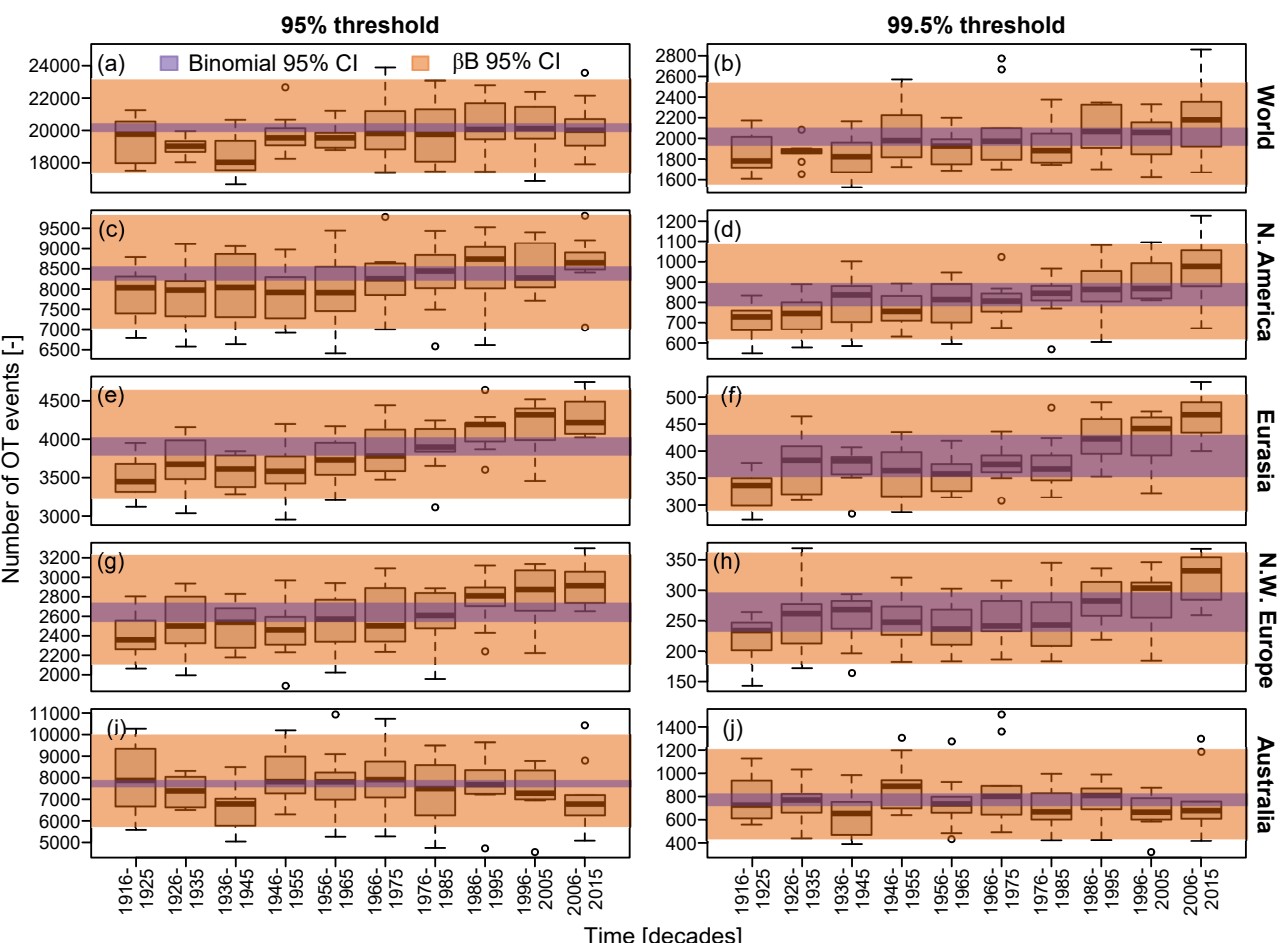

**Figure 14.** Box plots summarizing the decadal distributions of the number of OT events (for the 95% and 99.5% thresholds) occurring at annual time scale over different regions along with the 95% prediction intervals corresponding to Binomial and $\beta\mathcal{B}$ distributions.

and therefore the Binomial model. This explains the high rejection rate of the trend tests performed under independence. On the other hand, low-frequency fluctuations evolving over wide spatial scales, and time scales comparable or longer than the observation period are the expected behavior of spatio-temporal dependent processes. Therefore, we should ask ourselves whether such fluctuations can look unexpected or surprising because they are actually unusual or just because humans tend to systematically underestimate the actual uncertainty characterizing the surrounding environments, thus looking at hydro-climatic processes with a too anthropocentric point of view, which is inherently uncertainty-averse or behaviorally biased toward known outcomes.

# 6 General discussion and concluding remarks

## 6.1 Statistical tests for trend detection: unfit for purpose!

Disagreement between model-based and test-based methods are mainly related to the inherent problems affecting statistical hypothesis tests. They are statistical methods developed for evaluation of differences in repeatable experiments that "*have been misused to create an illusion of a scientific analysis of unrepeatable hydrologic events*" (Klemeš, 1986). Logical, conceptual, and practical inconsistencies of statistical tests have been widely discussed in both theoretical and applied literature (Pollard and Richardson, 1987; Gigerenzer et al., 1989; Flueck and Brown, 1993; McBride et al., 1993; Meehl, 1997; Gill, 1999; Johnson, 1999; Nicholls, 2001; Krämer and Gigerenzer, 2005; Levine et al., 2008; Ambaum, 2010; Clarke, 2010; Beninger et al., 2012; Ellison et al., 2014; Nuzzo, 2014; Briggs, 2016; Greenland et al., 2016; Wasserstein and Lazar, 2016; Serinaldi et al., 2018, 2020a; Wasserstein et al., 2019, and references therein).

One of the key drawbacks of statistical tests is the error of 'transposed conditional' (also known as 'converse inequality argument' or 'inverse probability problem' (Pollard and Richardson, 1987; Gill, 1999; Krämer and Gigerenzer, 2005; Ambaum, 2010; Beninger et al., 2012; Serinaldi et al., 2018, 2022)). It consists of confusing conditional and conditioning events, so that we are interested in the probability of the null hypothesis $\mathbb{H}_0$ given the observational evidence (data), and we end up calculating the probability observational data when $\mathbb{H}_0$ is assumed to be true. This is like confusing the probability $\mathbb{P}$[a man is a UK citizen | a man is the King of the UK] $\cong 1$ with the probability $\mathbb{P}$[a man is the King of the UK | a man is a UK citizen] $\cong 1/(33.7 \cdot 10^6)$.

In the context of statistical tests for trend detection applied to hydro-climatic data, rejection of $\mathbb{H}_0$ (e.g., 'no trend') does not provide information about the likelihood of $\mathbb{H}_0$ given the observations. Rejection does not allow any statement about possible deterministic trends because deterministic trends are not a property of the model $\mathbb{H}_0$ assumed to perform the test. In other words, 'rejection' can be due to something that is unknown and different from deterministic trends. Similarly, 'no rejection' might be related to violation of implicit assumptions of the model $\mathbb{H}_0$, thus introducing spurious effects due to exogenous factors (Serinaldi et al., 2022). One of these factors is the spatio-temporal dependence. Its effects on statistical inference have been widely studied in the literature (Jones, 1975; Katz, 1988a, b; Katz and Brown, 1991; Kulkarni and von Storch, 1995; Hamed and Rao, 1998; Douglas et al., 2000; Yue and Wang, 2002; Koutsoyiannis, 2003; Yue and Wang, 2004; Hamed, 2008, 2009a, b, 2011; Bayazit and Önöz, 2007; Serinaldi and Kilsby, 2016a, 2018; Serinaldi et al., 2018, 2020a). In particular von Storch and Zwiers (2003, p. 97) recalled that "*the use of statistical tests in a cookbook manner is particularly dangerous. Tests can become very unreliable when the statistical model implicit in the test procedure does not properly account for properties such as spatial or temporal correlation*". Nonetheless, the foregoing iterated warnings and recommendations are systematically ignored.

## 6.2 Models, tests, and their interpretation

The aim of most of the literature applying statistical tests for trend detection on hydro-climatic processes is to find the answer to a question that can generally be summarized as 'are these processes stationary or nonstationary?'. However, such a question

is scientifically ill-posed as natural processes cannot be either stationary or nonstationary. Only mathematical models used to describe physical processes can.

It can be argued that 'statistical trend testing attempts to assess whether the natural process has manifested in a stationary or nonstationary fashion during the period of observation to ultimately support decision-making in the future'. However, this type of statements confuses sampling fluctuations, which can look monotonic or not, with a population property such as stationarity. Statistical trend tests attempt to infer the latter, as theoretical properties/assumptions are the only one behind any statistical method. As every statistical method, statistical tests for trend detection make inference about '*population* stationarity' (Khintchine, 1934; Kolmogorov, 1938), not about sampling fluctuations, which can result from a variety of stationary and nonstationary processes. These tests attempt to establish what kind of population behavior is compatible with observed sampling fluctuations. Otherwise, we would not need any test to state that an observed sample shows a given (monotonic or non-monotonic) temporal pattern, as we would just need to look at the diagrams of time series. We infer the population properties because these allow us to assume a model and make out-of-sample predictions. We argue that the vague use of the term 'stationarity', overlooking a formal definition and its consequences, is one the main reasons of a widespread misuse and misinterpretation of the output of statistical tests (see Koutsoyiannis and Montanari, 2015; Serinaldi et al., 2018, and Section S3 in the Supplement for further discussion).

The comparison of test-based and model-based approaches discussed in this study attempts to clarify the foregoing concepts. For the same physical process (i.e., the OT occurrences of $P$) we showed two options. On the one side, we can choose to model the number $Z$ of OT occurrences by nonstationary Poisson distributions (NHP). In this way, (i) we neglect that the Poisson distributions are theoretically unsuitable to describe $Z$, and therefore do not reproduce the observed marginal distribution of the $Z$ process, and (ii) we assume that the rate of occurrence in NHP models change in time according to linear (or nonlinear) laws that have no physical justification. The aim of this type of regression models is exactly to follow sampling fluctuations, and they hardly ever provide information about the underlying population properties. This also explains why extrapolation for this type of models is always deprecated in textbooks and introductory courses in statistics, and when it is done, it might yield paradoxical results (Serinaldi and Kilsby, 2015; Luke et al., 2017; Iliopoulou and Koutsoyiannis, 2020; Anzolin et al., 2024). On the other side, we can attempt to preliminarily understand the general theoretical properties of spatio-temporal OT processes, look for appropriate models reproducing such expected *population* properties, and check if these models are general enough to reproduce the observations at various spatio-temporal scales. Using this approach, we ended up with the conclusion that the spatio-temporal correlation structure of a stationary stochastic process provides a good description of the behavior of the observed OT frequencies at various spatio-temporal scales. Thus, the actual question is not about the (non)stationarity of natural processes or '(non)stationary behavior of observed samples', but which kind of model we deem more suitable in terms of generality, reliability and parsimony.

Conversely to what is often iterated in the literature, accurate statistical trend analysis of observed and modeled $P$ time series are not key to validate hypotheses on the underlying physical mechanisms and do not improve our ability to predict the magnitude of these changes. On the contrary, the foregoing discussion shows that the statistical tests for trend detection might generate confusion, potentially concealing model mis-specification (see also Serinaldi and Kilsby, 2016a; Serinaldi et al.,

2018, 2020a, 2022, for further examples). Statistical tests can just reflect the properties of the underlying models at most, while well devised models do not need any statistical test to be validated. In fact, we did not use any statistical test to show the validity of the model-based approach. We only needed to visually compare observed properties with those expected from theory for a range of spatio-temporal scales. When we applied statistical tests for the sake of comparison with existing literature, tests' outcomes just reflected what was already known and expected from a theoretical point of view.

## 6.3 Confirmatory versus dis-confirmatory empiricism

Why do our results contrast with those reported in most of the existing literature on trend detection? Because most of this literature resorts to methods based on the same unrealistic assumption of independence and corresponding trivial models such as those discussed in this study. On the other hand, when dependence is accounted for, its true consequences on the entire inference procedure are commonly underestimated, partially missed, or neglected (e.g., Lombardo et al., 2014; Koutsoyiannis, 2020; Dimitriadis et al., 2021, and references therein). Therefore, the resulting (expected) high rejection rates are incorrectly interpreted as evidence of a specific alternative, whereas rejection can be due to a variety of causes that are not considered in the analysis. The missing key point is that the results of statistical analysis and their interpretation depend on the underlying assumptions and models according to the rationale of statistical inference (Aitken, 1947; Cramér, 1946; Papoulis, 1991; von Storch and Zwiers, 2003). Conversely to what is incorrectly stated in the literature too often, even the simplest diagnostic diagram relies on an underlying model.

Why does most of the literature on trend detection rely on the same methods? There are several causes. We argue that the main one is a too superficial approach probability and statistics along with insufficient exposure to the epistemological principles of science. Using similar methods based on the same assumptions always gives similar results. However, "*a million observations of white swans do not confirm the non-existence of black swans*" and "*a million confirmatory observations count less than a single disconfirmatory one... What is called 'evidence based' science, unless rigorously disconfirmatory, is usually interpolative, evidence-free, and unscientific*" (Taleb, 2020). Since most of the literature on trend detection is just an iterative application of test-based approach and eventually statistical tests under the assumption of independence or ill-defined dependence, one should wonder if the general agreement is related to the common misinterpretation of the output of the same inappropriate methodologies rather than to physical properties. In this study, we offered an alternative point view, which is nothing but the specialization for data analysis of the scientific method used for centuries and seemingly forgotten in the recent decades in some research areas. Obviously, according to the scientific method, also the content of this study should be taken critically, and interested readers should independently assess which approach (test-based, model-based or something else) looks more general, reliable and parsimonious, and eventually consistent with the epistemological principles of scientific inquiry.

*Data availability.* Data are freely available from the Global Historical Climatology Network repository (Menne et al., 2012a, https://www.ncei.noaa.gov/ac page/bin/iso?id=gov.noaa.ncdc:C00861).

*Author contributions.* FS: Conceptualization, Methodology, Software, Formal analysis, Writing - original draft, Writing - review & editing, Visualization.

*Competing interests.* The author declares that he has no known competing financial interests or personal relationships that could have appeared to influence the work reported in this paper.

*Acknowledgements.* The author acknowledges the support from the Willis Research Network. The author thanks the handling editor, Dr. Nadav Peleg (University of Lausanne), and the eponymous reviewers, Dr. Panayiotis Dimitriadis (National Technical University of Athens), Dr. Demetris Koutsoyiannis (National Technical University of Athens), and Dr. Giuseppe Mascaro (Arizona State University) for their critical

remarks that helped substantially improve content and presentation of the original manuscript. The author also thanks Dr. Hans von Storch for his feedback on a previous version of this manuscript. The analyses were performed in R (R Development Core Team, 2023).

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
