# Peer review of "Scientific logic and spatio-temporal dependence in analyzing extreme precipitation frequency: Negligible or neglected?"

_Hydrology and Earth System Sciences, 2023_

## Referee Comment (RC1)

**Review of: "Impact of spatio-temporal dependence on the frequency of precipitation extremes: Negligible or neglected?" by Dr. Francesco Serinaldi**

Reviewer: Giuseppe Mascaro

As well summarized in the title, the main goal of this paper is to demonstrate the importance of accounting for spatio-temporal dependence on the frequency of precipitation extremes when investigating the possible presence of non-stationarity. The paper is motivated, in part, by a recent study by Farris et al. (2021) who performed analyses of long-term daily precipitation records covering several regions of the world to investigate the importance of serial correlation and field significance in trend analysis. In the manuscript under review here, most of the analyses of Farris et al. (2021) are repeated using alternative models and approaches that the author believes are more correct for the purpose. The main critiques raised by the author against the work of Farris et al. (2021) are (1) the adoption of the INAR(1)-Poisson and Non-Homogeneous Poisson (NHP) models to simulate time series that reproduce realistic stationary autocorrelated series and uncorrelated series with trends, respectively; and (2) the use of statistical tests to detect trends (and in general the 'standard' approaches used in the literature on trend analyses). To pursue his main goal, the author presents figures and analyses aimed at showing (and erroneous proving) that:

- The INAR(1) and NHP used in Farris et al. (2021) are not proper models of count time series of over-threshold (OT) daily precipitation (P) series since they do not capture the marginal distribution of the dataset.
- The hypothesis of no-trend is verified in all cases using the Iterative Amplitude Adjusted Fourier Transform (IAAFT) model after the power spectrum is bias corrected to account for the sample size and the field significance is considered.
- The Beta-Binomial (BB) distribution parameterized through the empirical spatial, temporal, and spatiotemporal linear correlation structure of the binary process of daily OT P occurrences captures the distribution of annual OT counts. Since the binary process used to parameterize the BB is assumed stationary by the author, it is concluded that there is no need to apply any trend test.

I carefully read the paper by Dr. Serinaldi to properly understand the critiques of some of the results of Farris et al. (2021), of which I am a co-author, and whether the conclusions reached in this paper are supported by the data. As I demonstrate below, while some of the conclusions are reasonable, there are a number of assumptions made by the author that are (1) not empirically supported, and (2) similar in nature to the subjectivity that the author criticizes in the 'standard' approaches used in the literature on trend analyses. Based on these issues which are better explained in the points below, I recommend the paper rejection.

I first introduce some symbols following for the most part those used by the author:
- $X$ is the random variable of daily precipitation depth, P. $\{X_j\}$ is the random process of $X$ for discrete times $j$ = 1, 2, 3, … (here, days).

- $Y$ is the binary random variable based on the condition $X > x^*$, with $x^*$ being a given threshold. $\{Y_j\}$ is the binary random process of $Y$ for discrete times $j$ = 1, 2, 3, … (here, days).
- $Z$ is the nonnegative integer counts of P observations exceeding $x^*$ in a specified time window; here, within each year. $\{Z_i\}$ is the random process of $Z$ for discrete times $i$ = 1, 2, 3, … (here, years). $\{Z_i\}$ is related to $\{Y_j\}$ as $Z_i = \sum_{j=1}^{n} Y_j$, with $n$ = 365 (or 366). In words, $Z_i$ is the sum of all the ones in $\{Y_j\}$ that occur in year $i$. These are the OT samples mentioned above.

Now, I explain my concerns.

**1. Value of the BB distribution for the time series of annual counts $\{Z_i\}$**

While I understand the reasonings of adopting the BB as a reasonable distribution to represent the correct marginal distribution of stationary time series of counts that exhibit serial correlation, I disagree with the author attempts to show that this is also true from the practical point of view by comparing it with the Poisson distribution in Figures 2 and 3. I have separate concerns related to these figures.

**1.1. Scatterplot between mean and variance of Z**

In Figure 2, the author presents scatterplots between the mean and variance of $Z$ for the observed OT samples along with those of synthetic samples obtained from Poisson, NHP, and BB models. I tried to reproduce Figure 2 for OT above the 95% empirical quantiles and reported results in Figure R1. I first point out that the mean of the observed samples should not vary, because it is prescribed by the quantile adopted for $x^*$. For the 95% empirical quantiles, it should be $x^*$ = (1 − 0.95)*365.25 = 18.2625. This is not the case in Figure 2 of the paper, where the mean exhibits a negatively skewed spread around the expected value of 18.2625. Based on my interpretation, which I used to generate Figure R1, this is an artifact caused by the fact that the author did not account for the presence of repeated values in the $\{X_j\}$ series when applying the condition ($X_j > x^*$) to ultimately compute $\{Z_i\}$. Note that a positively skewed spread for the mean (essentially, a mirrored version of the spread in Figure 2) is instead obtained by applying the condition ($X_j \geq x^*$).

[Figure]

Figure R1. Relationship between mean and variance of observed OT samples and those corresponding to simulated samples from (a) Poisson, (b) NHP, and (c) BB distributions.

That said, I produced Figure R1 by (i) estimating the parameter/s of each model separately on each observed sample, and (ii) generating a sample with the same size as the observation with the estimated parameter/s. This approach mimics what the author indicated in the caption: "*Observed values are compared with those corresponding to simulated samples from Poisson, NHP, and Beta-Binomial (BB) distributions*". If one follows this approach, one random generation of the synthetic samples of the three models should result in a sampling variability for the mean larger than that of observations and symmetric around the expected value defined by the 95% threshold (by the way: note that, in Figure R1, the variability of mean and variance is the same for the Poisson variates, as expected). In Figure 2 of the paper, the means of the randomly generated samples have instead exactly the same range as the observations. This is incorrect.

**1.2.    CDFs of observed samples and fitted distributions**
Apart from the issue indicated above, Figure 2 (and Figure R1 above) shows that, for several cases, the variance of the observed samples is larger than the mean. The one-parameter Poisson distribution is unable to fully capture this spread, while, as well known, a two-parameter distribution like the BB can be fitted to reproduce both the mean and the variance. To further prove the point that the BB is a better distribution, the author presents in Figure 3 cumulative distribution functions (CDFs) and differences in probability of observed and fitted distribution (I believe). However, the author did not explain how that figure was created. What are the lower and upper limits and how were they calculated?

More importantly: can we clearly say that the two-parameter BB distribution provides a large improvement compared to the one-parameter Poisson distribution when looking at panels (e) and (f) of Figure 3 in the paper? Since I could not understand it well, I compared, for some representative gages (selected based on an equiprobability criteria to fairly explore all possible behaviors), the empirical CDF of $Z_i$ and the CDFs of the fitted Poisson and BB distributions (see comment 3.1 regarding how parameters of the BB were estimated). I chose the gages based on the variance of $Z_i$, whose empirical CDF across all gages is shown in Figure R2. I picked the gages with variance associated with a cumulative frequency, $F$, close to 0.1, 0.2, 0.3, …, 0.9. Results are shown in Figure R3: these graphical diagnostics (which the author recommends using in general) indicate that the two distributions are not markedly different, even for the largest values of the variance. This is also quantified by the values of the Cramer-von Mises goodness-of-fit metric (without any penalty) provided in the legend, which are very similar despite the BB distribution having an additional parameter compared to the Poisson, ranging from 8 to 22 for BB and from 9 to 24 for Poisson fitting. In addition, the variability of such metric does not seem related to the variance of $Z_i$, suggesting that the gain of applying the BB against the Poisson when the variance of $Z_i$ becomes increasingly larger than the mean is negligible. Therefore, for most cases, a parsimonious one-parameter distribution like the Poisson does not seem to do a bad job when characterizing the frequency of the empirical counts as compared to the BB (as proposed by the author), which depends on two parameters, and thus should be properly penalized in any comparison.

[Figure]

*Figure R2. Empirical CDF of the variance of the OT observed samples, Z. The vertical red line is the average of the sample means.*

**RSM00029866**
Var = 20.0 - F(Var of Z) = 0.1

**ASN00010592**
Var = 22.7 - F(Var of Z) = 0.2

**ASN00070027**
Var = 24.2 - F(Var of Z) = 0.3

**USC00246862**
Var = 25.7 - F(Var of Z) = 0.4

**ASN00010614**
Var = 27.7 - F(Var of Z) = 0.5

**ASN00088023**
Var = 30.0 - F(Var of Z) = 0.6

**NLE00100503**
Var = 32.7 - F(Var of Z) = 0.7

**ASN00079014**
Var = 36.8 - F(Var of Z) = 0.8

**ASN00024573**
Var = 42.5 - F(Var of Z) = 0.9

*Figure R3. CDF of the observed Z samples, along with fitted Poisson and BB distributions. In each panel, the gage ID is shown along with the values of the variance of $Z_i$ and the cumulative frequency of the variance from Figure R2. The values reported in the legends next to "Poisson" and "BB" are the values of the Cramer-von Mises goodness-of-fit statistics.*

**2. Use of the IAAFT model**

The author uses the Iterative Amplitude Adjusted Fourier Transform (IAAFT) model to make the case that there are alternative stationary models that reproduce the same linear trend slope and serial correlation of the observed samples. As mentioned by the author, "*IAAFT allows the simulation of synthetic time series that preserve the empirical marginal distribution and, to some error level, the empirical power spectrum of the original data*". The empirical marginal distribution is preserved because the data are reshuffled. As such, it seems to work like a bootstrapping of the data that preserves some lags of the serial correlation (see Section 5.3.5 in Wilks, 2011).

I have three concerns regarding this model. To show this, I picked the $\{Z_i\}$ for a few gages with different strengths of the serial correlation structure and used the IAAFT model to generate synthetic samples.

**2.1. Why is bias correcting the power spectrum needed?**

The author mentions the use of bias correction of the power spectrum, assuming the presence of fractional Gaussian noise, as described in the Supporting Information. However, the author did not properly explain nor demonstrate why this is needed for the series at hand and the level of subjectivity associated with the choice of the bias correction method. In Figures R4-R9, panels (a) and (b) show examples of time series for a few randomly picked synthetic samples along with the upper and lower limits and the median serial correlations of 10,000 synthetic samples generated without any bias correction. If the lag-1 serial correlation $\rho_1$ is used to measure the strength of the serial correlation structure, it is important to mention that $\rho_1 < 0.2$ (0.3) for the $Z_i$ of ~90% (97%) of the gages. Gages with values of $\rho_1$ up to 0.3 are reported in Figures R4-R7. For these cases, the autocorrelation function of the synthetic samples does not seem to be affected by any bias. Some bias starts appearing for stronger serial correlation structures, as shown in Figures R8 and R9. However, the time series of Figures R8 and R9 clearly show an increasing trend that induces those strong correlations… While one of the main concerns raised by the author is the subjectivity that is often adopted to choose trend forms, the reasons why the bias correction was applied is not motivated in the paper, nor it is shown the effectiveness of the bias correction across several strengths of the autocorrelation. I also wonder how, in Figure 9, results look like for the case of IAAFT without bias correction plus false discovery rate (FDR) test.

**2.2. Can the IAAFT model be used to generate the null distribution of a trend test?**

Another concern that I have is whether the IAAFT model is appropriate for generating the null distribution of a trend test. I estimated the slopes of the linear regression $\phi$ (which can be considered a proxy of a trend test metric, being a first order expansion of any trend behavior) for 10,000 synthetic samples generated by the (stationary, but correlated) IAAFT, as suggested by the author, and plotted the empirical density function. Then, I did the same with the (stationary uncorrelated) Poisson distribution, as a reference. Panels (c) of Figures R6-R9 show that the null distribution of $\phi$ is bimodal for the IAAFT. Under the hypothesis of stationarity, we should expect a symmetric distribution with the mode at $\phi = 0$, like in the trivial case of the

Poisson distribution. Thus, distributions like those shown in Figures R8 and R9 raise serious concerns on the power of any trend test based on the assumption of IAAFT distribution. I want also to stress that this aspect is completely neglected by the author, despite the large effort dedicated to criticizing some assumptions and conclusions of the paper of Farris et al. (2021). Consequently, why not dedicating a very small additional effort to evaluate the power of tests using the distribution/model that the author proposes as an alternative to test the null hypothesis of no trend?

[Figure]

*Figure R4. (a) Time series of an observed {$Z_i$} with $\rho_1$ close to 0 along with three synthetic time series generated with the IAAFT model. (b) Autocorrelation function of the observed {$Z_i$} along upper and lower limits (solid blue lines) and the median (dashed blue line) computed from 10,000 samples generated with the IAAFT model. (c) Empirical pdf of the slopes of the linear trend, $\phi$, of the 10,000 time series generated with the IAAFT model and the Poisson distribution fitted to the observed {$Z_i$}. The value of the linear slope of the observed time series is also reported in the title.*

[Figure]

*Figure R5. Same as Figure R4 but for another gage where $\rho_1$ of {$Z_i$} with close to 0.2.*

[Figure]

*Figure R6. Same as Figure R4 but for another gage where $\rho_1$ of $\{Z_i\}$ with close to 0.2.*

[Figure]

*Figure R7. Same as Figure R4 but for another gage where $\rho_1$ of $\{Z_i\}$ with close to 0.3.*

[Figure]

*Figure R8. Same as Figure R4 but for another gage where $\rho_1$ of $\{Z_i\}$ with close to 0.4.*

[Figure]

*Figure R9. Same as Figure R4 but for another gage where $\rho_1$ of $\{Z_i\}$ with close to 0.5.*

**2.3. Does the IAAFT model capture the variability of the observed linear slopes?**

Finally, since the author presents the IAAFT model as a proper stationary model that can explain the presence of trend in virtue of correlation, then one would expect that it should be able to capture the empirical distribution of the observed slopes of $\{Z_i\}$. I applied the IAAFT model for each gage (without bias correction), generated one synthetic sample, and estimated the slope. I then plotted the empirical PDF of slopes of the observed and synthetic samples. I repeated this nine times to explore the sampling variability. As shown in Figure R10, the IAAFT model is not able to reproduce the observed sampling variability and, in particular, the larger number of positive slopes.

[Figure]

*Figure R10. Empirical PDFs of the slopes of linear trend of $\{Z_i\}$, $\phi$, for the observed records and the corresponding synthetic sample (one per gage) generated by the IAAFT model. The nine panel refers to a different set of generations with the IAAFT model.*

**3. Assumption of stationarity for the BB model applied to count time series**

One of the points that the author wants to make is that the BB distribution parameterized based on the mean serial correlation of the binary process *Y* assumed to be stationary well captures the distribution of *Z*. I have several concerns related to this point.

**3.1. Lack of details regarding the estimation of the BB parameters**

The explanation given in the supporting information of how the intra-cluster correlation parameter of the BB is estimated from a time series $\{Y_j\}$ is confusing. The meaning of "cluster" is not explicitly provided, while it should be. Based on Ahn and Chen (1995), a cluster should correspond to $\{Y_j\}$ in a given year (i.e., $j$ = 1, 2, …, 365): is this the case? If so, each cluster has size $n$ = 365 (apart from leap years). The author talks instead about "experiments" on specific days (of the year) $j$ and $l$, but they do not introduce a symbol for the number of available clusters. This should be the size of the vector used to compute the correlation coefficients $\rho_{jl}$. Put simply, if we have $m$ = 100 clusters (or years of records), $\rho_{jl}$ is the correlation coefficient between the vectors of the $m$ Y's at day $j$ for all years and the $m$ Y's at day $l$ for all years. To the my best knowledge, this is the proper approach, and since the authors do not provide any detail, I followed it for the calculations made in this review.

**3.2. What serial correlations and slopes of $\{Z_i\}$ are generated by stationary and nonstationary $\{X_j\}$?**

As mentioned, the estimation of the BB intra-cluster correlation parameter is based on $\{Y_j\}$, assumed to be stationary. However, the stationarity of $\{Y_j\}$ derives from the stationarity of $\{X_j\}$. This is a very important link that the author has not mentioned nor explored. To do this, I performed Monte Carlo simulations with the Complete Stochastic Modelling Solution (CoSMoS; Papalexiou, 2018) to generate stationary and nonstationary time series of $\{X_j\}$. CoSMoS has been shown able to provide realistic precipitation time series that capture serial correlation and marginal distribution, and Dr. Serinaldi has co-authored a few papers where this model has been further developed and tested (Papalexiou et al., 2021; Papalexiou & Serinaldi, 2020). For simplicity, I used the parameters of CoSMoS presented by Papalexiou (2018) for the daily rain gage of the National Observatory of Athens, Greece (see section 4.3 and Figure 6 of that paper), including those controlling the serial correlation and the marginal distribution of the P depths, *X*. For the latter one, Papalexiou (2018) uses the Generalized Gamma (GG) distribution, which depends on two shape parameters and one scale parameter, $\beta$. I performed the following experiments, where, in each case, I generated 1000 time series $\{X_j\}$ of 100 years each (i.e., size of 100 x 365 which is basically the same as the observed series in the paper):

1) **StUncor**: stationary and uncorrelated time series obtained as variates from the GG distribution with observed parameters for Athens.
2) **NonStUncor**: non-stationary and uncorrelated time series obtained as variates from the GG distribution with $\beta$ that increases linearly each year by an arbitrary quantity starting from the $\beta$ estimated in Athens as the initial value, while the shape parameters of the GG are constant and equal to those estimated for Athens.
3) **StCor**: stationary and correlated time series using CoSMoS with the parameters of serial correlation structure and marginal distribution estimated for Athens.

4) **NonStCorr**: nonstationary and correlated time series using CoSMoS where: (1) the scale parameter $\beta$ increases linearly each year by an arbitrary quantity starting from the $\beta$ estimated in Athens as the initial value, (2) the shape parameters of the GG are constant and equal to those estimated for Athens, and (3) the serial correlation structure of $X$ is based on the observed parameters for Athens. Note that changes in $\beta$ do not affect the other CoSMoS parameters, including those controlling the inflation of the serial correlation structure involved in the application of CoSMoS (details are given by Papalexiou, 2018).

For each time series, I computed $\{Y_j\}$ based on $x^* = 95\%$ empirical quantile, and, from these, $\{Z_i\}$. I also calculated the lag-1 serial correlation $\rho_1$ and the slope of the linear trend $\phi$ from the $\{Z_i\}$. Results of these experiments are shown in the left panel of Figure R11 via the scatterplot between $\rho_1$ and $\phi$; the middle panel shows, for reference, the same scatterplot for the observed series; and the right panel presents instead the boxplots of the variance of the synthetic $\{Z_i\}$.

[Figure]

*Figure R11. Left: relationship between $\rho_1$ and $\phi$ of the 1000 $\{Z_i\}$ time series derived by continuous daily precipitation time series generated by the CoSMoS model for the four experiments (dots with different colors) along with the 95% confidence intervals of the INAR(1) model (black lines). Middle: relationship between $\rho_1$ and $\phi$ of the observed $\{Z_i\}$ time series (dots), along with 95% confidence intervals of the INAR(1) model (black lines). Right: boxplots of the variance of the 1000 $\{Z_i\}$ time series for the four experiments.*

The take-home messages of Figure R11 are as follows (note that I tested stronger serial correlations for $\{X_j\}$ or different parameters of the GG and obtained the same results):
1. Stationary time series of $\{X_j\}$ that are either correlated (StCor) or uncorrelated (StUncor) lead to practically the same variability of the $(\rho_1, \phi)$ pairs estimated on $\{Z_i\}$.
2. The variance of $\{Z_i\}$ for StUncor is similar to the mean and, thus, the distribution of $Z$ is well modeled by a Poisson distribution. On the other hand, the variance of $\{Z_i\}$ for StCor is larger than the mean, implying that the sample is overdispersed and, thus, better modeled by the BB distribution, as the author pointed out. However, the confidence interval (CI) of the INAR(1) model captures very well the variability of $(\rho_1, \phi)$ for both StUncor and StCor.

3. The presence of non-stationarity in the time series of $\{X_j\}$ leads to higher ($\rho_1$, $\phi$) of $\{Z_i\}$. These are outside the CI of the INAR(1) model based on the increment that was chosen for the scale parameter of the GG, and lie close to some of the observed pairs.
4. The presence of trends in $\{X_j\}$ inflates the variance of $\{Z_i\}$, although the largest factor that increases it is the presence of serial correlation.

Based on these findings, the assumption of stationarity for the application of the BB to model the distribution of $Z$ is not supported at all gages. Moreover, the INAR(1) is still useful to assess the nonstationary of the $Z$ time series derived from daily P records.

**3.3.* *Is the BB distribution "good" for $\{Z_i\}$ generated by nonstationary uncorrelated $\{X_j\}$?**
The author indicated that the BB is the theoretically correct distribution for serially correlated count series. The experiments conducted above show that trends in $\{X_j\}$ series introduce artificial serial correlations in the corresponding $\{Z_i\}$ even for uncorrelated $\{X_j\}$ (NonStUncor). In these situations, I found that the intra-cluster parameter of the BB can be "successfully" estimated based on the artificial correlations of $\{Y_j\}$. Therefore, since the BB "works", the theoretical considerations mentioned above would erroneously provide confidence in the presence of serial correlation and nonstationarity of the series. The results presented in section 3.2 indicate that this is most likely the case for several gages.

**4. Assumption of stationarity for the BB model applied to spatiotemporal precipitation time series**
The author applies the BB distribution for the counts at multiple sites with spatially and temporally correlated records under the assumption of stationarity of the correlations. Such an assumption has not been tested in any way by the author, and it might end up being as good or bad (as shown in my comments above) as the assumptions made to test trends that the author criticizes.

In this regard, the CIs in Figure 14 are obtained using the entire record of 100 years, but it could have been instead generated by applying the BB distribution using the first 50 years and results tested with the subsequent 50 years. I assume that other ways to test the assumption of stationarity of the correlations could be designed or, perhaps, found in the literature.

**5. Other comments:**

5.1: How were the CIs of the IAAFT model obtained in Figure 5?

5.2: Lines 302-303: How were independent peak-over-threshold events identified?

5.3: Lines 377-378: The author mentions that the INAR(1) model does not reproduce the autocorrelation of the observed data, but this was never proved.

5.4: More details are needed to explain how Figure 11 was generated. Also, the values of the estimated parameters of the BB model should be provided.

5.5: Line 539-540: is this a possible explanation of something that could not be proved? In other words, is this as "subjective" as the interpretation based on the presence of non-stationarity?

5.6: Line 594-596: I disagree with the author. Even if not perfect in terms of marginal distribution and based on some level of subjectivity on the trend (i.e., the simplest case of linearity), the calculations presented in this review confirm that the INAR(1) and NHP models, adopted by Farris et al. (2019), could be reasonably used as extreme cases to investigate where the statistics of the observed records are located compared to simple stationary and non-stationary models. Of course, better models are welcome to further improve the analyses.

**Reference**

Farris, S., Deidda, R., Viola, F., & Mascaro, G. (2021). On the role of serial correlation and field significance in detecting changes in extreme precipitation frequency. *Water Resources Research*, e2021WR030172. https://doi.org/10.1029/2021WR030172

Papalexiou, S. M. (2018). Unified theory for stochastic modelling of hydroclimatic processes: Preserving marginal distributions, correlation structures, and intermittency. *Advances in Water Resources*, *115*, 234–252. https://doi.org/10.1016/J.ADVWATRES.2018.02.013

Papalexiou, S. M., & Serinaldi, F. (2020). Random Fields Simplified: Preserving Marginal Distributions, Correlations, and Intermittency, With Applications From Rainfall to Humidity. *Water Resources Research*, *56*(2), e2019WR026331. https://doi.org/10.1029/2019WR026331

Papalexiou, S. M., Serinaldi, F., & Porcu, E. (2021). Advancing Space-Time Simulation of Random Fields: From Storms to Cyclones and Beyond. *Water Resources Research*, e2020WR029466. https://doi.org/10.1029/2020WR029466

Wilks, D. S. (2011). *Statistical methods in the atmospheric sciences*. International Geophysics Series, Academic Press. https://doi.org/10.1098/rspa.2007.0277

---

## Referee Comment (RC2)

**Review report on "Impact of spatio-temporal dependence on the frequency of precipitation extremes: Negligible or neglected?"**

by Demetris Koutsoyiannis

| | |
|---|---|
| Author(s): | Francesco Serinaldi (UK) |
| Journal: | *Hydrology and Earth System Sciences* |
| Journal's Ref.: | hess-2023-293 |
| Reviewer's Ref.: | DK-JR-383 |
| Date: | 2024-02-04 |
| Recommendation: | Major revision |

**Reviewer's assertion**: It is my opinion that a shift from anonymous to eponymous (signed) reviewing would help the scientific community to be more cooperative, democratic, equitable, ethical, productive and responsible. Therefore, it is my choice, consistent with my aesthetic attitude, to sign my reviews. Furthermore, I believe that the current trend in the review system to seek credit for anonymous transactions (by asking recognition for anonymous reviews through Web of Science, a practice also encouraged by journals) is problematic on ethical and aesthetic grounds. Only eponymous transactions can deserve recognition.

**Reviewer's clarification**: The references included in this review have the same meaning that references have in scientific documents. In brief, they justify the reviewer's statements and provide links where further details can be found. They are not meant to be suggestions for the author to include them in the paper in review.

1. I welcome this paper, which starts as "*Most of the methods reported in handbooks of applied statistics have been developed under the assumption of independence, distributional identity, and well-behaving bell-shaped/exponential distributions*" and intents to show the problems these assumptions cause in real-world hydrological applications, and to suggest remedies. I appreciate the investigation of temporal and spatial dependence as causes of spurious trends. I have devoted a book on these issues (Koutsoyiannis, 2023) as I believe they are important and not well known in the hydrological community. Hence, in principle I favour publication of the paper.

2. The paper makes several good points on epistemological grounds, related to concepts not well known or forgotten: the fact that statistical estimates (and diagrams) always rely on an underlying model; the importance of ergodicity for inference; the attention drawing on exploratory data analysis, including graphical diagnostics and theoretical considerations; the clarification of the meaning of stationarity and nonstationarity, which are not properties of the observed time series, but assumptions of the models we devise to describe physical processes.

3. At the same time, the paper tries to debunk some common misunderstandings, such as that dependence has limited effect on statistical inference and that statistical tests, typically devised for independent samples, are serious means of

inference for hydroclimatic processes. This may be acceptable for processes with short-range dependence but not for ones with long-range dependence (Cohn and Lins, 2005; Koutsoyiannis, 2023).

4. The paper is a result of a large amount of work and effort, based on analyses of a huge data set and using newly developed methodologies.

5. I particularly liked the graph in Fig. 8 (and similar figures in the Supplement). If I understand it well, there is not even one point with statistically significant trend. I also like the next figures that show that dependence alone can explain the behaviours observed, while independence would result in false trends.

6. A question not discussed in the paper is if trend identification has some usefulness or not. Let us assume that we have correctly (whatever this might mean) identified statistically significant trends based on past data. If our focus is on the past, what is the usefulness of a trend, once we have a better description by the data themselves? And if the focus is on the future, what is the value of such trend identification? Will a trend identified in the past continue in the future, or will it bend? Does the trend have any predictive value? By the way, this issue has been investigated by Iliopoulou and Koutsoyiannis (2020; see also the motto in that paper), but I would be interested to know the author's opinion on this. The current trend in detecting pointless trends continues to dominate in literature, and it would be useful and beneficial to the community if the paper discussed this issue.

7. Since the paper contains a sound epistemological part, it would be suitable to give a definition of a trend on a scientific (not colloquial) basis. I believe this notion, is more unclear than is popular and lacks a proper definition. I am interested to see what the author thinks about this issue.

8. On the negative side, the paper is difficult to read—and review. While the epistemological parts of the paper and the related questions are well discussed and clear, the technical parts are difficult to assimilate. Near the end of the paper, the author states that he offered an "*alternative point of view*". However, this is not clarified and is spread in many different sections without coherence. At least a summary of the approach proposed would be helpful and would improve the paper.

9. The structure of the paper is not optimal. Its style is not didactic, and, at places, it is too technical and unclear. The paper needs a substantial overhaul to make an attractive narrative—otherwise I think it would not be read.

10. A major negative issue is that the paper is aligned with another paper by Farris et al. (2021). It even attempts to present the methodology of that previous paper (section 3.1). In this respect, it does not look as an independent paper but a discussion on another paper. Yet it is not a formal discussion, as the paper by Farris et al. (2021) was published in another journal. I think the present paper reflects a sound work that could justify publication, but it needs reworking to be presented as a stand-alone paper.

11. It appears that the Hurst-Kolmogorov (HK) behaviour (long-term persistence; long-range dependence) is not present in the main paper but only in the Supplement. It has been shown (Iliopoulou and Koutsoyiannis, 2019) that

subsetting of a time series (using thresholds or block maxima) distorts the dependence and may hide that behaviour, but this does not mean its influence disappears. The main statistic used in the paper appears to be the lag one autocorrelation ($\rho_1$), but this does not capture the HK behaviour, so I doubt if it is appropriate. I think this would be useful to discuss in the paper.

12. Overall, it would be a pity if this work and the important points it makes were not published. On the other hand, its current form needs substantial improvement, before the paper can be publishable. I am sorry that I am not more specific in my comments, but, as I said, I had difficulties to read the paper.

**References**

Cohn, T.A., and Lins, H.F., 2005. Nature's style: Naturally trendy. *Geophysical Research Letters*, 32(23), L23402.

Farris, S., Deidda, R., Viola, F., and Mascaro, G., 2021. On the role of serial correlation and field significance in detecting changes in extreme precipitation frequency. *Water Resources Research*, 57, e2021WR030 172.

Iliopoulou, T., and Koutsoyiannis, D., 2019. Revealing hidden persistence in maximum rainfall records. *Hydrological Sciences Journal*, 64 (14), 1673–1689, doi: 10.1080/02626667.2019.1657578.

Iliopoulou, T., and Koutsoyiannis, D., 2020. Projecting the future of rainfall extremes: better classic than trendy. *Journal of Hydrology*, 588, doi: 10.1016/j.jhydrol.2020.125005.

Koutsoyiannis, D., 2023. *Stochastics of Hydroclimatic Extremes - A Cool Look at Risk*, Edition 3, ISBN: 978-618-85370-0-2, 391 pages, doi: 10.57713/kallipos-1, Kallipos Open Academic Editions, Athens.

---

## Author Comment (AC2)

**Impact of spatio-temporal dependence on the frequency of precipitation extremes: Negligible or neglected?**

**By F. Serinaldi**

**Submitted to *HESSD***

*MS-NR: hess-2023-293*
* * *
PRELIMINARY REPLY TO RC2 (DR. D. KOUTSOYIANNIS' REPORT)

(Note: In the text below, Referees' comments were copied verbatim in **black**. Replies are in **blue**)

I would like to thank the Reviewer for his remarks on the manuscript. Below, I provide a preliminary response to the main concerns. If the manuscript passes the first stage of review and revision is allowed, I will provide more detailed responses and an updated version of the manuscript.

6. A question not discussed in the paper is if trend identification has some usefulness or not. Let us assume that we have correctly (whatever this might mean) identified statistically significant trends based on past data. If our focus is on the past, what is the usefulness of a trend, once we have a better description by the data themselves? And if the focus is on the future, what is the value of such trend identification? Will a trend identified in the past continue in the future, or will it bend? Does the trend have any predictive value? By the way, this issue has been investigated by Iliopoulou and Koutsoyiannis (2020; see also the motto in that paper), but I would be interested to know the author's opinion on this. The current trend in detecting pointless trends continues to dominate in literature, and it would be useful and beneficial to the community if the paper discussed this issue.

**Response**

I understand Reviewer's concerns, but I think that the usefulness of detecting trends and prediction are secondary problems, as I believe that "*the trend in detecting pointless trends*" is related to more basic but fundamental problems that the paper tries to highlight.

Let me elaborate a bit more on this point before discussing my point of view on trends.

I'm used to distinguish statistical analysis and decision making, which can rely on statistical tools but is different. Now, if the information feeding decision making is based on a flawed preliminary statistical analysis, any discussion about usefulness, predictability, etc., is just pointless.

The aim of the paper is not to show that a model/method is better than another one, but to show that most of the "*the trend in detecting pointless trends*" results from systematically neglecting (or just lack of knowledge of) the epistemological rationale of statistical inference, which is is as follows (e.g., Aitken, 1947; Cramér, 1946; Box, 1976; Papoulis, 1991, von Storch and Zwiers, 2003… among others):

1) Make assumptions.
2) Build models and make inference accounting for the effect and consequences of those assumptions.
3) Interpret results according to the nature of the adopted models and their assumptions.

Most of the literature on trend analysis of (unrepeatable) hydroclimatic processes seems to neglect such a rationale and switches stage 1 with stage 2, resulting in the following fallacious procedure:

1) Select several models and methods based on different and often incompatible assumptions.
2) Make inference neglecting the constraints imposed by the different assumptions.
3) Interpret the results attempting to prove/disprove assumptions.

This approach, which corresponds to a widespread mechanistic use of statistical methods/software, suffers from logical fallacies. It neglects that models cannot be used to prove/disprove their own assumptions in the same way a mathematical theory cannot prove/disprove its own axioms and definitions. Indeed, those models and theories are valid only under those assumptions, axioms, and definitions. Of course, models cannot even be used to prove/disprove alternative assumptions as they might not even exists under those alternative hypotheses.

The paper compares the proper approach to statistical inference with its fallacious counterpart and shows practical consequences using a typical trend analysis of precipitation data as an example. I used the analysis reported by Farris et al. because it deals with a large data set and their analysis is rather detailed, thus looking rigorous and solid, but neglecting the effects of the original epistemological fallacies. However, I could have used many other examples: almost every paper reporting "trend analysis" suffers from the same problems.

For the sake of clarity, let me summarize how the twisting of inference logic impacts on the first step of the analysis reported in the paper, i.e. the selection of the marginal distribution, which is a typical exercise familiar to any hydrologist:

In a proper application of statistical inference (as it should be), we can consider for instance the following cases:

**Case 1**

- Assumption: precipitation occurrences are *assumed* to be consistent with a *stationary and independent* process.
- Under these conditions:
  - Poisson distribution is a suitable candidate for the marginal distribution of count process (of over-threshold occurrences).

- o If one wants, standard Goodness-of-Fit (GoF) tests (such as Kolmogorov-Smirnov, Cramer-von Mises, etc.) can be applied (… even if their application in this context is still problematic).
- Graphical and numerical results can say if Poisson is a defensible model *under stationarity and independence*. If Poisson does not fit satisfactorily, this does not prove/disprove its assumptions; in fact, there can be another distribution that works well under the same assumptions.

**Case 2**

- Assumption: precipitation occurrences are *assumed* to be consistent with a *stationary and dependent* process;
- Under these conditions:
  - o we expect overdispersion because of dependence. The Poisson distribution is known in advance not to be e suitable model from theoretical perspective. Therefore, we should consider some distribution allowing for overdispersion (e.g. negative Binomial, beta Binomial, etc.).
  - o In this case, GoF tests should account for variance-inflation of the test statistic due to dependence.
- Also in this case, empirical results do not prove/disprove any assumption; they just say if the adopted models provide a satisfactory description of data within the desired tolerance.

**Case 3**

- Assumption: precipitation occurrences are *assumed* to be consistent with a *nonstationary and independent* process;
- Under these conditions:
  - o we expect overdispersion because of non-stationarity. Indeed, in this case, every observation is *assumed* to come from different distributions. For example, if we *assume* that the rate of occurrence linearly increases in time, data might be *assumed* to come from a set of Poisson distributions with different rate parameter. Therefore, the resulting overall distribution is mixed/compound Poisson, which is over-dispersed. Such a distribution is not even unique, as it depends on the time window where it is computed.
  - o In this case, GoF tests cannot be applied to raw data because a unique (population) mixed Poisson does not exist, and such test can be applied at most to filtered (detrended) data (e.g., Coles, 2001) to check the behaviour of the conditional distribution, which is considered unique under the *assumption* that the filtered data are *conditionally stationary* (and a unique conditional distribution does exist). The Reviewer described this situation in various papers of him, and his book.
- Also in this case, empirical results do not prove/disprove any assumption, as the results are valid only under the assumptions used to make inference.

To summarize, following the rationale of statistical inference, both model selection and inference depend on the assumptions we make. Of course, we can use different assumptions (cases 1-3 above), develop the *complete* inference for each one (accounting for the corresponding constraints), and eventually choose the framework based on parsimony, accuracy, and generality of results (or predictability). What we cannot do is mix up models and tools that are valid under some assumptions in a different context and for different assumptions.

Such a misuse (corresponding to the fallacious approach mentioned above) is precisely what is routinely used in (too) many papers and generates logical contradictions and paradoxes. And this is the focus of my paper.

For example, Farris et al. use the Chi-square and Kolmogorov-Smirnov (Lilliefors) GoF tests concluding that the Poisson distribution cannot be rejected for more than 95% of cases at the 5% global significance in all cases (thresholds and samples sizes). Therefore *"the Poisson distribution is adopted as the parent distribution of count time series"*. However, their subsequent analysis is based on two models (INAR and NHP) that correspond to two assumptions (dependence and non-stationarity) that are incompatible with both Poisson distribution and the application of GoF tests as done in their Section 3.2.

Indeed, under such assumptions (dependence and non-stationarity), the distribution is expected to be over-dispersed. More precisely, under dependence, we have variance inflation, while under non-stationarity the distribution is not unique and it can be over-dispersed mixed-Poisson, at most. In other words, if the parent distribution of raw count data is assumed to be Poisson (based on GoF tests), NHP cannot be an option and vice versa: the same data cannot be simultaneously Poisson and mixed-Poisson, equi-dispersed and over-dispersed. This would violate the principle of non-contradiction.

Such a contradiction raises from neglecting the fact that (i) candidate models are different under different assumptions, (ii) such assumptions also impact on the form and interpretation of GoF tests, and (iii) outcomes of GoF tests under a specific assumption (e.g., dependence) are not valid under alternative assumptions (such as independence or non-stationarity).

The focus of the paper is on these issues extended to all the steps of the analysis of precipitation data.

In my opinion, these problems are more compelling than establishing the usefulness of some analysis' output, as they denote that statistical analysis is routinely performed (in some applied sciences) neglecting or ignoring the rationale of statistical inference and the practical negative consequences of such an approach to data analysis, i.e. a systematic and unavoidable misinterpretation of any result, independently of the models/frameworks used.

Going back to trends, Reviewer asked**: *'Can the author define what a trend is?"***

Let me try. In the context of the analyses reported in the paper and most of the literature on the topic, a trend should be intended as a well-defined (deterministic, if one wants) function or rule

linking a variable of interest (or one or more of its properties) to another variable describing a parametric support. For example, the Poisson regression model (non-homogeneous Poisson) used in the paper (and in Farris et al.) assumes that the rate of occurrence is a log-linear function of time

$$\log(\lambda(t)) = \lambda_0 + \phi \cdot t.$$

On the other hand, when we apply Mann-Kendall test, the tested trend is defined as follows (Mann 1945):

*"The hypothesis of a downward trend may be defined in the following way: The sample is still a random sample but $X_i$ has the continuous cumulative distribution function $f_i$ and $f_i(X) < f_{i+k}(X)$ for every i, every X, and every k >0. An upward trend is similarly defined with $f_i(X) > f_{i+k}(X)$."*

In this case, the property of concern is the so-called *stochastic dominance*. Thus, trends correspond to a well-defined rule implying that the distribution of *X* at time *i stochastically dominates* the distribution of *X* at any subsequent time *i+k*, or vice versa. In other words, it assumes that stochastic dominance is on play, and it depends on time.
Such a definition does not specify a precise formula *y* = g(*t*), but a precise rule, that is, systematic stochastic dominance. Since stochastic dominance often results in monotonic patterns of central tendency measures, Mann-Kendall test is often (but incorrectly) referred to as a test for "trends in the median/mean values".

The case of MK tests allows some considerations:
  - In contrast with many "tests" proposed nowadays, tests developed by professional statisticians in the past (century) were compliant with the rationale of statistical inference and were always explicit and rather precise about assumptions and preliminary definitions.
  - Based on the definition of trend given by Mann and the fact that the test statistic relies on ranks rather than absolute values, it is expected that such a test is less powerful than other parametric tests based on more restrictive assumptions of (conditional) Gaussianity and precise (non)linear function of the mean (or median or whatever else). On the other hand, parametric tests have higher Type I error under misspecification (which is always the case for non-repeatable processes), because there is no free lunch either in life or statistics.
  - Based on the foregoing remarks, it follows how many studies reporting power analysis of MK tests just show what is expected from the theory.

Therefore, different statistical methods and tests rely on their own definition of "trend". When tests are well devised, such definitions are clearly stated as they are necessary to set up a suitable and coherent test statistic and derive its properties.
However, (too) many end-users neglect the different definitions of trend implied by different tests/methods, which become just algorithms to be run over large data sets.

The consequence is not only merging methods that are devised to answer different questions (without recognizing such a difference) but also logical paradoxes.

For example, focusing on the analysis in the paper, some of the fitted non-homogeneous Poisson distributions (fitted under the assumption of non-stationarity) show negative slope, meaning decreasing rate of occurrence. Some of these models yield rate equal to zero in few decades. This has very practical and inconvenient consequences:

- The decreasing rate implies decreasing variance, which depends on time as well. This means that a single population variance does not exist, and autocovariance and autocorrelation are ill-defined as well. In other words, sample ACF has the same lack of meaning as the sample mean under the assumption that data come from a Cauchy distribution. Therefore, all ACFs or lag-1 ACF terms reported in the paper and in Farris et al. (… and in any other paper) under the assumption of non-stationarity do not correspond to any population counterpart. They are just numbers that do not allow any inference because the supposed inferred *population* counterpart does not even exist.
- Such models hinder the calculation of whatever index or summary statistics requiring integration over time, such as return periods and levels. Indeed, if the process no longer exists after e.g. 80 years (because the rate $\lambda$ become equal to zero after 80 years), calculating return levels over longer return period is just meaningless.

Moving from formal definitions to applications, the keyword in trend analysis of hydroclimatic data is "repeatability". A key aspect of science is experimentation: a result is scientifically sound if it can be replicated. Hydroclimatic processes cannot be repeated; however, some of them repeat themselves in time over time scales that allow multiple observations of the same phenomenon. In these cases, we implicitly or explicitly replace multiple independent experiments with multiple observations of the same process in time. In this respect seasonality is a trend (the distribution of a hydroclimatic variable is assumed to be a function of calendar days, or months, or similar).

What is the difference between seasonal trends and 'monotonic trends' over e.g. 100 years? Repeatability: the formers have been observed many times (let's say, Earth and Sun are so kind to repeat the experiment for us every 365 days), whereas 'monotonic trends' are not. The latter are unique. Making inference on them is like assessing if a die is loaded by throwing it just once.

Finally, there is a third aspect to consider. Referring to trends, I noted that Iliopoulou and Koutsoyiannis (2020) mentioned some econometric literature. This suggested a few additional considerations to me.

The world "trend" is used with different meanings not only in different disciplines but also in the same discipline. For example, when we deal with econometric models (ARCH, GARCH, or whatever else), trend and non-stationarity have necessarily the meaning resulting from the formal definition of stationarity (otherwise these models would not be compliant to mathematical derivations, asymptotic, etc.).

However, trends have a different meaning in technical (visual) analysis used to support investment and trading strategies. The figure below shows the last three months of daily data (Japanese candlesticks) of S&P500 stock index. At a first glance one can think that there is a "monotonic trend", and one can also attempt some prediction, as extrapolation does not seem so difficult.

[Figure]

However, if we expand to the view back to the last 7 months, we have a different picture. Thus, perhaps extrapolation could be not so easy.

[Figure]

Things are even more interesting if we go 13 and 24 months back (see figures below).
In such diagrams, "trends" are not mathematical functions, but only upward and downward fluctuations existing at various scales… I will not digress on the tons of papers and books dealing with scaling properties of these data after Mandelbrot's seminal works). Extrapolation based on the first diagram (last 3 months) can be very harmful.
Is this stuff predictable? Well, generally, the 95% of traders lose money, and "*Normally, forecasts should be as far from one another as they are from the predicted number*" (Taleb, 2010).

Here the massage is that there is a fundamental difference between stock market and hydroclimatic time series: the former result from a man-made process, while the latter do not.
People applying econometric models or statistical tests for trend detection (such as KPSS, Dickey-Fuller, etc.) often miss that such tests involve assumptions like (almost) infinite variance, which are unrealistic for hydroclimatic processes. This is, once again, the effect of using methods without checking their underlying theory, the context where they were developed, and the problems they attempted to solve.

Therefore, when dealing with 'trends', we should consider three aspects at least:

- Formal definition, which is required to develop any sound test or model. If such a definition is missing, the resulting methodology might be formally meaningless and useless. Results are just nonsense numbers. A few previous papers of mine (and co-authors) discuss some of these flawed algorithms.
- Type of data: experimental or not. Repeatability.
- Context: type of process we are dealing with (and we want to model), e.g., hydrological, biological, financial, etc.

Are these points exhaustive? No, of course. However, they indicate that these issues have multiple aspects, which should be considered.

[Figure]

[Figure]

7. Since the paper contains a sound epistemological part, it would be suitable to give a definition of a trend on a scientific (not colloquial) basis. I believe this notion, is more unclear than is popular and lacks a proper definition. I am interested to see what the author thinks about this issue.

**Response**

Please, see the foregoing discussion. I'll add a definition, but, as mentioned above, these issues have several facets, and would require another paper, at least.

In this respect, let me mention an anecdote that further highlights the actual problems affecting the literature on these topics. Recently, I read about a 'functional' definition of stationarity corresponding to a sort of lack of 'visual' trends in a sample.

Thus, even if one can try to be as rigorous as possible, this effort is rather useless if the potential readers do not even recognize the difference between sample and population properties described in the first chapter of every introductory handbook of applied statistics.

Thus, in my experience, the worrying problem is that there are many people working on data analysis (and publishing) with insufficient training (if any)… myself included, perhaps. Of course, the availability of powerful ready-to-use software contributes to this situation. In my opinion, easier access to computational resources and software should correspond to a more and more solid theoretical and epistemological background… however, the 'trend' seems to go in the opposite direction.

8. On the negative side, the paper is difficult to read—and review. While the epistemological parts of the paper and the related questions are well discussed and clear, the technical parts are difficult to assimilate. Near the end of the paper, the author states that he offered an "alternative point of view". However, this is not clarified and is spread in many different sections without coherence. At least a summary of the approach proposed would be helpful and would improve the paper.

**Response**
I'll try to clarify the text in the revised version (if any).

As mentioned above, the paper attempts to compare two approaches to statistical analysis, the usual one (which is the only one that is theoretically sound) and the fallacious one.

The 'alternative point of view' is just the standard way to make inference, which however looks like an 'alternative' nowadays, as the fallacious approach seems to be the rule in some literature.

The whole paper is a side-by-side comparison of the consequences of the two approaches when they are applied at each stage of the analysis of precipitation data.

In this respect, the structure and rationale of the paper is rather simple.

Moreover, understanding the logic (or lack of logic) of the various steps of analysis is much more important to me than focusing on technicalities. This is why all models and methods are reported in the supplementary material.

The aim is not to compare specific models but two different approaches to statistical inference, highlighting the logical inconsistencies of the second one, which is widespread but contrasts with the principles of scientific inquiry.

I'm aware that this type of content is not common and far from that of the typical technical paper showing "problem-(new) model-results", but I think that the above-mentioned issues are even more important, as they deal with how people use statistics, independently of the specific problem at hand.

9. The structure of the paper is not optimal. Its style is not didactic, and, at places, it is too technical and unclear. The paper needs a substantial overhaul to make an attractive narrative—otherwise I think it would not be read.

**Response**
I'll try to clarify the text. Please consider that, in the past, when I tried to be more didactic, I have been criticized because I was 'didactic', and this was deemed not appropriate in communications

among peers. When I try to be more technical (to be more consistent with the rules of communication among peers), I was asked to be more didactic.

I'll try to improve the presentation, but I'm sure that there will always be someone who does not like the paper for one reason and the opposite one at the same time.

By the way, the effort to be clear is often useless. For example, the abstract of a paper of mine states that a certain method "*is affected* by sample size, distribution shape, and serial correlation", but that paper is systematically cited as "*such a method is independent of sample size, distribution shape, and serial correlation (…, Serinaldi et al.,…)*".

This is not an isolated case. In my experience, often papers are systematically mis-cited just because readers seem not to recognize the difference between 'it is' and 'it is *not*'. Therefore, attempting to be clear is a good habit, but often pointless.

10. A major negative issue is that the paper is aligned with another paper by Farris et al. (2021). It even attempts to present the methodology of that previous paper (section 3.1). In this respect, it does not look as an independent paper but a discussion on another paper. Yet it is not a formal discussion, as the paper by Farris et al. (2021) was published in another journal. I think the present paper reflects a sound work that could justify publication, but it needs reworking to be presented as a stand-alone paper.

**Response**

Independent validation of results is another key aspect of science. Comparisons and replications of the same methods, analysis and experiments are the core of scientific enquiry, and are routinely applied in medicine, physics, etc.

However, in my experience, this is not the case of hydrological data analysis where we have a sort of binary approach: either a paper proposes something (supposedly) new, or we write a comment, which, however, must always be short because of journals' policies.

In my opinion, scientific debate cannot be relegated to short comments or verbal discussion 'in front of a beer' at some conference, and for sure most of the published papers (included mine) do not propose anything significantly new (despite the clichés emphasizing 'novelty' in abstracts and conclusions).

Therefore, the paper under review is part of a series of mine (with co-authors) falling in the class of so-called "neutral" validation studies, aiming at re-analysing methods, procedures, conjectures, etc., in detail.

This type of work attempts to double check methods and general concepts, using specific examples taken from previous works to provide a side-by-side comparison, to highlight the striking practical differences or possible inconsistencies resulting from different ways of reasoning (or lack of reasoning).

In my opinion, keeping the discussion general, as done in the past by Yevjevich (1968) and Klemes (1986), for instance, has historically been proven to be not very effective.

Scientific double-check needs to be as precise as possible, reproducing the results previously reported in the literature (to be sure why they are what they are) and then contrasting them (if required) with alternative methods.

To do that, we need to introduce the original procedures. Let me use an example to explain what I mean:

- Years ago, someone stated that they were able to make cold nuclear fusion (or something like that).

- These results were checked in detail.
- I was found that the claimed procedure did not work.

To do that double-check, it could not be sufficient to say, "We made some experiments, and we were not able to make cold fusion". Discussants had to explicitly refer to the original method and check every detail.

Was not independent check worth communication? Was this negative for science and technological advances?

Did the authors of the original findings feel happy? I do not think so. Probably they were not happy in the same way the supporters of Ptolemaic system were not about Copernican theory.

Should have people avoided to compare Ptolemaic with Copernican theory just not to hurt (the ego of) supporters of the former?

As mentioned above, I used the analysis reported by Farris et al. as quite a sophisticated example of how far can lead ignoring the rationale of statistical inference. Indeed, the message of that work is not the usual conclusion "precipitation is nonstationary" (which is already nonsense), but a bolder statement ("*Accounting for serial correlation in observed extreme precipitation frequency has limited impact on statistical trend analyses*"), which relies on the unscientific concepts that we can use models (generally the most trivial versions) to check their own assumptions or assumptions of other models. The criticisms reported in the paper are valid for most of the literature on the topic, and the concluding discussion is indeed fully general in this respect.

We need to refer to the original approaches if we want to highlight the paradoxes resulting from methodological misconceptions. Otherwise, the discussion would always be vague, and different approaches would look like different (but equally sound) points of view, which is not the case if one of them does not follow the logic of science.

Of course, one can call principles of science into question and introduce new ones. However, in any case, we must agree about which ones we want to use. Otherwise, it would be like attempting to communicate using different languages, or worse, using the same language with words referring to different meanings, that is, missing the link between signifiers and referents.

The paper focuses on these problems and was structured accordingly. I'll try to better clarify these issues.

11. It appears that the Hurst-Kolmogorov (HK) behaviour (long-term persistence; long-range dependence) is not present in the main paper but only in the Supplement. It has been shown (Iliopoulou and Koutsoyiannis, 2019) that subsetting of a time series (using thresholds or block maxima) distorts the dependence and may hide that behaviour, but this does not mean its influence disappears. The main statistic used in the paper appears to be the lag one autocorrelation ($\rho_1$), but this does not capture the HK behaviour, so I doubt if it is appropriate. I think this would be useful to discuss in the paper.

**Response**

HK is reported in the Supplementary along with all methods and technicalities (INAR, NHP, Beta-binomial, IAAFT, etc.).

This was purposely done to keep the discussion focused on the conceptual problem, that is, the consequences of switching from scientifically sound statistical inference to a logically fallacious approach that mixes up assumptions, models, and results.

The specific models used in the paper are secondary. We could use other models; what matters is how they are used, i.e., do we follow the rationale of statistical inference (which was obvious to any analysist until to the past century) or do we mix up everything confusing models, assumptions, etc, as routinely done in (too) many papers nowadays?

That said, I agree that ρ1 is not representative of HK (of course); moreover, (population) ρ1 does not even exist for NHP, as sample estimates (via standard estimators) do not correspond to any theoretical counterpart (inference is not possible). However, I had to use ρ1 estimates to make a direct comparison with previous results reported in the literature, thus showing precisely the logical and practical inconsistencies resulting from neglecting these theoretical issues.
I'll try to clarify these concepts in the revised version (if any).

12. Overall, it would be a pity if this work and the important points it makes were not published. On the other hand, its current form needs substantial improvement, before the paper can be publishable. I am sorry that I am not more specific in my comments, but, as I said, I had difficulties to read the paper.

**Response**
I'll do my best to improve the presentation.
Reviewers' comments made me realize that the actual message of the paper did not emerge clearly enough.

---

## Author Comment (AC3)

**Impact of spatio-temporal dependence on the frequency of precipitation extremes: Negligible or neglected?**

**By F. Serinaldi**

**Submitted to *HESSD***

*MS-NR: hess-2023-293*
* * *
**PRELIMINARY REPLY TO RC3**

As for the other Reviewers, I would like to thank the Reviewer for the time dedicated to the manuscript. Below, I provide a quick preliminary response. If the manuscript passes the first stage of review and revision is allowed, I will provide more detailed responses and an updated version of the manuscript.

If I correctly interpreted Reviewer's remarks, they raised two main points:

1) Proper acknowledgement of the literature, and
2) Making the paper more stand-alone.

Concerning the first point, some of the papers mentioned by the Reviewer are already cited, while others surely fit and will be added to the bibliography.

I have to say that I did not emphasize any specific method and the related literature just because the actual purpose of the paper is slightly different from what seems to emerge from the interpretation of the three Reviewers that commented the paper so far.

Obviously, if different people made similar interpretation, it means that the text is not clear and need to be revised accordingly.

Technical details concerning all models used in the paper are reported in the 'Supplementary' because the aim of the paper is not the comparison of specific models (or a specific types of analysis), but a discussion about the consequences of neglecting the rationale of statistical inference, using precipitation analysis as an example.

In this respect, let me refer to part of my responses to RC1 and RC2, as they apply to RC3 as well.

The aim of the paper is not to show that a model/method is better than another one, but to show that most of the trend analysis reported in the literature results from systematically neglecting (or

just lack of knowledge of) the epistemological rationale of statistical inference, which is is as follows (e.g., Aitken, 1947; Cramér, 1946; Box, 1976; Papoulis, 1991, von Storch and Zwiers, 2003… among others):

1) Make assumptions.
2) Build models and make inference accounting for the effect and consequences of those assumptions.
3) Interpret results according to the nature of the adopted models and their assumptions.

Most of the literature on trend analysis of (unrepeatable) hydroclimatic processes seems to neglect such a rationale and switches stage 1 with stage 2, resulting in the following fallacious procedure:

1) Select several models and methods based on different and often incompatible assumptions.
2) Make inference neglecting the constraints imposed by the different assumptions.
3) Interpret the results attempting to prove/disprove assumptions.

This approach, which corresponds to a widespread mechanistic use of statistical methods/software, suffers from logical fallacies. It neglects that models cannot be used to prove/disprove their own assumptions in the same way a mathematical theory cannot prove/disprove its own axioms and definitions. Indeed, those models and theories are valid only under those assumptions, axioms, and definitions. Of course, models cannot even be used to prove/disprove alternative assumptions as they might not even exists under those alternative hypotheses.

The paper compares the proper approach to statistical inference (which is the only one conceivable by every analyst until the past century) with its fallacious ('modern') counterpart and shows practical consequences, using a typical trend analysis of precipitation data as an example. I used the analysis reported by Farris et al. because it deals with a large data set and their analysis is rather detailed, thus looking rigorous and solid, but neglecting the effects of the original epistemological fallacies. However, I could have used many other examples: almost every paper reporting "trend analysis" suffers from the same problems.

For the sake of clarity, let me summarize how the twisting of inference logic impacts on the first step of the analysis reported in the paper, i.e. the selection of the marginal distribution, which is a typical exercise familiar to any hydrologist:

In a proper application of statistical inference (as it should be), we can consider for instance the following cases:

*Case 1*

- Assumption: precipitation occurrences are *assumed* to be consistent with a *stationary and independent* process.
- Under these conditions:
    o Poisson distribution is a suitable candidate for the marginal distribution of count process (of over-threshold occurrences).

- o If one wants, standard Goodness-of-Fit (GoF) tests (such as Kolmogorov-Smirnov, Cramer-von Mises, etc.) can be applied (… even if their application in this context is still problematic).
- Graphical and numerical results can say if Poisson is a defensible model *under stationarity and independence*. If Poisson does not fit satisfactorily, this does not prove/disprove its assumptions; in fact, there can be another distribution that works well under the same assumptions.

**Case 2**

- Assumption: precipitation occurrences are *assumed* to be consistent with a *stationary and dependent* process;
- Under these conditions:
  - o we expect overdispersion because of dependence. The Poisson distribution is known in advance not to be e suitable model from theoretical perspective. Therefore, we should consider some distribution allowing for overdispersion (e.g. negative Binomial, beta Binomial, etc.).
  - o In this case, GoF tests should account for variance-inflation of the test statistic due to dependence.
- Also in this case, empirical results do not prove/disprove any assumption; they just say if the adopted models provide a satisfactory description of data within the desired tolerance.

**Case 3**

- Assumption: precipitation occurrences are *assumed* to be consistent with a *nonstationary and independent* process;
- Under these conditions:
  - o we expect overdispersion because of non-stationarity. Indeed, in this case, every observation is *assumed* to come from different distributions. For example, if we *assume* that the rate of occurrence linearly increases in time, data might be *assumed* to come from a set of Poisson distributions with different rate parameter. Therefore, the resulting overall distribution is mixed/compound Poisson, which is over-dispersed. Such a distribution is not even unique, as it depends on the time window where it is computed.
  - o In this case, GoF tests cannot be applied to raw data because a unique (population) mixed Poisson does not exist, and such test can be applied at most to filtered (detrended) data (e.g., Coles, 2001) to check the behaviour of the conditional distribution, which is considered unique under the *assumption* that the filtered data are *conditionally stationary* (and a unique conditional distribution does exist).
- Also in this case, empirical results do not prove/disprove any assumption, as the results are valid only under the assumptions used to make inference.

To summarize, following the rationale of statistical inference, both model selection and inference depend on the assumptions we make. Of course, we can use different assumptions (cases 1-3

above), develop the *complete* inference for each one (accounting for the corresponding constraints), and eventually choose the framework based on parsimony, accuracy, and generality of results (or predictability). What we cannot do is mix up models and tools that are valid under some assumptions in a different context and for different assumptions.

Such a misuse (corresponding to the fallacious approach mentioned above) is precisely what is routinely used in (too) many papers and generates logical contradictions and paradoxes. And this is the focus of my paper.

For example, Farris et al. use the Chi-square and Kolmogorov-Smirnov (Lilliefors) GoF tests concluding that the Poisson distribution cannot be rejected for more than 95% of cases at the 5% global significance in all cases (thresholds and samples sizes). Therefore "*the Poisson distribution is adopted as the parent distribution of count time series*". However, their subsequent analysis is based on two models (INAR and NHP) that correspond to two assumptions (dependence and non-stationarity) that are incompatible with both Poisson distribution and the application of GoF tests as done in their Section 3.2.

Indeed, under such assumptions (dependence and non-stationarity), the distribution is expected to be over-dispersed. More precisely, under dependence, we have variance inflation, while under non-stationarity the distribution is not unique and it can be over-dispersed mixed-Poisson, at most. In other words, if the parent distribution of raw count data is assumed to be Poisson (based on GoF tests), NHP cannot be an option and vice versa: the same data cannot be simultaneously Poisson and mixed-Poisson, equi-dispersed and over-dispersed. This would violate the principle of non-contradiction.

Such a contradiction raises from neglecting the fact that (i) candidate models are different under different assumptions, (ii) such assumptions also impact on the form and interpretation of GoF tests, and (iii) outcomes of GoF tests under a specific assumption (e.g., dependence) are not valid under alternative assumptions (such as independence or non-stationarity).

The focus of the paper is on these issues extended to all the steps of the analysis of precipitation data.

In my opinion, these problems are very compelling, as they denote that statistical analysis is routinely performed (in some applied sciences) neglecting or ignoring the rationale of statistical inference and therefore the practical negative consequences of such an approach to data analysis, i.e. a systematic and unavoidable misinterpretation of any result, independently of the models/frameworks used.

These remarks lead to the second concern raised by the Reviewer (and RC2): a stand-alone paper.

Independent validation of results is a key aspect of science. Comparisons and replications of the same methods, analysis and experiments are the core of scientific enquiry, and are routinely applied in medicine, physics, etc.

However, in my experience, this is not the case of hydrological data analysis where we have a sort of binary approach: either a paper proposes something (supposedly) new, or we write a comment, which, however, must always be short because of journals' policies.

In my opinion, scientific debate cannot be relegated to short comments or verbal discussion 'in front of a beer' at some conference, and for sure most of the published papers (included mine) do not propose anything significantly new (despite the clichés emphasizing 'novelty' in abstracts and conclusions).

Therefore, the paper under review is part of a series of mine (with co-authors) falling in the class of so-called "neutral" validation studies, aiming at re-analysing methods, procedures, conjectures, etc., in detail.

This type of work attempts to double check methods and general concepts, using specific examples taken from previous works to provide a side-by-side comparison, to highlight the striking practical differences or possible inconsistencies resulting from different ways of reasoning (or lack of reasoning).

In my opinion, keeping the discussion general, as done in the past for instance by Yevjevich (1968) and Klemes (1986) without referring to specific examples, has historically been proven to be not very effective.

Scientific double-check needs to be as precise as possible, reproducing the results previously reported in the literature (to be sure why they are what they are) and then contrasting them (if required) with alternative methods.

To do that, we need to introduce the original procedures. Let me use an example to explain what I mean:

- Years ago, someone stated that they were able to make cold nuclear fusion (or something like that).
- These results were checked in detail.
- I was found that the claimed procedure did not work.

To do that double-check, it could not be sufficient to say, "We made some experiments, and we were not able to make cold fusion". Discussants had to explicitly refer to the original method and check every detail.

Was not independent check worth communication? Was this negative for science and technological advances?

Did the authors of the original findings feel happy? I do not think so. Probably they were not happy in the same way the supporters of Ptolemaic system were not about Copernican theory.

Should have people avoided to compare Ptolemaic with Copernican theory just not to hurt (the ego of) supporters of the former?

As mentioned above, I used the analysis reported by Farris et al. as quite a sophisticated example of how far can lead ignoring the rationale of statistical inference. Indeed, the message of that work is not the usual conclusion "precipitation is nonstationary" (which is already nonsense), but a bolder statement ("*Accounting for serial correlation in observed extreme precipitation frequency has limited impact on statistical trend analyses*"), which relies on the unscientific concepts that we can use models (generally the most trivial versions) to check their own assumptions or assumptions of other models. The criticisms reported in the paper are valid for most of the literature on the topic, and the concluding discussion is indeed fully general in this respect.

We need to refer to the original approaches if we want to highlight the paradoxes resulting from methodological misconceptions. Otherwise, the discussion would always be vague, and different approaches would look like different (but equally sound) points of view, which is not the case if one of them does not follow the logic of science.

Of course, one can call principles of science into question and introduce new ones. However, in any case, we must agree about which ones we want to use. Otherwise, it would be like attempting to communicate using different languages, or worse, using the same language with words referring to different meanings, that is, missing the link between signifiers and referents.

The paper focuses on these problems and was structured accordingly, and it follows the same structure of previous papers of mine of the same kind… I have to say that, in my experience, years ago this type of papers and discussions were more welcome, while lately they seem to be less 'tolerated'. By the way, *"the times they are a-changin'"* and perhaps a bit 'harsh' but detailed scientific debate is no longer suitable for the *"brave new world"* we are building. Not sure this is positive, but the time will say.

That said, I'll try to better clarify these issues in the revised manuscript if it passes the first stage of reviews.

---

## Author Response (AR1)

**Scientific logic and spatio-temporal dependence in analyzing extreme precipitation frequency: Negligible or neglected?**

**Previously: "*Impact of spatio-temporal dependence on the frequency of precipitation extremes: Negligible or neglected?*"**

**By F. Serinaldi**

**Submitted to *HESSD***

***MS-NR: hess-2023-293***
* * *
**REPLY**

(Note: In the text below, Referees' comments were copied verbatim in **black**.)

I would like to thank the Editor for handling the manuscript and the Reviewers for their feedback. In the following, point-by-point responses are reported in **blue**. Changes in the manuscript are highlighted in **blue** as well. Responses below update and/or complement the content of preliminary replies to Reviewers' reports.

**REPLY ON EDITOR'S REPORT**

Dear Francesco Serinaldi,

Thank you for posting your replies online to the comments of the three referees. The reviewers, experts in their fields, evaluated the manuscript with different recommendations, ranging from minor revisions to rejection of the paper. Reading the manuscript once again, as well as their comments and your replies, I believe the paper can be ultimately accepted for publication in HESS after considerable revisions are made.

The reviewers raised several critical points on different aspects that I would kindly ask you to consider. A common comment, which you also refer to in your reply, is that the text requires some polishing to better explain the goal, rationale, and outcomes of the cases you examined. Maybe consider simplifying the text in such a way that non-statistical experts (most hydrologists I would

argue) could be able to understand the question in hand and methodology in their first read of the text.

I am looking forward to receiving and reading the revised version of the manuscript. Please also provide a detailed point-by-point reply to all of the comments of the reviewer. The manuscript will be then sent for a second round of expert evaluation.

Sincerely,

Nadav Peleg

**Response**

Dear Editor,

Thanks again for handling the manuscript and allowing me to revise it.

The heterogeneity of the recommendations is not so surprising as this type of papers is generally quite controversial and divisive. In fact, their scope is to promote the scientific debate.

Concerning the presentation, I attempted to better clarify the aim of the manuscript as well as some technical details. This resulted in:

- New title.
- Expanded introduction.
- New Section 4.4.
- Updated Figures 2 and 9, and Table 1.
- New Section S3.
- Several changes throughout the text.

Nonetheless, I am not sure that the text can be simplified much more. As mentioned in my responses to Reviewers, all technicalities are reported in the Supplement to keep the discussion focused on conceptual/epistemological aspects, which should be easy to grasp for anyone.

In my opinion, the used methods are not very difficult or sophisticated (they have been around for years or decades), but I recognize that they can look "new" or "complicated" to non-experts. Nonetheless, technicalities are just instrumental to deliver a very simple message that is understandable even without statistical knowledge, that is, what really matters are not specific models/methods we use, but how we use them. In particular, any analysis is unavoidably misinterpreted if we analyze data neglecting the rationale of scientific method.

In this respect, there is no need to fully understand technicalities. It is sufficient to understand that the same (or similar) methods applied to the same data can yield contrasting results depending on if we apply those methods according to the rationale of scientific method or neglecting it.

I hope that the revised version of the paper conveys this message to non-experts as well. For those who would be interested in technicalities, a second reading and/or further study of such methods

and models is suggested, and perhaps required. In this respect, Supplement and references provide plenty of resources.

As well summarized in the title, the main goal of this paper is to demonstrate the importance of accounting for spatio-temporal dependence on the frequency of precipitation extremes when investigating the possible presence of non-stationarity. The paper is motivated, in part, by a recent study by Farris et al. (2021) who performed analyses of long-term daily precipitation records covering several regions of the world to investigate the importance of serial correlation and field significance in trend analysis. In the manuscript under review here, most of the analyses of Farris et al. (2021) are repeated using alternative models and approaches that the author believes are more correct for the purpose. The main critiques raised by the author against the work of Farris et al. (2021) are (1) the adoption of the INAR(1)-Poisson and Non-Homogeneous Poisson (NHP) models to simulate time series that reproduce realistic stationary autocorrelated series and uncorrelated series with trends, respectively; and (2) the use of statistical tests to detect trends (and in general the 'standard' approaches used in the literature on trend analyses). To pursue his main goal, the author presents figures and analyses aimed at showing (and erroneous proving) that:

• The INAR(1) and NHP used in Farris et al. (2021) are not proper models of count time series of over-threshold (OT) daily precipitation (P) series since they do not capture the marginal distribution of the dataset.

• The hypothesis of no-trend is verified in all cases using the Iterative Amplitude Adjusted Fourier Transform (IAAFT) model after the power spectrum is bias corrected to account for the sample size and the field significance is considered.

• The Beta-Binomial (BB) distribution parameterized through the empirical spatial, temporal, and spatiotemporal linear correlation structure of the binary process of daily OT P occurrences captures the distribution of annual OT counts. Since the binary process used to parameterize the BB is assumed stationary by the author, it is concluded that there is no need to apply any trend test.

**Response**
I think that such a summary does not describe the actual content of the manuscript very well.

- I criticize the misuse of INAR(1) and NHP to draw conclusions about their own assumptions and/or alternative assumptions, neglecting the effect of such assumptions on the models themselves, their inference, and the interpretation of results.

- The paper does not verify any hypothesis; conversely, it shows the logical fallacy of attempting to use specific models and tests to check their own assumptions. In the present context, statistical tests are used only to show their inconsistency.

- There is no logical link between using a Beta Binomial distribution (and/or the stationary assumption) and the need to perform whatever statistical test. Such a link or conclusion is not stated anywhere in the paper. What I state throughout the paper (and discuss in detail in section 6.2) is that the outcome of any statistical test is redundant as it just confirms what is expected from its own assumptions (and underlying models). Using trend tests in the context of unrepeatable processes is uninformative even if we assume non-stationarity, non-homogenous Poisson (NHP) models, or whatever else.

Let me summarize once again the content of the paper: it attempts to clarify the practical consequences of neglecting the epistemological rationale of statistical inference, which is as follows (e.g., Aitken, 1947; Cramér, 1946; Papoulis, 1991, von Storch and Zwiers, 2003):

1) Make assumptions.
2) Build models and make inference accounting for the effect and consequences of those assumptions.
3) Interpret results according to the nature of the adopted models and their assumptions.

Most of the literature on trend analysis of (unrepeatable) hydroclimatic processes, including the work by Farris et al. and Reviewer's report, seems to neglect such a rationale and switches stage 1 with stage 2, resulting in the following fallacious procedure:

1) Select several models and methods based on different and often incompatible assumptions.
2) Make inference neglecting the constraints imposed by the different assumptions.
3) Interpret the results attempting to prove/disprove models' assumptions.

This approach, which corresponds to a widespread mechanistic use of statistical methods/software, suffers from logical fallacies. It neglects that models cannot be used to prove/disprove their own assumptions in the same way a mathematical theory cannot prove/disprove its own axioms and definitions. This is because those models and theories are valid only under those assumptions, axioms, and definitions. Of course, such models cannot even be used to prove/disprove alternative assumptions as they might not even exists under those alternative hypotheses.

The paper compares the proper approach to statistical inference with its fallacious counterpart and shows practical consequences using a typical trend analysis of precipitation data as an example. Even if the paper focuses on the data and analysis presented by Farris et al., the discussion is valid for many similar works approaching data analysis in the same (questionable) way.

For the sake of further clarity, let me summarize how flawed inference logic impacts on the first step of the analysis reported in the paper, i.e. the selection of the marginal distribution, which is a typical exercise familiar to any hydrologist:

In a proper application of statistical inference (as it should be), we can consider for instance the following cases:

*Case 1*

- Assumption: precipitation occurrences are *assumed* to be consistent with a *stationary and independent* process.
- Under these conditions:
    o Poisson distribution is a suitable candidate for the marginal distribution of count process (of over-threshold occurrences).
    o If one wants, standard Goodness-of-Fit (GoF) tests (such as Kolmogorov-Smirnov, Cramer-von Mises, etc.) can be applied.

- Graphical and numerical results can say if Poisson is a defensible model *under stationarity and independence*. If Poisson does not fit satisfactorily, this does not prove/disprove its assumptions; in fact, there can be another distribution that works well under the same assumptions.

**Case 2**

- Assumption: precipitation occurrences are *assumed* to be consistent with a *stationary and dependent* process.
- Under these conditions:
    o we expect overdispersion because of dependence. The Poisson distribution is known in advance not to be a suitable model from theoretical perspective. Therefore, we should consider some distribution allowing for overdispersion (e.g. negative Binomial, beta Binomial, etc.).
    o In this case, GoF tests should account for variance-inflation of the test statistic due to dependence.
- Also in this case, empirical results do not prove/disprove any assumption; they just say if the adopted models provide a satisfactory description of data within the desired tolerance.

**Case 3**

- Assumption: precipitation occurrences are *assumed* to be consistent with a *nonstationary and independent* process.
- Under these conditions:
    o we expect overdispersion because of non-stationarity. Indeed, in this case, every observation is *assumed* to come from different distributions. For example, if we *assume* that the rate of occurrence linearly increases in time, data might be *assumed* to come from a set of Poisson distributions with different rate parameter. Therefore, the resulting overall distribution is mixed/compound Poisson, which is over-dispersed. Such a distribution is not even unique, as it depends on the time window where it is computed.
    o In this case, GoF tests cannot be applied to raw data because a unique (population) mixed Poisson does not exist, and such test can be applied at most to filtered (detrended) data (e.g., Coles, 2001) to check the behaviour of the conditional distribution, which is considered unique under the *assumption* that the filtered data are *conditionally stationary* (and a unique conditional distribution does exist).
- Also in this case, empirical results do not prove/disprove any assumption, as the results are valid only under the assumptions used to make inference.

To summarize, following the rationale of statistical inference, both model selection and inference depend on the assumptions we make. Of course, we can use different assumptions (cases 1-3 above), develop the *complete* inference for each one (accounting for the corresponding

constraints), and eventually choose the framework based on parsimony, accuracy, and generality of results. What we cannot do is mix up models and tools that are valid under some assumptions in a different context and for different assumptions.

Such a misuse (corresponding to the fallacious approach mentioned above) is precisely what is routinely used in (too) many papers and generates logical contradictions and paradoxes.

For example, Farris et al. use the Chi-square and Kolmogorov-Smirnov (Lilliefors) GoF tests concluding that the Poisson distribution cannot be rejected for more than 95% of cases at the 5% global significance in all cases (thresholds and samples sizes). Therefore "*the Poisson distribution is adopted as the parent distribution of count time series*". However, their subsequent analysis is based on two models (INAR and NHP) that correspond to two assumptions (dependence and non-stationarity) that are incompatible with both Poisson distribution and the application of GoF tests as done in their Section 3.2.

Indeed, under such assumptions (dependence and non-stationarity), the distribution is expected to be over-dispersed. More precisely, under dependence, we have variance inflation, while under non-stationarity the distribution is not unique and it can be over-dispersed mixed-Poisson, at most. In other words, if the parent distribution of raw count data is assumed to be Poisson, NHP cannot be an option and vice versa: the same data cannot be simultaneously Poisson and mixed-Poisson, equi-dispersed and over-dispersed. This would violate the principle of non-contradiction… unless we make the *assumption* that the precipitation is well described by quantum mechanics, and Heisenberg's indetermination principle is acceptable at such spatio-temporal scales.

Such a logical contradiction raises from neglecting the fact that (i) candidate models are different under different assumptions, (ii) such assumptions also impact on the form and interpretation of GoF tests, and (iii) outcomes of GoF tests under a specific assumption (e.g., dependence) are not valid under alternative assumptions (such as independence or non-stationarity).

The focus of the paper and responses below is on these issues extended to all the steps of the analysis of precipitation data.

New title, abstract and expanded introduction better clarify the scope of the paper and its organization.

**Value of the BB distribution for the time series of annual counts {$Z_i$}**

While I understand the reasonings of adopting the BB as a reasonable distribution to represent the correct marginal distribution of stationary time series of counts that exhibit serial correlation, I disagree with the author attempts to show that this is also true from the practical point of view by comparing it with the Poisson distribution in Figures 2 and 3. I have separate concerns related to these figures.

**Scatterplot between mean and variance of Z**
In Figure 2, the author presents scatterplots between the mean and variance of $Z$ for the observed OT samples along with those of synthetic samples obtained from Poisson, NHP, and BB models. I

tried to reproduce Figure 2 for OT above the 95% empirical quantiles and reported results in Figure R1. I first point out that the mean of the observed samples should not vary, because it is prescribed by the quantile adopted for $x^*$. For the 95% empirical quantiles, it should be $x^* = (1 - 0.95)*365.25 = 18.2625$. This is not the case in Figure 2 of the paper, where the mean exhibits a negatively skewed spread around the expected value of 18.2625. Based on my interpretation, which I used to generate Figure R1, this is an artifact caused by the fact that the author did not account for the presence of repeated values in the $\{Xj\}$ series when applying the condition ($Xj > x^*$) to ultimately compute $\{Zi\}$. Note that a positively skewed spread for the mean (essentially, a mirrored version of the spread in Figure 2) is instead obtained by applying the condition ($Xj \geq x^*$).

[Figure]

Figure R1. Relationship between mean and variance of observed OT samples and those corresponding to simulated samples from (a) Poisson, (b) NHP, and (c) BB distributions.

That said, I produced Figure R1 by (i) estimating the parameter/s of each model separately on each observed sample, and (ii) generating a sample with the same size as the observation with the estimated parameter/s. This approach mimics what the author indicated in the caption: "*Observed values are compared with those corresponding to simulated samples from Poisson, NHP, and Beta-Binomial (BB) distributions*". If one follows this approach, one random generation of the synthetic samples of the three models should result in a sampling variability for the mean larger than that of observations and symmetric around the expected value defined by the 95% threshold (by the way: note that, in Figure R1, the variability of mean and variance is the same for the Poisson variates, as expected). In Figure 2 of the paper, the means of the randomly generated samples have instead exactly the same range as the observations. This is incorrect.

**Response**
- $x^*$ is the threshold of the precipitation intensity process, whereas 18.2625 is the theoretical expectation of the number of over-threshold occurrences over 365.25 time steps (Bernoulli trials).
- I recognize that the text might be not clear enough. Obviously, the orange points in Fig.2 are not the coloured points that the Reviewer reports in his Fig. R1 (indeed, the diagrams are obviously different).
  Let me explain. If we simulate 100 values from a Poisson distribution, the sample average is obviously characterized by large variability. To properly link the sample variances of the simulated samples with the parent distributions for effective visualization, we can choose different strategies. For example: (1) plotting the variance of the simulated sample versus the mean of the observed sample (i.e., the parameter of the generating Poisson/NHP/BB distribution), or (2) simulating $B$ time series for each location (e.g., $B$=100), then plotting the averages of the $B$ values of mean and variance (along with their ranges) or (3) plotting directly observed variances vs simulated ones.
  The first approach is what is reported in the original submission, as it provides very simple and clear representation, while the second method is used to create the figure

below (which is used in the revised manuscript), where the orange dots denote the averages over 100 simulations for each location, and the vertical and horizontal lines the range around those averages.

[Figure]

The third method yields the figure below.

[Figure]

One can see that the three approaches provide the same message: Poisson and NHP do not describe the observed variability. Is BB perfect? No, of course: it is just a model. However, BB works much better than Poisson and NHP.

So, why do Reviewer's single "*random generations*" look different from the observed "cloud" of points? Because the 1106 observed means are not random samples from a set of Poisson/NHP/BB distributions (obviously), but the parameters (expectations) of the Poisson/NHP/BB models used to simulate and resulting from a thresholding procedure.

Do things change if we extract events such that the average number is exactly 18.2625 events/year? No, really. Both variances of *Z* (see figure below) and CDFs (not shown) do not change very much.

[Figure]

Therefore, independently of OT selection and diagnostic diagrams:
1) the number of OT occurrences is over-dispersed, as expected from theoretical considerations.
2) Poisson and NHP cannot describe overdispersion, and they are not suitable models for these data.
3) If the marginal distribution is assumed to be Poisson (in contrast with empirical evidence), data cannot be modelled with NHP because the marginal distribution cannot be simultaneously Poisson and mixed-Poisson (this is a logical contradiction… at least in classic mechanics). For NHP, one could check *conditional* Poisson behaviour, at most.
4) Whatever additional analysis based on Poisson or NHP models (as those reported for instance by Farris et al.) is uninformative, as it relies on models that do not describe the observations.

**CDFs of observed samples and fitted distributions**
Apart from the issue indicated above, Figure 2 (and Figure R1 above) shows that, for several cases, the variance of the observed samples is larger than the mean. The one-parameter Poisson distribution is unable to fully capture this spread, while, as well known, a two-parameter distribution like the BB can be fitted to reproduce both the mean and the variance. To further prove the point that the BB is a better distribution, the author presents in Figure 3 cumulative distribution functions (CDFs) and differences in probability of observed and fitted distribution (I believe). However, the author did not explain how that figure was created. What are the lower and upper limits and how were they calculated?

*Response*
Please read L316-317: *"We compare ECDFs Fn(z) with the CDFs FP(z) and FβB(z) of Poisson and βB models, respectively. Probability plots are complemented with diagrams of the differences $(F_n(z) - F_{model}(z))$ versus z."*

Interval calculation is standard: take the 1106 CDFs, calculate the 1106 probability values corresponding to a set of quantiles, and calculate minimum, mean (or median), and maximum probability (or pointwise CIs) for each quantile. Alternatively, one can draw the 1106 CDFs (as further discussed below). The resulting diagram (see Fig.3b, reported below for convenience) shows that the ensemble of BB distributions can describe the global over-dispersion characterizing the observed samples reasonably well. On the other hand, the Poisson distribution cannot. Of course, these results are consistent with the foregoing mean/variance diagrams.

[Figure]

More importantly: can we clearly say that the two-parameter BB distribution provides a large improvement compared to the one-parameter Poisson distribution when looking at panels (e) and (f) of Figure 3 in the paper? Since I could not understand it well, I compared, for some representative gages (selected based on an equiprobability criteria to fairly explore all possible behaviors), the empirical CDF of $Zi$ and the CDFs of the fitted Poisson and BB distributions (see comment 3.1 regarding how parameters of the BB were estimated). I chose the gages based on the variance of $Zi$, whose empirical CDF across all gages is shown in Figure R2. I picked the gages with variance associated with a cumulative frequency, $F$, close to 0.1, 0.2, 0.3, …, 0.9. Results are shown in Figure R3: these graphical diagnostics (which the author recommends using in general) indicate that the two distributions are not markedly different, even for the largest values of the variance. This is also quantified by the values of the Cramer-von Mises goodness-of-fit metric (without any penalty) provided in the legend, which are very similar despite the BB distribution having an additional parameter compared to the Poisson, ranging from 8 to 22 for BB and from 9 to 24 for Poisson fitting. In addition, the variability of such metric does not seem related to the variance of $Zi$, suggesting that the gain of applying the BB against the Poisson when the variance of $Zi$ becomes increasingly larger than the mean is negligible. Therefore, for most cases, a parsimonious one-parameter distribution like the Poisson does not seem to do a bad job when characterizing the frequency of the empirical counts as compared to the BB (as proposed by the author), which depends on two parameters, and thus should be properly penalized in any comparison.

[Figure]

Figure R3. CDF of the observed Z samples, along with fitted Poisson and BB distributions. In each panel, the gage ID is shown along with the values of the variance of $Z_i$ and the cumulative frequency of the variance from Figure R2. The values reported in the legends next to "Poisson" and "BB" are the values of the Cramer-von Mises goodness-of-fit statistics.

*Response*
- Diagrams are powerful tools either to emphasize evidence or to conceal it. Using nine separate figures with a variable range of the x-axis tends to conceal the fact that the Poisson distribution shows (almost) no variability compared with the BB.
- A fairer visualization is possible by drawing the nine samples in the same panel. Panels (a) and (b) in figure below provide such a fairer comparison, showing that the BB model allows for capturing the observed variability of the lower/upper tails of the empirical distributions (over-dispersion), while the nine Poisson distributions are almost identical (they are identical if we use the rule of extracting events such that the average is exactly 18.2625 events/year).
- More importantly, the comparison of panels (a) and (b) with panels (c) and (d) in figure below show that *the nine time series selected by the Reviewer are not representative of the over-dispersion of the 1106 time series*. When we consider the whole sample, the difference between BB and Poisson is more evident (obviously).
- Drawing conclusions from the 0.8% of data is rather questionable, especially if we can easily look at the whole sample (by proper diagrams). For sure, the 0.8% of data are far from being "*most cases*".

[Figure]

- Reporting values of test statistics is uninformative as they are distance metrics affected by their own sampling variability. They can be compared only when they are associated to their p-values or the statistical test is non-parametric (i.e., the sampling distribution of the test statistics does not depend on the tested distribution), which is not the case here.
- Of course, roughly speaking, reporting p-values would mean performing a GoF test. In this respect, the Reviewer does not even need to report such information because Farris et al. already concluded that the Poisson model would be a good distribution for the whole sample. However, such conclusions contradict the evidence reported in the foregoing figures as well as the p-values reported in Fig. 4 of the paper (for three different GoF tests), which indicate that the Poisson distribution should be rejected up to 53% (as expected) when using a more powerful test for Poisson hypothesis.
- Parsimony: BB is not a distribution selected among a bunch of models by playing with GoF tests (as routinely done in the literature). BB is one of the models theoretically justified under some specific assumptions, which are deemed to be reasonable for the process at hand. This means that, if it works, it is expected to do so over a range of spatio-temporal scales in the same way any sound mathematical theory or physical model is expected to work in a range of situations. In this respect, the foregoing figures along with Figs. 3, 11, and 13 clearly show that such models work at least from daily to annual scale and from at-site to global scale. Such figures are reported below for the sake of convenience, as some reader can find some evidence in them.

[Figure]

**Figure 11.** ECDFs of number of OT events (for the 95% and 99.5% thresholds) occurring at daily time scale over different regions along with Binomial and $\beta\mathcal{B}$ CDFs.

[Figure]

**Figure 13.** ECDFs of number of OT events (for the 95% and 99.5% thresholds) occurring at annual time scale over different regions along with Binomial and $\beta\mathcal{B}$ distributions.

**Why is bias correcting the power spectrum needed?**

The author mentions the use of bias correction of the power spectrum, assuming the presence of fractional Gaussian noise, as described in the Supporting Information. However, the author did not properly explain nor demonstrate why this is needed for the series at hand and the level of subjectivity associated with the choice of the bias correction method. In Figures R4-R9, panels (a) and (b) show examples of time series for a few randomly picked synthetic samples along with the upper and lower limits and the median serial correlations of 10,000 synthetic samples generated without any bias correction. If the lag-1 serial correlation $\rho_1$ is used to measure the strength of the serial correlation structure, it is important to mention that $\rho_1 < 0.2$ (0.3) for the $Z_i$ of ~90% (97%) of the gages. Gages with values of $\rho_1$ up to 0.3 are reported in Figures R4-R7. For these cases, the autocorrelation function of the synthetic samples does not seem to be affected by any bias. Some bias starts appearing for stronger serial correlation structures, as shown in Figures R8 and R9. However, the time series of Figures R8 and R9 clearly show an increasing trend that induces those strong correlations… While one of the main concerns raised by the author is the subjectivity that is

often adopted to choose trend forms, the reasons why the bias correction was applied is not motivated in the paper, nor it is shown the effectiveness of the bias correction across several strengths of the autocorrelation. I also wonder how, in Figure 9, results look like for the case of IAAFT without bias correction plus false discovery rate (FDR) test.

***Response***
Following the rationale of statistical inference, bias correction is not subjective and is not even an option, but a necessity resulting from the assumption of dependence.
If we *assume* "dependence", we know *a priori* that the estimators of ACF and spectrum are biased when the estimation rely on short samples. This is well known and widely discussed in the cited literature (e.g., Marriott and Pope, 1954; White, 1961; Wallis and O'Connell, 1972; Lenton and Schaake, 1973; Mudelsee, 2001; Koutsoyiannis, 2003; Koutsoyiannis and Montanari, 2007; Papalexiou et al., 2010; Dimitriadis and Koutsoyiannis, 2015; Serinaldi and Kilsby, 2016a).

Let me use an example familiar to the Reviewer (Mascaro 2018, JH). The problem of ACF/spectrum bias is analogous to that of the estimation of the shape parameter of GEV/GP models for short samples. In these cases, estimates might point to apparent exponential tails; however, such a behaviour might be consistent with heavy tailed *population* models. In other words, short samples from heavy tailed models can look "exponential". Therefore, taking the rough estimates as the "truth" might be a mistake.
According to the rationale of statistical inference this does not allow us to conclude that the analyzed process is "heavy tailed" or "exponential" or something else: it just means that under the *assumption* of heavy tails, we can obtain short-sample behaviour coherent with the observations; therefore, if we **assume** heavy tailed models, we **must** correct for the estimation bias.
The take-home message (that the paper attempted to deliver) is that the choice and the use of estimators **depend on** the assumptions we make; we cannot use the same estimator/method under different assumptions, because it might be not valid.

Going back to precipitation data, **under dependence**, empirical ACF and power spectrum are known to be rather unreliable estimators for short samples (in the same way estimators based on product moments might be inferior in terms of bias and variance to L-moments in hydrological frequency analysis). Therefore, we used the climacogram as a benchmark estimator, as it is known to perform better in these circumstances, allowing for bias correction, etc. In other words, the ACFs shown in fig. R4-R9 are like the estimate of the shape parameter of a GEV distribution made by product moments estimators and neglecting bias correction.
The key point, which is systematically missed in the literature, is that the inference on indices/measures of dependence implies that we **assume** (implicitly or explicitly) that dependence does exist, and therefore inference should be made accordingly (accounting for its effects). The same holds for non-stationarity: if we make inference for non-stationary models, we already **assume** that non-stationarity is in play; this means for instance that single population moments (mean, variance and covariance) might not exist. As shown below, estimating the ACF for an observed sample under the assumption of non-stationarity (e.g. under NHP) makes no sense, because **population** ACF depends on time, and the estimate over a sample is not representative of any theoretical counterpart (like the sample average estimated for instance from a sample drawn from a Cauchy distribution).

That said, contrarily to what stated by the Reviewer, Fig. S1 in the supporting material (reported below for convenience) shows the climacogram for two extreme situations that cover the whole

spectrum of cases: Figure S1a,c show a case where bias correction is negligible, while Fig. S1b,d shows an opposite case where the time series looks strongly correlated.

[Figure]

Finally, the Reviewer states "*However, the time series of Figures R8 and R9 clearly show an increasing trend that induces those strong correlations…*"

Saying that the trend induces autocorrelation means **assuming** (implicitly, at least) that the process is non-stationary and autocorrelation is an artifact. This is legitimate. However, what about the other way around? Could not that "trend" be an effect of dependence?

Referring to the series GM00004115 of Fig. S1, the figure below shows some simulations from IAAFT and NHP (with linearly increasing rate of occurrence): Are we sure that the observed data linearly increase? Which approach describes the observed low-frequency fluctuations better?

Based on the analyses reported in Reviewer's report, we should conclude that the second case (stationarity) is not realistic because it does not agree with INAR(1) results. However, the fundamental questions are:

- Can we rely on a model (INAR(1)) that does not even describe the observed marginal distributions?
- is INAR(1) (or whatever specific model) representative of the assumption of stationarity-dependence (and the corresponding infinite models)?
- Are INAR(1) structure and related inference compliant with the consequences of the assumption of dependence?

Please note that I am rather neutral about the use of stationary or non-stationary assumptions, as they are just modelling frameworks commonly used to describe inherently unknown/unrepeatable physical processes.

My criticism is about the lack of coherent and proper use according to the rationale of statistical inference, thus resulting in unscientific conclusions. In other words, using non-stationary models is fully legitimate (of course), but the whole inference procedure should be done accordingly accounting for all the effects of such an assumption, and results cannot be used to discard alternative assumptions, as these generally require other models and alternative inference methods.

[Figure]

Finally, Figure 9 has been updated to consider the case required by the Reviewer (biased ACF with FDR). Of course, the revised figure provides the same message.

**Can the IAAFT model be used to generate the null distribution of a trend test?**
Another concern that I have is whether the IAAFT model is appropriate for generating the null distribution of a trend test. I estimated the slopes of the linear regression φ (which can be considered a proxy of a trend test metric, being a first order expansion of any trend behavior) for 10,000 synthetic samples generated by the (stationary, but correlated) IAAFT, as suggested by the author, and plotted the empirical density function. Then, I did the same with the (stationary uncorrelated) Poisson distribution, as a reference. Panels (c) of Figures R6-R9 show that the null distribution of φ is bimodal for the IAAFT. Under the hypothesis of stationarity, we should expect a symmetric distribution with the mode at φ = 0, like in the trivial case of the Poisson distribution. Thus, distributions like those shown in Figures R8 and R9 raise serious concerns on the power of any trend test based on the assumption of IAAFT distribution. I want also to stress that this aspect is completely neglected by the author, despite the large effort dedicated to criticizing some assumptions and conclusions of the paper of Farris et al. (2021). Consequently, why not dedicating a very small additional effort to evaluate the power of tests using the distribution/model that the author proposes as an alternative to test the null hypothesis of no trend?

*Response*
The null distribution depends on the null assumption and the corresponding models. The bimodality reported in Figs R6-R9 depends on the fact that IAAFT yields "constrained" simulations instead of "typical" simulations. This means that all simulated time series share approximately the same power spectrum and therefore the observed "trendy" patterns (either increasing or decreasing), while zero slopes are less likely for that observed spectrum.

If we relax that assumption and use "typical" simulation (CoSMoS-like, so to speak), we recover unimodality (see figure below). Do things change? No much, because the aim of the paper is not to find inexistent "perfect models", but to show that there is a "world of options" beyond INAR(1) and NHP, and discarding one of them cannot imply proving/disproving stationarity or non-stationarity.

Thus, "*why not dedicating a very small additional effort to evaluate the power of tests?*"
Because:
- tests' outcomes are expected and redundant once we know the behaviour of the theoretical processes behind $H_0$ (here, the matter of fact is not bimodality or unimodality, but variance inflation), and
- the aim of the paper is exactly to show that the same observed behaviour might be described by different frameworks, and discrimination is not possible in this context.

Indeed, in the context of unrepeatable hydroclimatic processes, power analysis has little usefulness, because the observed data might always be described by a virtually infinite number of frameworks and assumptions, which will never be covered by the usually simple/trivial models used for $H_0$ and $H_1$. Moreover, such assumptions refer to models not to physical processes.

Things are different when we refer to designed experiments, where we can control influencing factors and therefore, we can measure power (because we can control $H_0$ and $H_1$), effect size, sample size, etc.

This is the meaning of the discussion in Section 6.2, which summarizes the message reported in previous papers of mine and co-authors.

[Figure]

**Does the IAAFT model capture the variability of the observed linear slopes?**
Finally, since the author presents the IAAFT model as a proper stationary model that can explain the presence of trend in virtue of correlation, then one would expect that it should be able to capture the empirical distribution of the observed slopes of {$Zi$}. I applied the IAAFT model for each gage (without bias correction), generated one synthetic sample, and estimated the slope. I then plotted the empirical PDF of slopes of the observed and synthetic samples. I repeated this nine times to

explore the sampling variability. As shown in Figure R10, the IAAFT model is not able to reproduce the observed sampling variability and, in particular, the larger number of positive slopes.

***Response***
As shown in the figure below, IAAFT *does not capture* the variability of the observed linear slopes as well as NHP *does not capture* the variability of the 'observed' $\rho_1$.
Therefore, if we think that IAAFT (stationarity) is not satisfactory, we should also discard NHP (non-stationarity) for its poor reproduction of $\rho_1$.

[Figure]

[Figure]

Of course, the terms of the problem are a bit different, and correct interpretation requires (… once again) to account for assumptions:

1) Results in Fig. R10 assume that the raw estimates of $\rho_1$ are presentative of the population values, neglecting the problems affecting $\rho_1$ estimators both **under dependence** and **under non-stationarity**.

2) Results in the foregoing figure and Fig. R10 refer to independent at-site simulations, i.e. they do not account for the effect of spatial dependence (which is a reasonable assumption in rainfall fields), which is identical to the effect of temporal dependence, resulting in spatio-temporal 'trends' and 'clusters'. This means that possible local temporal trends are shared by several stations because of spatial correlation. The overall effect of spatial clustering combined with short samples sizes is that we can have asymmetry over some areas (or the whole domain) and/or some time windows.
More generally, the foregoing simulations reasonably underestimate the actual variability of sampling statistics (not only $\rho_1$ and $\varphi$).
Furthermore, we should consider the effect of unknow factors affecting precipitation records, including very basic issues related to data collection, handling, and storage. Such global data should be taken with a pinch of salt. And finally, observed data do not come from models, let alone trivial statistical models.

3) ***Under non-stationarity***, the estimates of $\rho_1$ over 100-size samples do not represent any population counterpart, because the **population** moments of NHP are non-stationary and vary with time. For NHP processes, the figure below shows how $\rho_1$ estimates evolve when we increase the sample size from 50 to 150 years. Why just up to 150 years? Because NHP models with negative slopes $\varphi$ yield smaller and smaller rate of occurrence as time increases, until the rate of occurrence becomes zero, and the distributions degenerate (i.e., they do not exist). This is a typical problem of nonstationary methods when non-stationarity (temporal evolution) does not result from robust physical justifications, but just from curve fitting (... usually, untenable straight lines, as for the NHP).

[Figure]

The fallacious approach to statistical inference (i.e. inverting the role of assumptions and inferential tools) leads to think (incorrectly) that the estimates of $\rho_1$ (via standard estimators) can be used to compare models for which such estimates might be not valid.

On the other hand, proper statistical inference would imply to select an assumption and perform the whole inference coherently, accounting for its effect. As for the GoF tests discussed above, this means to consider dependence and non-stationarity such that:

- **Under dependence**, we have some estimates of $\rho_1$ obtained from proper estimators, accounting for possible bias, etc.
- **Under non-stationarity**, we must be aware that the moments depend on time and any estimate of $\rho_1$ is not representative of any invariant ACF function, which does not exist, being time dependent.

In other words, under dependence and non-stationarity, *population* statistics/properties, such as ACF, have a different meaning and interpretation (and might not even exist). We cannot use the same estimators (usually, valid under i.i.d.), as the resulting estimates refer to different and often incompatible *population* objects, thus making direct comparison technically meaningless.

**Lack of details regarding the estimation of the BB parameters**

The explanation given in the supporting information of how the intra-cluster correlation parameter of the BB is estimated from a time series {$Yj$} is confusing. The meaning of "cluster" is not explicitly provided, while it should be. Based on Ahn and Chen (1995), a cluster should correspond to {$Yj$} in a given year (i.e., $j = 1, 2, …, 365$): is this the case? If so, each cluster has size $n = 365$ (apart from leap years). The author talks instead about "experiments" on specific days (of the year) $j$ and $l$, but they

do not introduce a symbol for the number of available clusters. This should be the size of the vector used to compute the correlation coefficients $\rho_{jl}$. Put simply, if we have $m = 100$ clusters (or years of records), $\rho_{jl}$ is the correlation coefficient between the vectors of the $m$ Y's at day $j$ for all years and the $m$ Y's at day $l$ for all years. To the my best knowledge, this is the proper approach, and since the authors do not provide any detail, I followed it for the calculations made in this review.

**Response**
For $m$ locations and clusters of size $n=365$, the BB overdispersion parameter is just the mean of the lagged cross-correlation values up to lag 365. That's it. This is clarified in the revised version of the Supplement. Moreover, this also clear by reading (more carefully) Ahn and Chen (1995).

**What serial correlations and slopes of {Zi} are generated by stationary and nonstationary {Xj}?**

**Response**
I think that Reviewer's simulations and their interpretation suffer from the effect of the epistemological problems discussed throughout the paper and in the responses above:

i)   An AR(20) model with GG marginals is not even representative of the whole daily rainfall time series recorded in Athens; it was used as a proof of concept to show the reproduction of ACF and marginals for October rainfall.
     Realistic rainfall simulation should account for seasonality, and more importantly for high/low frequency variations at various spatio-temporal scales... it is a bit trickier task than running an AR(20).
     Thinking that such a model, which is not even representative of a single precipitation time series, can be used to discard the assumption of stationarity and the infinite set of corresponding models means iterating the same logical misconception discussed above.

ii)  As mentioned above, $\rho_1$ values have a different meaning and interpretation under the four different assumptions used in the MC experiments. Here the mistake is to think that the same estimator, which is strictly valid for the first set of assumptions, can be used in the other three cases, and that it corresponds to the same population counterpart in all cases. As discussed above, it is not so.

iii) The left and middle panels of Fig. R11 should be compared with panels (b) and (d) of Fig.5 (reported below for convenience), whose interpretation is straightforward: if we account for the effects of dependence, and we make inference accordingly, we obtain a coherent picture showing that:
     a. Poisson marginals are inconsistent with the observed overdispersion, which in turn is instead consistent with dependence.
     b. INAR(1) does not provide a suitable description of $Z$.
     c. If we use a simulation approach coherent with the assumption of dependence, we are able to reasonably reproduce the observed variability (over 1000+ real world series, not just unrealistic AR(20) samples) shown in Fig.R11 (middle panel). Fig. 5b is rather clear in this respect, while Fig.5d shows the difference between the CI obtained by INAR(1) and IAAFT. These results should be read in conjunction with the performance of BB (Fig. 3), which is another model coherent with the assumption of dependence.

[Figure]

**Figure 5.** Scatter plots of the pairs $(\hat{\rho}_1, \hat{\phi})$ for the 1,106 observed $Z$ time series over the 95% threshold and 100 years (1916-2015) along with pairs corresponding to Poisson-INAR(1) samples (a), pairs corresponding to IAAFT samples (b), 95% CIs of $(\phi|\mathrm{P}_1 = \rho_1)$ for IAAFT and NHP (c), and 95% CIs of $(\rho_1|\Phi = \phi)$ for IAAFT and Poisson-INAR(1) (d).

To summarize, we can build a coherent modelling framework under stationarity-dependence that can describe observations over several spatio-temporal scales reasonably well.

**Does this mean that IAAFT or BB are a panacea? No, of course.**
These are just example models used to show that INAR(1) is inappropriate (it does not even describe the marginals) and cannot be used to discard a whole class of stationary-dependent models. This class includes models that can reproduce a variety of patterns much richer than one can think (Mandelbrot, 1982, p. 384).

**Does this mean that precipitation is "stationary"? No, of course.**
Such unrepeatable hydroclimatic processes are neither stationary nor non-stationary! Such concepts only apply to the models we use to describe natural processes.

Everyone can use the modelling framework they like more.
My criticism is about a distorted approach to modeling and inference, which leads to believe that
the (poor) performance of a single model (such as INAR) can be used to conclude that a general
assumption can be discarded, taking implicitly for granted that (i) such a single model is
representative of an infinite class of models, and (ii) it can be used to prove its own assumptions or
different assumptions (under which INAR might not even exist).

The scientific method, summarized at the beginning of this reply, implies an opposite approach,
whereby inference follows assumptions. Different frameworks should be compared at the end of
their own inference, in terms of parsimony, generality, and fit for purposes.

Based on these findings, the assumption of stationarity for the application of the BB to model the
distribution of $Z$ is not supported at all gages. Moreover, the INAR(1) is still useful to assess the
nonstationary of the $Z$ time series derived from daily P records.

**Response**
I repeat, this statement results from switching assumptions with models, distorting the rationale of
statistical inference:

1. The AR(20) model is not representative of a single time series, let alone worldwide
   spatio-temporal variability of precipitation "*at all gages*".
2. INAR(1) does not even describe the marginal distribution of $Z$. It does not represent the
   whole class of stationary models whatsoever, and it cannot say anything about different
   assumptions for which it does not even exist.
3. Figure 5 shows that there might be stationary modeling strategies (different from
   INAR(1)) that tell a different story.
4. Note that Figures 11 and 13 (shown below once again) report BB derived under the
   assumption of dependence and stationarity. I would be glad to see an equally simple,
   general, and parsimonious non-stationary model showing the same goodness-of-fit
   over the same range of spatio-temporal scales (from daily to annual, and from at-site to
   worldwide).

[Figure]

**Figure 11.** ECDFs of number of OT events (for the 95% and 99.5% thresholds) occurring at daily time scale over different regions along
with Binomial and $\beta B$ CDFs.

[Figure]

**Figure 13.** ECDFs of number of OT events (for the 95% and 99.5% thresholds) occurring at annual time scale over different regions along with Binomial and $\beta\mathcal{B}$ distributions.

**Is the BB distribution "good" for {Zi} generated by nonstationary uncorrelated {Xj}?**

The author indicated that the BB is the theoretically correct distribution for serially correlated count series. The experiments conducted above show that trends in $\{X_j\}$ series introduce artificial serial correlations in the corresponding $\{Z_i\}$ even for uncorrelated $\{X_j\}$ (NonStUncor). In these situations, I found that the intra-cluster parameter of the BB can be "successfully" estimated based on the artificial correlations of $\{Y_j\}$. Therefore, since the BB "works", the theoretical considerations mentioned above would erroneously provide confidence in the presence of serial correlation and nonstationarity of the series. The results presented in section 3.2 indicate that this is most likely the case for several gages.

**Response**

*"intra-cluster parameter of the BB can be "successfully" estimated based on the artificial correlations of $\{Y_j\}$."* I guess that the Reviewer has simulated realistic non-stationary binary processes for all 1106 stations, and therefore he has estimated the BB parameters for various spatio-temporal scales, obtaining an equally good or better fit than that shown in Figs. 11 and 13.
If so, it would be interesting to see such results.

However, the matter of fact is different: even if such 1106 non-stationary binary models existed and worked well, what would be the benefit of introducing additional complexity, which is also difficult to theoretically justify?
I recall once again the rationale of statistical inference, which is:

> **Correct approach**
> 1) Make assumptions.
> 2) Build models and make inference accounting for the effect and consequences of those assumptions.
> 3) Interpret results according to the nature of the adopted models and their assumptions.

and ***not***:

> **Fallacious approach**
> 1) Select several models and methods based on different and often incompatible assumptions.

2) Make inference neglecting the constraints imposed by the different assumptions.
3) Interpret the results attempting to prove/disprove the assumptions.

Therefore, I do not use BB to prove "stationarity" or disprove "nonstationarity": this would be scientifically meaningless.
I just follow the foregoing "correct approach":
1- ***Let's assume*** (not "let's prove!") stationarity and dependence. Under such assumptions, can we build a simple, parsimonious, and general modeling framework that describes the observations over a range of scales reasonably well?
2- We select models and make inference fulfilling the assumptions.
3- We compare model/inference output with observations. If the models perform satisfactorily for our purposes, they are usable and valid within the limitations of the original assumptions.
4- If the models do not work, we can try other models under the same assumptions or different assumptions bearing in mind parsimony, generality, and reasonable simplicity.

That's it. We do not try to prove model assumptions or alternative model assumptions (… "model assumptions", not "physical process assumptions"!)

Instead, following the "fallacious approach", this is what most of the literature on these topics does:
1- Take several models under different assumptions.
2- Make inference (e.g., GoF tests and ACF estimation) neglecting the effect of those assumptions on tests, estimators, etc.
3- Discard an assumption based on the performance of a single model, which is obviously far from being representative of the whole class of models corresponding to that assumption.
4- Attribute the retained assumptions to physical processes, whereas they only apply to models.

This fallacious approach is like trying to use the Euclidean geometry to prove or disprove the definition of "point" and "line".
It is well-known that deterministic trends yield spurious 'correlation' and dependence yields spurious 'trends' (based on inappropriate estimators). However, this is irrelevant because precipitation is neither "stationary" nor "non-stationary" and does not come from any model.
One can only use the assumptions that they prefer and check which approach yields the most convenient, parsimonious, and general description for the purposes of interest.
On the contrary, mixing incompatible methods and models attempting to prove assumptions that are incorrectly attributed to physical processes results in confusion, misinterpretation, and misleading conclusions that are inconsistent with scientific reasoning.

The results presented in section 3.2 of Reviewer's report do not indicate anything because they refer to models that do not even represent a single complete precipitation series, and all estimated values suffer from the theoretical inconsistencies discussed above.

**Assumption of stationarity for the BB model applied to spatiotemporal precipitation time series**
The author applies the BB distribution for the counts at multiple sites with spatially and temporally correlated records under the assumption of stationarity of the correlations. Such an assumption has

not been tested in any way by the author, and it might end up being as good or bad (as shown in my comments above) as the assumptions made to test trends that the author criticizes.

In this regard, the CIs in Figure 14 are obtained using the entire record of 100 years, but it could have been instead generated by applying the BB distribution using the first 50 years and results tested with the subsequent 50 years. I assume that other ways to test the assumption of stationarity of the correlations could be designed or, perhaps, found in the literature.

*Response*

The NHP models used by Farris et al. *"are obtained using the entire record of 100 years, but it could have been instead generated by applying the"* NHP *"distribution using the first 50 years and results tested with the subsequent 50 years"*.

Anyway, the figure below shows that the results in Fig. 14 do not change if we estimate the BB overdispersion parameter over the first 50 years. This is not surprising as the underlying spatio-temporal correlation matrices are estimated on daily binary time series of size 365*50 = 18250 rather than 365*100 = 36500, that is, the sample size is enough to obtain reliable estimates.

Instead, the sample size is a problem for the models used by Farris et al. as their parameters are estimated on 100 data points, which are barely enough to get a decent estimation of central tendency measures assuming i.i.d. and well-behaving bell-shaped light-tailed distributions.

[Figure]

That said, Reviewer's remarks suffer from the same problems already discussed in the foregoing responses:
- Observed correlations are not (and cannot be) either stationary or non-stationary.
- We can only observe sampling fluctuations, which should not be confused with population properties.
- The estimation of the correlation itself depends on the assumption of stationarity, dependence, non-stationarity, etc.

- Inference depends on the assumptions behind diagrams, estimators, models, etc. **not vice versa**.
- I do not criticize "*the assumptions made to test trends*" anywhere, because models' assumptions are what they are.
  I criticize the idea that statistical tests can provide information about whatever assumption for unrepeatable processes! Throughout the paper, I actually stress several times that the output of the tests is just what is expected from their underlying *assumptions*.
- If we assume stationarity and dependence, we obtain the kind of fit shown in Figs. 11 and 13 (reported below once again). The merit of such assumptions is that they correspond to simple models that describe the observed behavior over a range of scales.
- Thinking that the model assumptions can be tested on observations of unrepeatable processes (that do not correspond to any model, for sure), means overlooking the rationale of scientific enquiry:
  "*The sciences do not try to explain, they hardly even try to interpret, they mainly make models. By a model is meant a mathematical construct which, with the addition of certain verbal interpretations, describes observed phenomena. The justification of such a mathematical construct is solely and precisely that it is expected to work - that is, correctly to describe phenomena from a reasonably wide area. Furthermore, it must satisfy certain aesthetic criteria - that is, in relation to how much it describes, it must be rather simple.*" (von Neumann 1955).

[Figure]

**Figure 11.** ECDFs of number of OT events (for the 95% and 99.5% thresholds) occurring at daily time scale over different regions along with Binomial and $\beta B$ CDFs.

[Figure]

**Figure 13.** ECDFs of number of OT events (for the 95% and 99.5% thresholds) occurring at annual time scale over different regions along with Binomial and $\beta\mathcal{B}$ distributions.

1. **Other comments:**

5.1: How were the CIs of the IAAFT model obtained in Figure 5?

*Response*
This is explained in L365-375 of the original submission.

5.2: Lines 302-303: How were independent peak-over-threshold events identified?

*Response*
This is clarified in revised text. Please, note that this aspect is irrelevant in the present paper.

5.3: Lines 377-378: The author mentions that the INAR(1) model does not reproduce the autocorrelation of the observed data, but this was never proved.

*Response*
INAR does not reproduce the dependence structure. It reproduces the biased ACF, which is however an unreliable estimator of dependence structure for such short data sets.

5.4: More details are needed to explain how Figure 11 was generated. Also, the values of the estimated parameters of the BB model should be provided.

*Response*
Figure 11 just shows standards probability plots, that is, ECDFs and binomial and Beta-binomial CDFs, where BB overdispersion parameters are the averages of the lagged cross-correlation matrices of the parent binary process *Y*. Making these figure is straightforward (if not trivial, like overlapping ECDF and GEV models of annual maxima) and already described in L489-498 and 516-524 of the original submission.

5.5: Line 539-540: is this a possible explanation of something that could not be proved? In other words, is this as "subjective" as the interpretation based on the presence of non-stationarity?

*Response*
This remark suffers (once again) of the effect of a fallacious approach to data analysis.
The paper does not attempt to prove anything!
Figure 14 only shows that the observed behavior falls within the uncertainty allowed by a stationary spatio-temporal dependent model. Under this _assumption_, apparent trends look consistent with the uncertainty expected under stationarity and dependence. That's it. This does not mean proving or disproving stationarity or nonstationarity or any other model property.

5.6: Line 594-596: I disagree with the author. Even if not perfect in terms of marginal distribution and based on some level of subjectivity on the trend (i.e., the simplest case of linearity), the calculations presented in this review confirm that the INAR(1) and NHP models, adopted by Farris et al. (2019), could be reasonably used as extreme cases to investigate where the statistics of the observed records are located compared to simple stationary and non-stationary models. Of course, better models are welcome to further improve the analyses.

*Response*
The Reviewer misses the key point. The problems of Reviewer's simulations and Farris et al.'s analysis is not the use of some specific (trivial) models, but their misuse.
All numerical results make little sense because they neglect the logic of statistical inference, which comes before any modeling and/or analysis, and allows for discriminating between science and "collecting model outputs":
- if GoF tests are done under stationarity, results are meaningless under nonstationarity! Those results cannot be used to support the choice of INAR or NHP. These tests are not even valid under the underlying assumptions of INAR and NHP!
- Classical ACF estimates are biased under dependence and are meaningless under nonstationarity! Therefore Figure 4 in Farris et al., and Figure R11 in Reviewer's report make no sense.
- INAR(1) and NHP models cannot be used "*to investigate where the statistics of the observed records are located*" because models cannot be used to validate or discard their own assumptions. The attempt to validate or discard assumptions based on model's performance, independently of their degree of "sophistication", contrasts with the epistemology of statistical inference. And INAR and NHP are far from being extreme cases in their respective classes of models. They are only inappropriate because they do not even reproduce the most elementary properties of the observed records.
The actual problem is a fallacious approach to statistical inference that makes any numerical result and graphical comparison informative in the best-case scenario, and scientifically meaningless in the average scenario (i.e., in most of the literature on these topics).

I recognize that the standard rationale of statistical inference is the opposite of Reviewer's idea about data analysis emerging from his report. As many other misconceptions, that flawed approach is so widespread that people take it for granted, without asking themselves if it is consistent with the principles of science.
Therefore, I am quite confident that the Reviewer will not change his mind about these issues, as it would mean calling into question rooted believes that probably have guided his work for years.
Thus, the aim of this paper is not to convince him or anyone else, but just to introduce a doubt about what we take for granted, bearing in mind that taking something for granted is another epistemological mistake, as science is provisional.

1. I welcome this paper, which starts as "Most of the methods reported in handbooks of applied statistics have been developed under the assumption of independence, distributional identity, and well-behaving bell-shaped/exponential distributions" and intents to show the problems these assumptions cause in real-world hydrological applications, and to suggest remedies. I appreciate the investigation of temporal and spatial dependence as causes of spurious trends. I have devoted a book on these issues (Koutsoyiannis, 2023) as I believe they are important and not well known in the hydrological community. Hence, in principle I favour publication of the paper.

2. The paper makes several good points on epistemological grounds, related to concepts not well known or forgotten: the fact that statistical estimates (and diagrams) always rely on an underlying model; the importance of ergodicity for inference; the attention drawing on exploratory data analysis, including graphical diagnostics and theoretical considerations; the clarification of the meaning of stationarity and nonstationarity, which are not properties of the observed time series, but assumptions of the models we devise to describe physical processes.

3. At the same time, the paper tries to debunk some common misunderstandings, such as that dependence has limited effect on statistical inference and that statistical tests, typically devised for independent samples, are serious means of inference for hydroclimatic processes. This may be acceptable for processes with short-range dependence but not for ones with long-range dependence (Cohn and Lins, 2005; Koutsoyiannis, 2023).

4. The paper is a result of a large amount of work and effort, based on analyses of a huge data set and using newly developed methodologies.

5. I particularly liked the graph in Fig. 8 (and similar figures in the Supplement). If I understand it well, there is not even one point with statistically significant trend. I also like the next figures that show that dependence alone can explain the behaviours observed, while independence would result in false trends.

***Response to points 1-5***
I thank the Reviewer for his encouraging feedback. Obviously, several concepts reported in the paper are the same as those reported in various Reviewer's papers published in the last 30 years or so.

The paper is also a sort of critique to some compulsory habit to develop supposedly "new" or "innovative" techniques, which are often just irrelevant, if not flawed, because they are derived neglecting the logic of scientific method.

In this respect, instead of using popular claims such as "We propose a new method" or "We do this or that for the first time", I can say that the paper does not contain anything new! In fact, the aim is to show that supposed "fashionable and apparently sophisticated novelties" can proliferate if we

neglect the principles and the bounds of scientific method. In that way, everything becomes possible like magic (or any other human activity that do not obey the rules of scientific inquiry).

In this study (and some previous ones of mine with co-authors), I attempted to highlight how the main problems affecting a large body of literature on certain topics is not only related to technical choices but also (and more) to a systematic negligence (or lack of knowledge) of the logic of science (and statistics).

6. A question not discussed in the paper is if trend identification has some usefulness or not. Let us assume that we have correctly (whatever this might mean) identified statistically significant trends based on past data. If our focus is on the past, what is the usefulness of a trend, once we have a better description by the data themselves? And if the focus is on the future, what is the value of such trend identification? Will a trend identified in the past continue in the future, or will it bend? Does the trend have any predictive value? By the way, this issue has been investigated by Iliopoulou and Koutsoyiannis (2020; see also the motto in that paper), but I would be interested to know the author's opinion on this. The current trend in detecting pointless trends continues to dominate in literature, and it would be useful and beneficial to the community if the paper discussed this issue.

*Response*
As mentioned in my preliminary response, out-of-sample prediction is surely interesting. In this respect, the inconsistency of "physically ungrounded" nonstationary models has been discussed in some papers, including Iliopoulou and Koutsoyiannis (2020), which is cited in the revised text.

However, as clarified in the revised introduction, the aim of the paper is to stress the importance of epistemological concepts that precede any statistical analysis (in-sample or out-of-sample).

Since I already discussed these issues in Serinaldi and Kilsby (2015), I prefer to keep this study focused on the foregoing epistemological problems.

7. Since the paper contains a sound epistemological part, it would be suitable to give a definition of a trend on a scientific (not colloquial) basis. I believe this notion, is more unclear than is popular and lacks a proper definition. I am interested to see what the author thinks about this issue.

*Response*
An attempt to formalize such a definition is discussed in Serinaldi et al. (2018). Their point of view is now summarized (and elaborated a little bit) in new Section S3 in the Supplement.

8. On the negative side, the paper is difficult to read—and review. While the epistemological parts of the paper and the related questions are well discussed and clear, the technical parts are difficult to assimilate. Near the end of the paper, the author states that he offered an "alternative point of view". However, this is not clarified and is spread in many different sections without coherence. At least a summary of the approach proposed would be helpful and would improve the paper.

*Response*
I hope that the revised Section 1 and the expanded Section 4 can clarify a little bit the rationale of the analyses reported in the paper as well as the meaning of "alternative point of view" (please, refer to the preliminary responses for further details in this respect).

As explained in the revised introduction, technicalities are secondary in this specific paper, and are indeed reported in the Supplement.

On the other hand, to be honest, I think that the technical content will always be a bit hard because some of the methods are typical of "test-based" approach, whereas others are generally applied in "model-based" methods. Commonly, those who use the former approach (and related methods) do not apply the latter, as these two approaches are mutually exclusive from a logical perspective.

This means that a quick understanding of the methodology requires to be rather familiar with both approaches. This does not mean that the methodology is particularly difficult to grasp, but this requires a bit more effort for those who are mainly familiar with just one of the two methods.

Loosely speaking, the review process seems to reflect the foregoing statements. In fact, the panel of Reviewers clearly splits in two groups: "model-based" Reviewers recognized the key epistemological points of the paper but found some "test-based" methods a bit unclear, while the "test-based" Reviewer easily recognized typical methods of "test-based" method but seems to miss the logic of "model-based" approach. The difficulties of the two "groups" mainly reflect the fact that the "test-based" methods can look illogic and often meaningless to "model-based" people, and vice versa.

In my opinion, this is the main reason why this paper may not be easy to read, independently of its form: it discusses two mutually exclusive approaches, which imply two mutually exclusive mindsets to approach data analysis. Therefore, it can be quite "indigestible" for most readers, who generally belong to one of the two groups.

9. The structure of the paper is not optimal. Its style is not didactic, and, at places, it is too technical and unclear. The paper needs a substantial overhaul to make an attractive narrative—otherwise I think it would not be read.

*Response*
I tried to clarify the text to better explain the rationale and structure of the text, as well as the "model-based" procedure. However, as mentioned above and the preliminary response, the problem might be the nature of the paper itself, rather than the presentation. In my experience, papers are rarely read even when they are cited. Indeed, they are often cited incorrectly.

10. A major negative issue is that the paper is aligned with another paper by Farris et al. (2021). It even attempts to present the methodology of that previous paper (section 3.1). In this respect, it does not look as an independent paper but a discussion on another paper. Yet it is not a formal discussion, as the paper by Farris et al. (2021) was published in another journal. I think the present paper reflects a sound work that could justify publication, but it needs reworking to be presented as a stand-alone paper.

*Response*
The revised introduction explains the rationale behind this kind of validation papers, which are not just discussions of other papers but thorough investigations about the validity of methods and procedures.
Please, refer to preliminary responses for further details.

11. It appears that the Hurst-Kolmogorov (HK) behaviour (long-term persistence; long-range dependence) is not present in the main paper but only in the Supplement. It has been shown (Iliopoulou and Koutsoyiannis, 2019) that subsetting of a time series (using thresholds or block maxima) distorts the dependence and may hide that behaviour, but this does not mean its influence disappears. The main statistic used in the paper appears to be the lag one autocorrelation ($\rho 1$), but this does not capture the HK behaviour, so I doubt if it is appropriate. I think this would be useful to discuss in the paper.

***Response***
HK is now mentioned in the revised main text. Concerning $\rho_1$, please refer to preliminary responses.

12. Overall, it would be a pity if this work and the important points it makes were not published. On the other hand, its current form needs substantial improvement, before the paper can be publishable. I am sorry that I am not more specific in my comments, but, as I said, I had difficulties to read the paper.

***Response***
I did my best to improve the presentation. In particular, the new abstract and expanded introduction better explain the actual aim of the paper and the rationale behind its structure.

In this study, the authors pose the question of whether there is a negligible or stronger impact of the spatio-temporal dependence on the frequency of precipitation extremes. In my opinion, the analysis and the discussion are intriguing and I believe it fits the purpose of the journal. Please see some minor comments below that I hope can serve as an open discussion and exchange of scientific opinions between the Reviewer and the Authors:

1) One of the main conclusions of this study is that "... most of this literature resorts to methods based on the same unrealistic assumption of independence and corresponding trivial models...", "On the other hand, when dependence is accounted for, its true consequences on the entire inference procedure are commonly underestimated, partially missed, or neglected.", and "...we focus on the iterated underestimation and misinterpretation of the role of spatio-temporal dependence in statistical analysis of hydro-climatological processes.". Although this conclusion is very common to me, since my main research is based on the long-term persistence, I strongly agree that dependence (and especially in its simplest form of the long-term persistence, which basically includes only 1 key parameter) is often neglected in the modern literature, not being tested at all in some studies (although there are many papers that have shown its existence, from subdaily records to climatic scales; see for example, one of the largest global-scale station-analyses by Dimitriadis et al., 2021, where short- and long-term dependence is traced in all key hydrological-cycle processes; or the work by Koutsoyiannis 2020, where there are extended discussions on how observed random trends could be misleading and can be easily explained by the LTP behaviour, as well as applications and strong traces of LTP to global-scale gridded ground observations, satellite data and reanalyses), and being replaced by others more complicated methods that some have been proven to be unrealistic (like the classical detection of trends, which by definition can support only the past values, and it is almost certain that will fail many times even in the near future). Therefore, I believe that, even solely based on the main analysis and conclusion of this study, this is a robust work that deserves publication.

Dimitriadis, P., D. Koutsoyiannis, T. Iliopoulou, and P. Papanicolaou, A global-scale investigation of stochastic similarities in marginal distribution and dependence structure of key hydrological-cycle processes, Hydrology, 8 (2), 59, doi:10.3390/hydrology8020059, 2021.

Koutsoyiannis, D., Revisiting the global hydrological cycle: is it intensifying?, Hydrology and Earth System Sciences, 24, 3899–3932, doi:10.5194/hess-24-3899-2020, 2020.

*Response*
Thanks. The suggested references are pertinent and have been added.

2) Regarding the motivation of this paper by Farris et al. (2021), I also agree with the other Reviewers that more focus should be given to the main conclusion "Therefore, accounting for the effect of dependence in the analysis of the frequency of extreme precipitation has huge impact that cannot be ignored.". I also understand that there are some differences between the work by Farris

et al. (2021) and the current study, such as that more data-values are used and additional methods are tested, and therefore, differences in the results should be expected, making it difficult to exactly replicate their results (that again, as I stated, there is no need to focus this research on that, but trying, as a unique study, to support independently the main conclusion shown above, while discussing the similarities and differences from other scientists' work in the literature; in this way, I believe the paper could benefit and be more unambiguous to the Readers).

Farris, S., R. Deidda, F. Viola, F., and G. Mascaro, On the role of serial correlation and field significance in detecting changes in extreme precipitation frequency,Water Resources Research, 57, e2021WR030 172, 2021.

***Response***
Thanks. As mentioned in the preliminary reply, the paper contrasts the standard (epistemologically sound) approach to data analysis (denoted as "model-based") with an epistemologically inconsistent approach (denoted as "test-based"), which is however widespread in hydro-climatology. The study exactly replicates Farris et al. (2021)'s results to compare them with those obtained by applying the model-based approach. The aim is to show that "how" we build inference and apply statistical methods (and any quantitative method) comes before and is more important than the specific techniques we use. Any method is misleading if it is applied without following the logic of science. The paper attempts to highlight the very different interpretation and conclusions resulting from the two approaches mentioned above.

I hope that the new introduction and changes in the text can better explain the problems that I'm focusing on, and why I chose a critical style based on side-by-side comparisons with existing literature.

3) One of the main highlights of the current study is the clear separation between the model options independence-nonstationarity and dependence-stationarity in extreme precipitation since a mix between nonstationatity and dependence could lead to erroneous results due to the validation of the ergodicity property. For example, I much enjoyed statements like "Similar remarks hold for nonstationarity. Dealing with it does not mean just adding time dependent parameters to a stationary model, using for instance Generalized Linear Models (GMLs) and their available extensions: it means that the ergodicity property, which is key in the interpretation of statistical inference, is no longer valid. In these cases, any estimate of whatever summary statistics, such as the sample mean, is uninformative as it does not have a corresponding unique population counter part, as the latter does not exist anymore." or "Stationarity and nonstationarity are not properties of the observed hydro-climatic processes (finite observed time series), but assumptions of the models we deem suitable to describe physical processes." or "... the underlying question is whether possible monotonic fluctuations are deterministic (resulting from a well identifiable mechanism) or stochastic (as an effect of dependence, for instance). In the former case, we work under the assumption of independence and nonstationarity, whereas in the latter under the assumption of dependence and stationarity.", since this is a very common misinterpretation in the literature (several such studies are cited in the paper), and so, the highlighting of such issues is very useful. I

would also recommend looking Iliopoulou and Koutsoyiannis (2019; 2020) works, which I think are very much related and focused on similar subjects and conclusions.

Iliopoulou, T., and D. Koutsoyiannis, Revealing hidden persistence in maximum rainfall records, Hydrological Sciences Journal, 64 (14), 1673–1689, doi:10.1080/02626667.2019.1657578, 2019.

Iliopoulou, T., and D. Koutsoyiannis, Projecting the future of rainfall extremes: better classic than trendy, Journal of Hydrology, 588, doi:10.1016/j.jhydrol.2020.125005, 2020.

*Response*
Thanks. The suggested references are relevant and have been added in the main text.

4) Regarding the statements "This approach allows us to highlight the actual impact of spatio-temporal dependence and finite sample size on statistical inference, resulting in over-dispersed marginal distributions and biased estimates of dependence properties, such as autocorrelation and power spectrum density. These issues also affect the outcome and interpretation of statistical tests for trend detection." or "Generally, dependence implies information redundancy and reduced effective sample size, along with variance inflation and bias of standard estimators of summary statistics such as marginal and joint moments.", etc., please see if found interesting and helpful, the study by Koutsoyiannis (2004), which includes an analysis on the marginal-estimation-bias of moments and how it can hide the true nature of the distribution, and the study by Dimitriadis and Koutsoyiannis (2015), where is theoretically/mathematically derived (and linked to the climacogram, thus uniting all three metrics) the dependence-estimation-bias and impact to size-sample of autocorrelation and power-spectrum-density, while shown how one can handle it to perform robust estimations for interpretation and simulation, and thus tackling variance inflation (in any scale) that "... affects any sampling summary statistics, including sample variance and auto-/cross-correlation."

Dimitriadis, P., and D. Koutsoyiannis, Climacogram versus autocovariance and power spectrum in stochastic modelling for Markovian and Hurst–Kolmogorov processes, Stochastic Environmental Research & Risk Assessment, 29 (6), 1649–1669, doi:10.1007/s00477-015-1023-7, 2015.

Koutsoyiannis, D., Statistics of extremes and estimation of extreme rainfall, 1, Theoretical investigation, Hydrological Sciences Journal, 49 (4), 575–590, doi:10.1623/hysj.49.4.575.54430, 2004.

*Response*
Thanks. The suggested references are relevant and have been added in the main text accordingly.

5) Regarding the trace of long-term persistence and its impact on the high-order moments of skewness and kurtosis and the tail-heaviness that "imply possible non existence of the moments of high order as well as bias and/or high variability in the estimates of the moments themselves, including variance, covariance, and autocorrelation, as well as long range dependence", if the

authors find it useful, please see the recently introduced K-moments by Koutsoyiannis (2023) that are found to tackle many of the, aforementioned by the authors, issues, as also shown in Dimitriadis et al. (2021) applications to a massive big-data analysis of subdaily water-cycle processes (see for example, Fig. 10 in this analysis).

Koutsoyiannis, D., Stochastics of Hydroclimatic Extremes - A Cool Look at Risk, Edition 3, ISBN: 978-618-85370-0-2, 391 pages, doi:10.57713/kallipos-1, Kallipos Open Academic Editions, Athens, 2023.

**Response**
Thanks. The suggested references are relevant and have been added in the main text accordingly.

---

## Author Response (AR2)

**Scientific logic and spatio-temporal dependence in analyzing extreme precipitation frequency: Negligible or neglected?**

**By F. Serinaldi**

**Submitted to *HESSD***

***MS-NR: hess-2023-293***
* * *
**REPLY**

I would like to thank the handling Editor, Dr. N. Peleg, and Referee #3, Dr. P. Dimitriadis, for their additional remarks, which have been addressed in the revised version of the manuscript.

Yours sincerely,

Francesco Serinaldi